# Wave–sea-ice interactions in a brittle rheological framework

Guillaume Boutin[1], Timothy Williams[1], Pierre Rampal[2,1], Einar Olason[1], and Camille Lique[3]

[1]Nansen Environmental and Remote Sensing Center, and the Bjerknes Center for Climate Research, Bergen, Norway
[3]Univ. Brest, CNRS, IRD, Ifremer, Laboratoire d'Océanographie Physique et Spatiale, IUEM, Brest 29280, France
[2]CNRS, Institut de Géophysique de l'Environnement, Grenoble, France

**Correspondence:** Guillaume Boutin (Guillaume.boutin@nersc.no)

**Abstract.** As sea ice extent decreases in the Arctic, surface ocean waves have more time and space to develop and grow, exposing the Marginal Ice Zone (MIZ) to more frequent and more energetic wave events. Waves can fragment the ice cover over tens of kilometres, and the prospect of increasing wave activity has brought a recent interest on the interactions between wave-induced sea ice fragmentation and lateral melting. The impact of this fragmentation on sea ice dynamics, however, remains mostly unknown, although it is thought that fragmented sea ice experiences less resistance to deformation than pack ice. Here, we introduce a new coupled framework involving the spectral wave model WAVEWATCH III and the sea ice model neXtSIM, which includes a Maxwell-Elasto Brittle rheology. This rheological framework enables the model to efficiently track and keep a memory of the level of damage of sea ice. We propose that the level of damage of sea ice increases when wave-induced fragmentation occurs. We use this coupled modelling system to investigate the potential impact of such local mechanism on sea ice kinematics. Focusing on the Barents Sea, we find that the decrease of the internal stress of sea ice resulting from its fragmentation by waves results in a more dynamical MIZ, in particular in areas where sea ice is compact. Sea ice drift is enhanced for both on-ice and off-ice wind conditions. Our results stress the importance of considering wave–sea-ice interactions for forecast applications. They also suggest that waves likely modulate the area of sea ice that is advected away from the pack by the ocean, potentially contributing to the observed past, current and future sea ice cover decline in the Arctic.

## 1 Introduction

The interactions between ocean surface waves and sea ice have been receiving a significant amount of attention in recent years, particularly motivated by the decreasing Arctic sea ice extent (Meier, 2017) resulting in larger areas of open water exposed to the wind and thus available for wave generation. As a consequence, wave events in the Arctic are expected to be more frequent and more intense (Thomson and Rogers, 2014), with waves penetrating far into the ice cover, and breaking large sea ice plates into floes of less than a few hundred metres (see e.g. Langhorne et al., 1998; Collins et al., 2015). The attenuation of waves by sea ice, however, limits this fragmentation to the interface between the open ocean and the pack ice, in the so-called Marginal Ice Zone (MIZ). The MIZ is a highly dynamic area characterized by strong interactions between the ocean, sea ice and atmosphere. State-of-the-art sea ice models used in climate prediction systems have been shown to fail at representing the complexity of these interactions, having their biggest errors in the MIZ (Tietsche et al., 2014). On shorter time-scales, large uncertainties remain in the forecasts of the position of the sea ice edge (Schweiger and Zhang, 2015; DeSilva

and Yamaguchi, 2019), whereas this information is essential for the safety of the increasing number of human activities in polar regions (Yumashev et al., 2017). These inaccuracies can certainly be attributed (at least in part) to the lack of representation of some of the processes occurring in the MIZ, and the impact of the waves on sea ice dynamics is one of them.

Waves can impact sea ice dynamics in the MIZ through a variety of processes. For instance wave attenuation transfers momentum from waves to sea ice through the wave radiation stress (WRS Longuet-Higgins, 1977), which acts as a force that pushes the sea ice in the direction of the incident waves. Being mostly directed on-ice, its main effect is to maintain a compact sea ice pack near the ice edge, but its importance is still discussed. Estimating wave attenuation from SAR images, Stopa et al. (2018b) found it to be as important as the wind stress over the first 50 km of the MIZ in the Southern Ocean, whereas Alberello et al. (2020) do not observe any wave-induced sea ice drift in pancake ice in the Southern Ocean from *in situ* measurements, despite a strong wave-in-ice activity. Fragmentation is also likely to change the mechanical properties of the ice, but the evolution of dynamical and mechanical properties of a sea ice cover with the floe size remains poorly understood.

Intuitively, we expect that broken ice will be more mobile than continuous ice (e.g McPhee, 1980), having lower internal stress. This seems to be consistent with the deformation observations of Oikkonen et al. (2017) collected by ship radar during the N-ICE-2015 expedition. In their observations, deformation in fragmented sea ice was an order of magnitude higher than in the pack. However, in the absence of routinely available datasets providing synoptic information on sea ice drift, wave height, and floe size in the MIZ, observations do not allow us to arrive at any explicit relationship between the level of fragmentation and the ability of sea ice cover to be deformed.

A few attempts have been made to relate floe size to sea ice dynamics in theoretical models. Shen et al. (1986) used a collisional stress term accounting for floe size to represent sea ice behaviour in the MIZ. The fluctuations of the velocity field they obtain, however, did not reproduce observations from the MIZEX campaigns, being too small by an order of magnitude. Feltham (2005) uses a similar collisional stress but allows the velocity fluctuations to be time-dependent. He shows that it enables the generation of ice jets in the MIZ, but on smaller scales than the reported observations of ice jets by Johannessen et al. (1983). Dynamics in state-of-the-art sea ice models however do not account for the floe size. Instead, the region of the MIZ where sea ice behaves almost in free drift is mostly function of sea ice concentration. The potential impact of fragmentation in compact ice is therefore neglected, whereas regions of low sea ice concentration and high wave activity do not necessarily coincide (Horvat et al., 2019). Vichi et al. (2019) analyse a cyclone and show how sea ice in the Antarctic MIZ is visibly deformed on sea ice concentration maps despite being highly compact. They state that this behaviour could not be reproduced using a concentration-only criterion to distinguish dynamically pack ice from the MIZ, stressing the need to account for other properties like the floe size. Linking floe size, or fragmentation, to sea ice dynamics remains however a challenging task, because of the few data available firstly, but also due to the poor understanding of the way wave propagate in the MIZ, although modelling progress have been made in this particular field.

Modelling efforts relating to waves-in-ice have made wave models progress a lot in recent years, although the heterogeneous nature of sea ice and the wide variety of wave attenuation processes in the MIZ still make wave prediction in ice highly challenging (Thomson et al., 2018; Squire, 2018). The importance of each wave attenuation process varies with waves and sea ice properties. Scattering for instance is efficient to attenuate short waves in fragmented sea ice covers made of consolidated

floes (Wadhams et al., 1986; Montiel et al., 2016), while dissipative mechanisms, like under-ice friction, are expected to dominate in the case of forming ice and long swells propagating in the pack. A sequence of reviews by Squire et al. (1995) and Squire (2007, 2020) gives a more detailed history of this area of work. Liu and Mollo-Christensen (1988); Collins et al. (2015) have stressed the importance of the floe size on wave attenuation. These reports have motivated the implementation of interactions between waves and floe size, through fragmentation, in numerical models. First studies used one-dimensional models to look at the feedbacks between ice break-up and wave attenuation (Dumont et al., 2011; Williams et al., 2013a, b). These models assume that break-up occurs if wave-induced flexural stress overcomes sea ice strength, and the resulting Floe Size Distribution (FSD) follows a truncated power-law, with its upper-limit (often called maximum floe size, $D_{\max}$) depending on the wave field (Dumont et al., 2011). This assumption on the shape of the FSD is based on the observations by Toyota et al. (2011) of power-law FSD in the MIZ that they explain by successive fragmentation of floes by waves. More recently, Boutin et al. (2018) included a parameterization in the spectral wave model WAVEWATCH III (WW3: The WAVEWATCH III Development Group, 2019), also assuming a power-law FSD. Their parameterization enables interactions between sea ice floe size and wave attenuation processes like scattering and inelastic dissipation, and was shown to explain well the wave height evolution during the ice break-up event reported by Collins et al. (2015). Ardhuin et al. (2018) evaluated this model by comparing their results to remote sensing and field measurements during a storm event in the Beaufort Sea, showing good agreement for the measured and modelled wave-in-ice attenuation and broken sea ice extent. Sea ice representation in these wave models remains however too simplistic to investigate deeper the impact of waves on sea ice. It has led to the development of coupled wave-ice models, firstly using 1-D wave-in-ice models (Williams et al., 2017; Bennetts et al., 2017; Bateson et al., 2020; Roach et al., 2018), and more recently more complex spectral wave models like WW3 (Boutin et al., 2020; Roach et al., 2019). These developments were made possible by the implementation of wave-ice interactions in state-of-the-art sea ice models, and in particular representations of the FSD (Zhang et al., 2015; Horvat and Tziperman, 2015). These FSDs have been mostly used to investigate the effects of lateral melting on sea ice properties over timescales of a few weeks to a few years, in a context where lateral melt is expected to be enhanced by the wave-induced sea ice fragmentation (Asplin et al., 2012).

The dynamical aspects of wave-ice interactions have received less attention. Boutin et al. (2020) found that the WRS could regionally impact sea ice melt and sea surface properties in the Arctic MIZ at the end of summer. Concerning the impact of wave-induced fragmentation, Rynders (2017) suggests combining the classical elasto-visco-plastic rheology used in most sea ice models with a granular rheology in the MIZ to better represent floe-floe interactions. This granular rheology depends on the floe size. Numerical simulations with this approach show an overall increase of the sea ice drift speed in the Arctic all year round compared to a reference simulation using a standard version of the sea ice model CICE (Hunke, 2010). Williams et al. (2017) suggest another approach to relate sea ice dynamics to wave-induced sea ice fragmentation using the sea ice model neXtSIM (neXt generation Sea Ice Model, Rampal et al., 2019) in a stand-alone set-up . The elasto-brittle rheology (EB, Girard et al., 2011) used by neXtSIM stores a variable for sea ice, called damage, which tracks the level of mechanical damage of the sea ice over each grid cell (Bouillon and Rampal, 2015; Rampal et al., 2016). The higher the damage is, the lower the sea ice internal stress is, and resistance to deformation is a function of both sea ice concentration and floe size. The originality of the study by Williams et al. (2017) was to link the damage variable with wave-induced fragmentation, making the extension of the

ice region behaving in free drift dependent on the wave field. This was done by linking the damage variable with wave-induced fragmentation. Using idealized simulations of waves compressing ice, they showed that the movement of the ice edge was not very sensitive to either wave fragmentation or the WRS. The investigations of Williams et al. (2017) were however limited to very idealized cases, and the EB rheology in neXtSIM has now been upgraded to the Maxwell-Elasto-Brittle (MEB) rheology (see Dansereau et al., 2016), greatly improving sea ice deformation in pack ice Rampal et al. (2019). This upgrade could also affect the MIZ, as it led to the removal of an ice pressure term that was added to prevent damaged ice from piling up in EB, but which caused the modelled deformations to deteriorate too much with MEB.

In this paper, we present results obtained with a new coupled wave–sea-ice modelling system (WW3-neXtSIM). This modelling system benefits from recent wave-ice developments in WW3 (Ardhuin et al., 2016; Boutin et al., 2018; Ardhuin et al., 2018) and extends the work done by Williams et al. (2017) in neXtSIM. We again use the damage variable to link the sea ice dynamics and the fragmentation due to waves, allowing us to represent the link between the wave-induced fragmentation of sea ice and its mobility in MIZ areas. Our model also benefits from the advancements of FSD implementations in sea ice models done by Zhang et al. (2015) and Horvat and Tziperman (2015), and of the coupling of WW3 with the sea ice model LIM3 described by Boutin et al. (2020). We also propose a way to incorporate some floe size memory of previous fragmentation events due to waves by introducing two time-evolving FSDs in each grid cell. These developments provide a coupled wave–sea-ice framework able to provide all-year-round regional or pan-Arctic simulations. After describing the details of our implementation, we evaluate our new coupled framework to check that the produced wave attenuation, broken sea ice extent, and refreezing timescales are reasonable. We then investigate the effects of wave-induced sea ice fragmentation using a regional case study. Finally, we discuss the different assumptions made in our study, and suggest perspectives for future studies.

## 2 Implementation of the coupling between the wave and sea ice models

In this study we make use of the spectral wave model WAVEWATCH III® (The WAVEWATCH III Development Group, 2019), building on the previous developments performed by Boutin et al. (2018) who included an FSD in WW3 as well as some representations of the different processes by which sea ice can affect the propagation and modulation of waves in the MIZ. These attenuation processes are scattering (which redistributes the wave energy without dissipation), friction under sea ice (with a viscous and a turbulent part depending on the wave Reynolds number), and inelastic flexure. All these processes depend on sea ice thickness, concentration, and floe size. Wave attenuation increases with thickness and concentration, and tends to decrease when the floe size is lower than the wave wavelength, as floes are not flexed anymore. This parameterization was chosen as it was shown to reproduce well wave attenuation in two different events in the Arctic: waves breaking a continuous sea ice cover near Svalbard as reported by Collins et al. (2015), and waves propagating in forming ice in the Beaufort Sea (Ardhuin et al., 2018). As in the study by Ardhuin et al. (2018), we assume that deviations from the ice-free wave dispersion relationship induced by the presence of ice are small and can be neglected. This is likely to be the case once sea ice has been broken (Sutherland and Rabault, 2016).

The sea ice model we use for this coupling is neXtSIM (Rampal et al., 2019), in which an FSD is first implemented as described in Section 2.2. The two models are coupled through the coupler OASIS-MCT (Craig et al., 2017). Figure 1 shows the variables that are exchanged. To give a brief summary: WW3 determines if the waves will break the ice and calculates a representative wavelength $\lambda_{\mathrm{break}}$ in the manner of Boutin et al. (2018). This is then used by neXtSIM to modify the FSD as described below in Section 2.2.2. WW3 also computes the WRS which is used in the momentum equation of neXtSIM. neXtSIM gives WW3 the sea ice concentration and thickness, and the mean and maximum floe size, which are used by WW3 to determine the amount of attenuation. The mean floe size is used to determine the amount of scattering, while the maximum floe size is used to determine the amount of dissipation due to inelastic attenuation. The evolution of the FSD and the computation of the exchanged variables are described in more details in the following paragraphs. Parameters introduced in this section are summarised in Table A1.

## 2.1 Wave radiation stress on sea ice

As in Boutin et al. (2020), the WRS is computed in WW3 and sent to the ice model. This computation provides an estimate of the WRS which is likely to be an upper-bound of its real value, as it assumes that all the momentum lost by attenuated waves is transferred to sea ice, therefore ignoring a potential partitioning of this momentum transfer between the ocean, sea ice and atmosphere. In neXtSIM, the WRS is added to the sea ice momentum equation in the same way as in Williams et al. (2017) study. As discussed by Williams et al. (2017), the estimation of the WRS and its distribution in the MIZ strongly depend on the parameterization chosen for the wave attenuation.

## 2.2 Floe size distribution modelling

As mentioned in the introduction, this study makes use of two different FSDs to represent the evolution of the floe size. The first FSD represents the evolution of the floe size considering floes as pieces of ice separated from each other by leads or cracks. This is the FSD that would be seen by a satellite image, or an an aerial photography. In this FSD, floe size growth is governed by mechanisms like surface refreezing and floe welding (as in e.g. Roach et al., 2018, 2019). In freezing conditions, sea ice forming at the surface and floes welding together can therefore turn fragmented small floes into a continuous ice cover (*i.e.* not seperated by leads) in a few hours to a few days. This FSD is particularly relevant for thermodynamical processes like lateral melting, which are likely to be unaffected by the mechanical properties of the ice cover. In neXtSIM, this FSD is represented by the variable we call "fast-growth" FSD $g_{\mathrm{fast}}(D_{\mathrm{fast}}, \boldsymbol{x}, t)$, where $\boldsymbol{x}$ is the position vector, $t$ time, and $D_{\mathrm{fast}}$ defined as the mean caliper diameter of the floes separated from each other by leads or cracks, as introduced by Rothrock and Thorndike (1984).

The second FSD considers the floe size as the length scale associated with mechanically homogeneous pieces of ice, whether they are welded with other floes by thinner ice or not. In this second FSD, the floe size grows more slowly than for the first FSD, as it takes time for the ice joining the consolidated floes to thicken. The timescale associated with the consolidation is certainly more similar to the mechanical "healing" of sea ice in neXtSIM described in Rampal et al. (2016), with values of $\simeq$

10-30 days. In neXtSIM, this FSD is represented by the variable we call "slow-growth" FSD $g_{\text{slow}}(D_{\text{slow}}, \boldsymbol{x}, t)$ , where $D_{\text{slow}}$ is defined as the mean caliper diameter of the floes considered as mechanically homogeneous pieces of ice.

The picture in Figure 2 illustrates the different definitions of the floe size in our study. On the one hand, sea ice concentration is about 100% with a continuous sea ice cover, represented by the fast FSD with unbroken floes. Processes like lateral melting are unlikely to occur in these conditions. On the other hand, consolidated floes in Figure 2 are easy to distinguish from the thin ice joining them. The "slow-growth" FSD represents the distribution of sizes of these consolidated floes. In this case, the "slow-growth" FSD will be dominated by small floes (of the order of ten metres). This information can be useful for the study of mechanical processes. For example, it can represent the inhomogeneous nature of the ice cover, which is particularly relevant for wave attenuation processes like scattering and flexure-induced dissipation, for which the mechanical properties and ice thickness continuity of the ice cover are the quantities of interest.

The introduction of this second "slow-growth" FSD is motivated by the fact the sea ice cover, as a dynamical system, exhibits memory properties that can be illustrated by e.g. scaling laws of deformation in both time and space (Rampal et al., 2008; Marsan and Weiss, 2010). An ice cover such as illustrated in Figure 2 is likely to have a very different mechanical strength/response under external stresses compared to a consolidated continuous ice cover, due to the high likelihood of break-up of the thin ice between the consolidated floes. In the case of a new wave event, the fragmentation of the ice cover is likely to occur at its weakest points, hence the thin ice joints. There have been very few reports of sea ice break-up events in the litterature (Liu and Mollo-Christensen, 1988; Collins et al., 2015; Kohout et al., 2016), but our intuition is supported by the recent observations of Kohout et al. (2016) in the Antarctic who report waves breaking ice preferentially at refrozen cracks and pressure ridges. Introducing the "slow-growth" FSD in our model is therefore a way to keep a memory of the floe size resulting from the last wave-induced fragmentation event in the model.

These two FSDs are implemented in neXtSIM as areal FSDs (as done by e.g. Zhang et al., 2015; Roach et al., 2018; Bateson et al., 2020; Boutin et al., 2020), which are defined as:

$$\int\limits_{D}^{D+dD} g(D, \boldsymbol{x}, t) dD = \frac{1}{A} a_D(D, D + dD), \qquad (1)$$

and

$$\int\limits_{0}^{\infty} g(D, \boldsymbol{x}, t) dD = 1. \qquad (2)$$

In these definitions, $g(D)$ represents an FSD with $D$ being its associated floe size, $A$ is the total area considered around the position $\boldsymbol{x}$ at a time $t$, and $a_D$ are the areas within $A$ covered by sea ice with floes with diameters between $D$ and $D + dD$. The value $D=0$ corresponds to open water, and Eq. 2 is equivalent to $\int_{0+}^{\infty} g(D, \boldsymbol{x}, t) dD = c$, $c$ being the sea ice concentration. In practice, we have a number $N$ of FSD bins of constant width $\Delta D$, with edges between a minimum and and a maximum floe size, respectively $D_0$ and $D_N$. Thus, floes with sizes in $[D_0 \, D_1]$ cannot be broken into smaller pieces, and we refer to floes with sizes in $[D_{N-1} \, D_N]$ as unbroken floes. Using fixed-width bins may bias our ability to represent or examine scale-invariant

behaviour (Stern et al., 2018), but it has the advantage of being simple, and the study of the FSD evolution and its impact on sea ice is outside the scope of this study.

Both FSDs evolve as in Roach et al. (2018) or Boutin et al. (2020), following:

$$\frac{\partial g(D)}{\partial t} = -\nabla \cdot (\mathbf{u}g(D)) + \Phi_{\text{th}} + \Phi_{\text{m}}, \tag{3}$$

in which $\mathbf{u}$ corresponds to the sea ice velocity vector, $\Phi_{\text{th}}$ is a redistribution function of floe size due to thermodynamic processes (i.e. lateral growth/melt), and $\Phi_{\text{m}}$ is a mechanical redistribution function associated with processes like fragmentation, lead opening, ridging, and rafting. In our implementation, we assume that the only mechanical process modifying the shape of the FSD is the wave-induced sea ice fragmentation, and $\Phi_{\text{m}}$ is therefore the redistribution term associated with this process. The advection terms for both FSD are identical, and similar to what is done for other conservative sea ice properties in neXtSIM. The terms $\Phi_{\text{th}}$ and $\Phi_{\text{m}}$ differ between $g_{\text{fast}}$ and $g_{\text{slow}}$ and are described below.

### 2.2.1  Lateral sea ice melt/growth

In this section, we describe the implementation of the terms $\Phi_{\text{th,slow}}$ and $\Phi_{\text{th,fast}}$ that represent the thermodynamical redistribution of floes associated with lateral melt/growth in each FSD. The evolution of the FSD due to ice growth and melt processes is first performed in the "fast-growth" FSD, and is quite similar to the implementation described by Roach et al. (2018):

$$\Phi_{\text{th,fast}} = -2G_r \left( -\frac{\partial g_{\text{fast}}}{\partial D} + \frac{2}{D}g_{\text{fast}} \right) + \delta(D - D_N)\dot{c}_{\text{new}} + \beta_{\text{weld}} \tag{4}$$

where $G_r$ is the lateral melt rate of floes, $\dot{c}_{\text{new}}$ is the rate of formation of new ice, and $\beta_{\text{weld}}$ is the FSD redistribution term associated with welding of floes, using the Smoluchowski equation as implemented by Roach et al. (2018).

Lateral melting is implemented following Horvat and Tziperman (2015) and Roach et al. (2018). Here, we neglect lateral melt for the largest floe size category as floes with size $O(100)$ m and more are not resolved in this study and are expected to have very little contribution to lateral melt. We also do not make any distinction between what they call the "lead region" and the "open water fraction" of each grid cell, which means the factor called $\phi_{\text{lead}}$ in Roach et al. (2018) is taken to be 1. Note that lateral melt is included in the model in this study but will not be discussed here, as we focus on the impact of waves on sea ice dynamics during a time period dominated by freezing.

In contrast to Roach et al. (2018), sea ice is assumed to be unbroken when initialised in our model, and there is therefore no need for an explicit thermodynamical lateral growth due to the agglomeration of frazil ice at the edge of existing floes. If, after a wave-induced fragmentation event, the sea ice concentration reaches 1 in freezing conditions, it is assumed that the newly formed sea ice is filling all the leads, creating joints between the floes, and the "fast-growth" FSD is therefore redistributed so that all ice is considered to be made of unbroken floes.

The growth of small floes resulting from wave-induced fragmentation in our model is also ensured by welding, which is shown by Roach et al. (2018) to generate a lateral growth rate one order of magnitude higher than that arising from the lateral accumulation of frazil ice. We, however, found the algorithm they use to be very dependent on the choice of the FSD

categories. After some discussion with the authors of Roach et al. (2018), we decided to carry on with this formulation but with an appropriate tuning of the coefficient that Roach et al. (2018) call $\kappa$, which represents the rate at which the number of floes decreases due to welding per surface area. We tune $\kappa$ so that the timescale at which a delta-function FSD made of the smallest floes allowed in the model evolves into a delta-function FSD made of the biggest possible floes is similar in our model and in the one by Roach et al. (2018). To give an idea of the time-scales involved, the model of Roach et al. (2018), starting from a delta-function FSD only made of floes with an average size of 20 m, ends up with half of the ice cover being made of floes larger than 200 m in about 5 days within compact sea ice ($c = 0.95$). In the simulations presented in this study, setting $\kappa = 5 \times 10^{-8}$ m$^{-2}$s$^{-1}$ reproduces a similar evolution of the FSD for our choice of FSD categories.

As mentioned earlier, the redistribution of the "slow-growth" FSD due to lateral growth in $\Phi_{\mathrm{th,slow}}$ is expected to happen on longer timescales, related to the time needed by the fractures to heal. This healing phenomenon is related to the thickening and the consolidation of the joints of which formation is described in $\Phi_{\mathrm{th,fast}}$. It is very similar to the "damage healing" already included in neXtSIM (see Rampal et al., 2016), associated with a timescale $\tau_{\mathrm{heal}}$, that we reuse in the computation of $\Phi_{\mathrm{th,slow}}$ following:

$$\Phi_{\mathrm{th,slow}} = \frac{1}{\tau_{\mathrm{heal}}} \big( g_{\mathrm{fast}}(D_{\mathrm{fast}}) - g_{\mathrm{slow}}(D_{\mathrm{slow}}) \big) + \Phi_{\mathrm{conservation}}, \tag{5}$$

where $\Phi_{\mathrm{conservation}}$ is an *ad hoc* term ensuring the conservation of sea ice area (i.e Eq. 2) for the "slow-growth" FSD when sea ice is created or melted. The "slow-growth" FSD therefore relaxes to the "fast-growth" FSD over a time $\tau_{\mathrm{heal}}$, representing the (slow) strengthening of the joints between the floes. Note that this healing only occurs if the sea ice is exposed to freezing conditions. If new sea ice is formed, all the new ice is added to the largest floe size category, as in the "fast-growth" FSD. If sea ice is melting, we assume that lateral melting has little effect on the shape of the FSD. In this case, $\Phi_{\mathrm{conservation}}$ is used to scale the "slow-growth" FSD without changing its shape.

### 2.2.2 Wave-induced sea ice fragmentation

In this section, we describe the implementation of the terms $\Phi_{\mathrm{m,slow}}$ and $\Phi_{\mathrm{m,fast}}$ that represent the mechanical redistribution of floes associated with the fragmentation of sea ice by waves in each FSD.

Similar to the work by Boutin et al. (2020), the occurrence of sea ice fragmentation in our coupled system is decided in the wave model. In WW3, sea ice breaks up if the wave curvature induces a stress that exceeds the sea ice flexural strength. The shortest wave wavelength for which the wave-induced stress exceeds the critical stress for flexural failure, that we call $\lambda_{\mathrm{break}}$, is passed to neXtSIM (see Fig. 1) and used in the mechanical redistribution scheme of the FSD. The determination of the value of $\lambda_{\mathrm{break}}$ in WW3 is explained in details in section 2.3 of Boutin et al. (2018) (where it is called $\lambda_{i,\mathrm{break}}$). If no fragmentation has occurred in WW3, neXtSIM receives $\lambda_{\mathrm{break}} = 1000$ m, which corresponds to the default unbroken value, and no fragmentation occurs in neXtSIM (resulting in $\Phi_{\mathrm{m,slow}} = 0$ and $\Phi_{\mathrm{m,fast}} = 0$ ). If neXtSIM receives a value of $\lambda_{\mathrm{break}} < 1000$ m, then

FSD redistribution occurs in neXtSIM. Fragmentation occurrence is then determined every coupling time step.

If fragmentation occurs, we assume that the thin ice joining aggregated floes is very likely to break, as reported by Kohout et al. (2016). This is where the "memory" of previous cracks stored in the "slow-growth" FSD plays a role: in the model, the

quick failure of the cementing thin ice is represented by relaxing the "fast-growth" FSD $g_{\text{fast}}$ to the "slow-growth" FSD $g_{\text{slow}}$. In practice, we set $\Phi_{\text{m,thermo}}\Delta t_{\text{ice}} = g_{\text{fast}} - g_{\text{slow}}$, where $\Delta t_{\text{ice}}$ is the ice model time step, therefore assuming that this relaxation is almost instantaneous. We justify this short relaxation time by the fact that (i) waves can fragment a consolidated sea ice cover in a few tens of minutes only (Collins et al., 2015) and (ii) the "fast-growth" FSD $g_{\text{fast}}$ is only used for thermodynamical processes associated with timescales of at least a few hours, and is therefore relatively unaffected by the choice of a relaxation

time value one order of magnitude lower.

Once fragmentation has occurred, the relaxation of the "fast-growth" FSD to the "slow-growth" FSD gives $g_{\text{fast}} = g_{\text{slow}}$. This equality represents the fact that $D_{\text{fast}} = D_{\text{slow}}$ when fragmentation occurs and before thin ice starts cementing the floes. In the model, the shape of the two FSDs after a fragmentation event is then controlled by the mechanical redistribution occurring in

the "slow-growth" FSD represented by the term $\Phi_{\text{m,slow}}$. This term can be written in the same form as in Zhang et al. (2015):

$$\Phi_{\text{m,slow}} = -Q(D)g_{\text{slow}}(D) + \int_{0+}^{\infty} Q(D')\beta(D',D)g_{\text{slow}}(D')dD' \tag{6}$$

where $Q(D)$ is a redistribution probability function characterising which proportion of floes of a given size $D$ is going to be broken (with $D$ representing indistinctively $D_{\text{fast}}$ and $D_{\text{slow}}$ during fragmentation), and $\beta(D',D)$ is a redistribution factor quantifying the fraction of sea ice concentration transferred from floe size $D'$ to $D$ as fragmentation occurs. The choices of

$Q(D)$ and $\beta(D',D)$ therefore shape the FSD resulting from wave-induced fragmentation. This shape is important as it will strongly impact processes involved in wave attenuation (Boutin et al., 2018) and lateral melting (Bateson et al., 2020), but the evolution of the FSD during wave-induced fragmentation is still not well understood.

Toyota et al. (2011) have suggested from field observations that the shape of the FSDs could be interpreted as two truncated power-laws separated by a cut-off floe size, and hypothesise this cut-off corresponds to the critical floe size under which floes

cannot be broken by waves. A cut-off floe size also seems to be visible in the FSDs that Herman et al. (2018) obtain from a laboratory experiment. The existence of a cut-off floe size is however contested: the use of cumulative distribution functions to interpret FSDs may give a false impression of scale-invariance, and the apparent change of regime could originate from finite measurement windows (Burroughs and Tebbens, 2001; Stern et al., 2018). The division of the FSD into two truncated power-laws suggested by Toyota et al. (2011) has nevertheless been used to redistribute FSDs in the wave-in-ice models by

Dumont et al. (2011) and Williams et al. (2013b), which have been reused for the wave-in-ice attenuation parameterization implemented in WW3 by Boutin et al. (2018), and in many other studies using wave-in-ice models (see, e.g., Aksenov et al., 2017; Williams et al., 2017; Bennetts et al., 2017; Bateson et al., 2020; Boutin et al., 2020).

In this study, the FSD is mostly used to provide WW3 with information on the floe size to estimate the wave attenuation. The FSD does not impact the amount of new ice formed, and we focus on periods during which lateral melting can be neglected. In these conditions, it is advantageous to stay close to what has been done already for the FSD in WW3, in order to ensure that wave attenuation will not be too different from the one evaluated by Boutin et al. (2018) and Ardhuin et al. (2018). We therefore build on the work by Zhang et al. (2015) and Boutin et al. (2020) to suggest a new parameterization for $Q(D)$ and $\beta(D', D)$ that redistributes the FSD in a way similar to the assumptions made in wave-in-ice models derived from Williams et al. (2013b). However, there are two main differences from the FSDs assumed in Williams et al. (2013b): we allow the exponent of the power-law corresponding to the "small-floe" regime to vary, and we smooth the transition between the "small-floe" regime and the "large-floe" regime, as it can be seen in the data by Toyota et al. (2011).

$Q(D)$ represents the amount of ice in each floe size categories that is going to be broken. Williams et al. (2013b) and a number of further studies using their approach (Bennetts et al., 2017; Bateson et al., 2020; Boutin et al., 2020) use a step function with $Q(D) = 1$ if $D \geq 0.5\lambda_{\mathrm{break}}$ and $D \geq D_{\mathrm{FS}}$, $D_{\mathrm{FS}}$ being the minimum floe size for which flexural failure can occur (see Mellor, 1986), and $Q(D) = 0$ if not. The transition of $Q(D)$ from 0 to 1 occurs at a critical floe size under which floes do not break, corresponding to what Toyota et al. (2011) interpreted as the cut-off floe size. In the model of Williams et al. (2013b), it therefore occurs at $\max(D_{\mathrm{FS}}, 0.5\lambda_{\mathrm{break}})$. $D_{\mathrm{FS}}$ depends on sea ice properties and is computed as:

$$D_{\mathrm{FS}} = \frac{1}{2}\left(\frac{\pi^4 Y h^3}{48\rho g(1-\nu^2)}\right)^{1/4} \tag{7}$$

where $g$ is gravity, $Y$ the Young modulus of sea ice, $\nu$ Poisson's ratio, $h$ mean sea ice thickness, and $\rho$ is the density of sea water. For ice thinner than 1 m, which is often the case in the MIZ, $D_{\mathrm{FS}}$ is lower than 15 m, and the cut-off floe size is likely to be determined by the value of $\lambda_{\mathrm{break}}$. To define $Q(D)$, we take an approach similar to Williams et al. (2013b) and set:

$$Q(D) = \frac{1}{\tau_{WF}} p_{\mathrm{FS}}(D, D_{\mathrm{FS}}) p_\lambda(D, \lambda_{\mathrm{break}}), \tag{8}$$

in which $\tau_{\mathrm{WF}}$ is a relaxation time associated with wave-induced fragmentation events, and $p_{\mathrm{FS}}$ and $p_\lambda$ are probabilities that floes break depending on their size. We introduced $\tau_{\mathrm{WF}}$ to avoid dependency of the FSD redistribution to the coupling time step. It represents the timescale needed for the FSD of a fragmenting sea ice cover to reach a new equilibrium under a constant sea state. We set it to 30 minutes, as it corresponds to the timescale of the fragmentation event described in Collins et al. (2015). The probability functions $p_{\mathrm{FS}}$ and $p_\lambda$ express the idea that the smaller the floes are, the less chance they have to break. The function $p_\lambda$ compares the floe size $D$ with the value of $D_{\mathrm{FS}}$, and $p_\lambda$ compares the floe size $D$ with $\lambda_{\mathrm{break}}$, introducing a dependency of $Q(D)$ on the wave field. The difference with the model Williams et al. (2013b) is that instead of step functions, we use hyperbolic tangents to get a continuous transition of $Q(D)$ between 0 and 1:

$$p_{\mathrm{FS}}(D) = \max\left(0, \tanh\left(\frac{D - c_{1,\mathrm{FS}}D_{\mathrm{FS}}}{c_{2,\mathrm{FS}}D_{\mathrm{FS}}}\right)\right), \tag{9a}$$

$$p_\lambda(D, \lambda_{\mathrm{break}}) = \max\left(0, \tanh\left(\frac{D - c_{1,\lambda}\lambda_{\mathrm{break}}}{c_{2,\lambda}\lambda_{\mathrm{break}}}\right)\right), \tag{9b}$$

in which $c_{1,\mathrm{FS}}$, $c_{2,\mathrm{FS}}$, $c_{1,\lambda}$, and $c_{2,\lambda}$ are parameters of the FSD that control the range of floe size that will be broken or not. The use of a continuous $Q(D)$ instead of a step function aims to relax the constraint on the FSD shape imposed by Williams et al.

(2013b). With a step function, the probability of having floes larger than the cut-off floe size is 0, *i.e* above the cut-off floe size, the FSD is suddenly infinitely steep. This approach is particularly problematic, as firstly the FSDs reported in Toyota et al. (2011) rather show a gradual steepening than a sharp transition, and secondly the steepening of the FSD slope that led to the identification of the two floe size regimes by Toyota et al. (2011) could actually be due to windowing issues (Stern et al., 2018).

Here, instead of having a single cut-off floe size, we have a transition occurring in the floe size range for which $0 < Q(D) < 1$. The width of this range is controlled by $c_{2,\mathrm{FS}}$ and $c_{2,\lambda}$. $Q(D)$ tends towards a step function when $c_{2,\mathrm{FS}}$ and $c_{2,\lambda}$ tend towards 0. We found that setting $c_{2,\mathrm{FS}} = 2$ and $c_{2,\lambda} = 2$ leads to FSDs with a gradual steepening that look very similar to what is reported by Toyota et al. (2011) or in the lab experiments by Herman et al. (2018) for instance. The location of this floe size range for which the transition occur is controlled by $c_{1,\mathrm{FS}}$ and $c_{1,\lambda}$. Like Williams et al. (2013b), we set $c_{1,\mathrm{FS}} = 1$. Instead of using

$c_{1,\lambda} = 0.5$ as in Williams et al. (2013b), we set $c_{1,\lambda} = 0.3$. It is coherent with the hypothesis made by Boutin et al. (2018) that floes smaller than $0.3\lambda_{\mathrm{break}}$ are tilted by waves but do not bend, and have therefore no chance of suffering flexural failure. Reducing the value of $c_{1,\lambda}$ reduces the value of the cut-off floe size we should get, therefore leading to floes in general smaller than assumed in Williams et al. (2013b). It is however compensated by using a continuous $Q(D)$, which gives more weight to large floes than the FSD assumed in the model of Williams et al. (2013b). FSDs generated by this redistribution function are

presented and discussed in section 4.1.2.

   The redistribution factor $\beta$ controls the shape of the FSD after it has been redistributed. Similar to the work by Zhang et al. (2015) and Boutin et al. (2020), the redistribution factor $\beta$ follows the form:

$$
\begin{cases}
\beta(D_n, D) = \dfrac{D_n^q - D_{n-1}^q}{D^q - D_0^q} \text{ if } D_n \geq D \\
\beta(D_n, D) = 0 \text{ otherwise}
\end{cases}
\tag{10}
$$

where $n$ corresponds to the index of the $n^{\mathrm{th}}$ FSD category, and $q$ is an exponent that controls the shape of the redistributed FSD. Zhang et al. (2015) use $q = 1$, arguing that the fragmentation of floes being a stochastic process, any floe size lower than the initial unbroken floe can be generated with a similar probability. It leads to power-law FSDs after a succession of fragmentation events. Boutin et al. (2020) note that $q$ is related to the (negative) exponent $\alpha$ of the power-law FSD associated with the smaller floe sizes resulting from the redistribution as assumed by Toyota et al. (2011) and later by Williams et al.

(2013b), with $q = 2 + \alpha$. In their study, Toyota et al. (2011) relate the exponent $\alpha$ to a quantity they call fragility, which represents the probability of fragmentation of floes. Williams et al. (2013b) assume that fragility is constant and equal to 0.9, giving $\alpha \simeq -1.85$. Boutin et al. (2020) set the value $q$ to prescribe power-law FSDs with this same value of $\alpha$ and stay fully coherent with what was done in wave models before. This is however a big constraint on the shape of the FSD, and it contradicts the variations of the exponent of the power-law fitted to the small-floe regime reported by Toyota et al. (2011).

Here, we have already expressed the probability of fragmentation of floes, hence the fragility, as the product of $p_{\mathrm{FS}}$ and $p_\lambda$. Using this relationship, we get:

$$
q = -\log_2(p_{\mathrm{FS}}(D)p_\lambda(D, \lambda_{\mathrm{break}})).
\tag{11}
$$

This definition of $q$ still generates FSD that tend to a power-law for floe sizes lower than the cut-off floe size, but does not prescribe a fixed value to the exponent $\alpha$ of this power-law. Instead, $\alpha$ decreases as the ice thickens and wave wavelength increases, as it is shown and discussed in section 4.1.2.

The processes we use for the wave-in-ice attenuation computation in WW3 require the estimation of two floe size parameters: the average floe size $\langle D \rangle$ and the maximum floe size $D_{\mathrm{max}}$ (Boutin et al., 2018). When WW3 is run in stand-alone mode, $D_{\mathrm{max}}$ is taken to be $\lambda_{\mathrm{break}}/2$ and an assumption made on the shape of the FSD after fragmentation allows to estimate the value of $\langle D \rangle$. When WW3 is coupled with neXtSIM, the FSD is free to evolve to the sea ice model, and it is necessary to estimate the value of $D_{\mathrm{max}}$ and $\langle D \rangle$ in neXtSIM so it can be sent to WW3.

The average floe size $\langle D \rangle$ can be simply defined as:

$$\langle D \rangle = \int_0^\infty D \, g(D) dD. \tag{12}$$

Here we use the "slow-growth" FSD $g_{\mathrm{slow}}(D_{\mathrm{slow}})$ assuming that wave scattering, which is the wave attenuation process depending on $\langle D \rangle$, is more affected by consolidated floes than by the thin ice jointing them. The maximum floe size $D_{\mathrm{max}}$ definition is less straightforward, as it was originally designed in the FSD parameterization of Dumont et al. (2011) to rep-

resent the largest floe size of a fragmented sea ice cover, and if no fragmentation had occurred it was set to a large default value. Here, this definition needs to be extended to a coupled-system with an FSD free to evolve under the effects of both mechanical and thermodynamical processes, able to represent a mix of fragmented floes and large ice plates. We suggest a definition based on the percentage of the ice cover area occupied by large floes, computing $D_{\mathrm{max}}$ as the 90[th] percentile of the areal FSD. Besides, the flexure dissipation mechanisms included in WW3 by Boutin et al. (2018) requires to discriminate

between a sea ice cover made of large floes with size of the order of $O(100)$m and an unbroken sea ice cover for which the default $D_{\mathrm{max}}$ in WW3 is set to 1000m. This is because flexure only occurs if the wavelength is shorter or of the same order as the floe size. Knowing that long swells can have wavelengths of the order of $O(100)$m, they will only be fully attenuated by inelastic dissipation if floe size is of the order of $O(1000)$m, which can be larger than the floe size range covered by the FSD defined in neXtSIM. In the case where $D_N < 1000$m, to make sure that swells are still attenuated in an unbroken sea ice cover

by WW3, we linearly increase the value of $D_{\mathrm{max}}$ sent to WW3 from $D_{\mathrm{max}} = D_N$ to $D_{\mathrm{max}} = 1000$m with the proportion of sea ice in the largest floe size category $\int_{D_{N-1}}^{D_N} g_{\mathrm{slow}}(D) dD/c$.

## 2.3   Link between wave-induced sea ice fragmentation and damage

As mentioned in the Introduction, it is expected that sea ice fragmentation by waves results in lowering the ice internal stress. The lowering of sea ice resistance to deformation due to the high density of cracks is already included in neXtSIM, with

the variable called damage. This variable takes value between 0 and 1, with 0 corresponding to undamaged sea ice, and 1 to a highly damaged sea ice cover, i.e. presenting a high-density of cracks. Sea ice damaging in neXtSIM is usually due to the wind. In our study, we would like it to have an additional dependence on wave-induced sea ice fragmentation. In our

implementation, it is possible to quantify the sea ice cover area that is susceptible to be broken by waves if WW3 provides a value of $\lambda_{\mathrm{break}} < 1000$ m as $c_{\mathrm{broken}} = c\left(1 - \mathrm{e}^{-\Delta t_{\mathrm{cpl}}/\tau_{WF}}\right)$, and $c_{\mathrm{broken}} = 0$ if $\lambda_{\mathrm{break}} \geq 1000$ m. As the fragmentation of sea ice by waves can break ice plates into floes with sizes up to a few hundred metres, and as the horizontal resolution of the model mesh is in general at least a few kilometres, we make the hypothesis that areas of the sea ice cover fragmented by waves are associated with high values of damage, i.e. close to 1. We thus suggest to compute the new damage value associating a value of $d_w =$0.99 to $c_{\mathrm{broken}}$, which gives the following evolution of the damage $d$:

$$d = \min\left(1, d(1 - c_{\mathrm{broken}}) + d_w c_{\mathrm{broken}}\right). \tag{13}$$

This process is repeated every time fragmentation occurs in the sea ice model. Note that, because wave events generally last for a few hours, this damaging process is generally repeated enough times to result in little sensitivity of the model to values of $d_w$ between 0.1 and 1. Note also that floe size and damage are not explicitly linked by this relationship, but the relaxation time associated with the healing of damage and of the "slow-growth" FSD are the same, making their evolution parallel in the regions of broken ice.

## 3 Model set-up

### 3.1 General description of the pan-Arctic Configuration

Similarly to Boutin et al. (2020), the coupled framework is run on a regional $0.25°$ grid (CREG025), which covers the Arctic Ocean at an approximate resolution of $12\,$km, as well as some of the North Atlantic. As neXtSIM is a finite element sea ice model using a moving Lagrangian mesh, it is not run on a grid. Its initial mesh is, however, based on a triangulation of the CREG025 grid, giving a prescribed mean resolution (i.e. mean length of the edges of the triangular elements) of $12\,$km. In coupled mode, neXtSIM interpolates the fields to be exchanged onto the fixed grid, so that the coupler, OASIS, is only required to send and receive different fields (i.e. OASIS does not need to do any interpolation).

neXtSIM is run with a timestep of $20\,$s, and WW3 with a timestep of $800\,$s. Fields between the two models are exchanged every $2400\,$s. Atmospheric forcings are provided by 6-hourly fields from the CFSv2 atmospheric reanalysis (Saha et al., 2014). In addition, neXtSIM is also forced by ocean fields from the TOPAZ4 reanalysis (Sakov et al., 2012). For more details on the forcings used by neXtSIM, see Rampal et al. (2016). Wave-current interactions in WW3 are not considered in this study.

For the FSD in neXtSIM, we use 20 categories with a width $\Delta D =$10 m. The lower bound $D_0$ is set to 10 m, which is about the size of the smallest floes susceptible to undergo flexural failure (Mellor, 1986). The upper bound of the FSD, $D_N$, is therefore equal to 210 m, which is of the order of magnitude of the largest floes resulting from wave-induced sea ice fragmentation generated by WW3. The healing relaxation time $\tau_{\mathrm{heal}}$ used by neXtSIM is set to its default value of 25 days (Rampal et al., 2016).

## 3.2 Evaluation of the FSD implementation: model set-up

In section 4.1, we make use of sea state and sea ice observations realised in the Beaufort Sea in the framework of the Arctic Sea State and Boundary Layer Physics Program (Thomson et al., 2018) to evaluate the wave attenuation and broken sea ice extent in our coupled simulations, similar to what Ardhuin et al. (2018) did with stand-alone WW3 simulations. To do so, we run 5 simulations from October 10[th], 2015 to October 13[th], 2015, a period covering the storm event investigated in a study by Ardhuin et al. (2018). This storm generates $\simeq$4-m-waves fragmenting the sea ice edge in the Beaufort Sea from 11-12 October 2015.

The first of these simulations is a WW3 uncoupled simulation hereafter labelled ARD18 as it uses the exact same parameterization as the one labelled REF2 in Ardhuin et al. (2018). The only difference is that here it is run on the CREG025 grid used for all our simulations. This parameterization of the wave model is chosen as it is the one showing the best match with observations for both wave height and broken sea ice extent in the study by Ardhuin et al. (2018). We also use the same sea ice concentration data as Ardhuin et al. (2018) to force the uncoupled wave model. They are obtained from a reanalysis of the 3-km resolution sea ice concentration dataset derived from the AMRS2 radiometer using the ASI algorithm (Kaleschke et al., 2001; Spreen et al., 2008)[1] (last access: November 2020). This reanalysis produces 12-hourly maps that gather all the AMSR2 passes acquired between 00:00 and 13:59 UTC, and 10:00 and 23:59 UTC for the morning (AM) and evening (PM) fields, respectively. Similarly to Ardhuin et al. (2018), sea ice thickness is set constant to 15 cm. Sea ice concentration is also kept constant to make the comparison with a coupled simulation easier. Sea ice concentration for this 3-day run corresponds to the conditions of the evening of October 12[th] provided by the AMSR-2 sea ice concentration reanalysis, at the same time as illustrated in Ardhuin et al. (2018) study. Initial wave conditions are provided by an initial 10-days run of this simulation, from October 1[st] to October 10[th], 2015, in which sea ice concentration is updated every 12h.

Secondly, we run a coupled neXtSIM-WW3 simulation (hereafter labelled NXM/WW3). Initial conditions are the same as in ARD18. Sea ice dynamics and thermodynamics are switched off in neXtSIM, so that we can compare the two simulations with a similar constant sea ice cover (thickness and concentration), the only difference between ARD18 and NXM/WW3 being the way the evolution of floe size is treated, which is what we want to evaluate.

Then, to illustrate the sensitivity of our results to the sea ice thickness value, we re-run ARD18 and NXM/WW3 while setting this time the sea ice thickness to 30 cm. We name these two additional simulations ARD18_H30 and NXM/WW3_H30 respectively.

Finally, to investigate the sensitivity of the FSD evolution to the floe size categories used for the FSD, we run a a simulation similar to NXM/WW3 but with a refined FSD that we call NXM/WW3_refine. For this simulation, the number $N$ of

---

[1] AMSR2 data are available at https://seaice.uni-bremen.de/data/amsr2/asi_daygrid_swath/n3125/2015/oct/Arctic3125/

categories is set to 41 instead of 20, and we set $\Delta D$=5 m and $D_0$ =5 m. We also evaluate the evolution of the two FSDs with refreezing/healing using the CPL_DMG simulation described in the following section 3.3.

## 3.3 Estimation of the impact of wave-induced fragmentation on sea ice dynamics: model set-up

In section 4.2, we compare the results of 3 simulations in order to investigate the wave impact on sea ice dynamics in the
MIZ. The first one (called REF) is a stand-alone simulation of neXtSIM. The second one (CPL_WRS) includes all the features presented in section 2 but the relationship between wave-induced sea ice fragmentation and damage presented in 2.3. The third (CPL_DMG) is similar to CPL_WRS except that it also includes a link between the damage variable $d$ and wave-induced sea ice fragmentation as described in section 2.3. These simulations are run for a period going from September 15[th] to November 1[st] 2015. This period was selected as refreezing occurs in the MIZ, meaning that the differences between REF and the two
coupled simulations are not due to the change in lateral melting parameterization. This period of the year is also characterised by the combination of a low sea ice extent (thus a large available fetch) and regular occurrence of storms in the Arctic, which increases the opportunities to evaluate the impact of waves on sea ice with fragmentation events over wide areas. The level of damage in the ice cover is initially set to zero where sea ice is present. Initial sea ice concentration and thickness are set from the TOPAZ4 reanalysis (Sakov et al., 2012), and sea ice is unbroken. The wave field in WW3 is initially at rest. The wave-in-ice
attenuation parameterization in WW3 in CPL_WRS and CPL_DMG is the same as in ARD18 (i.e. REF2 in Ardhuin et al., 2018). We investigate the results of these simulations from October 1[st], thus allowing for 16 days of spin-up, which is enough for the wave and damage fields to develop.

## 4 Results

### 4.1 Evaluation of wave–sea-ice interactions in the coupled framework

We first evaluate the representation of wave-ice interactions in our coupled framework. As our goal is to investigate the potential impact of waves on sea ice dynamics, we must ensure that the coupled framework produces a consistent wave-in-ice attenuation, as it is directly proportional to the WRS, as well as reasonable extents of broken sea ice and timescales of the ice recovery from fragmentation. In the following, we will consider the wave attenuation and extent of fragmented sea ice on the one hand, and the evolution of the FSDs after fragmentation events on the other hand.

#### 4.1.1 Evaluation of wave attenuation and extent of fragmented sea ice

To evaluate the capacity of our coupled framework to produce reasonable wave-in-ice attenuation and extent of broken sea ice, we focus on the same event used to evaluate the WW3 parameterization of Boutin et al. (2018) in the study by Ardhuin et al. (2018). Here, we use the ARD18 simulation as a reference, as it was shown to give a good match with observations for both the extent of broken ice and the wave attenuation in this particular case. The comparison is done on October 12th, at 17:00
GMT (Fig. 3). We also compare our model results with estimated wave height from SAR images (Stopa et al., 2018a) and buoy

measurements (AWAC, see Thomson et al., 2018) along a transect in Figure 3d.

We evaluate the wave attenuation by looking at the evolution of the maximum floe size $D_{\max}$ and significant wave height in the ice. The spatial distribution of these two quantities is overall very similar between ARD18 and NXM/WW3, and also very similar to the results of Ardhuin et al. (2018) (see Figure 9 of their study). Figure 3d shows the wave height evolution along a transect following the footprint of Sentinel 1-a, and again we see almost no difference between ARD18 and NXM/WW3. Both simulations show reasonable agreement with the wave heights estimated from SAR and from the AWAC buoy. Similar to the results of Ardhuin et al. (2018), the model, however, seems to slightly overestimate the wave height within the ice cover. This overestimation could result from the assumption of constant thickness and its low value (15 cm). This is visible on Figure 3d where most observations actually show higher significant wave height values than the one yielded by ARD18_H30 and NXM/WW3_H30, that use a constant thickness of 30 cm.

Sea ice break-up occurrence depending on wave properties, comparable wave attenuation in between ARD18 and NXM/WW3 result in little difference in the extent of broken sea ice between the two simulations (Fig. 3a,c). Although the extent of broken ice is slightly smaller in the coupled run, the difference does not exceed 2 grid cells, therefore representing a distance of about 25 km, which is acceptable given the large uncertainties associated with wave attenuation in ice (see for instance Nose et al., 2020). Moreover, as in Ardhuin et al. (2018), the broken sea ice region extends up to about 15 km in-ice beyond the AWAC buoy (red square), which matches well with the observation as on Synthetic Aperture Radar (SAR) images for that same day.

### 4.1.2   Evaluation of the evolution of the FSDs

Our coupled framework introduces two FSDs to represent the evolution of the floe size from two different points of view. It also introduces a new redistribution scheme used when wave-induced sea ice fragmentation occurs. This section provides a quick evaluation of these new features.

We first look at the FSD resulting from wave-induced fragmentation in neXtSIM by plotting the cumulative distribution of floes (CDF, see e.g. Toyota et al., 2011; Herman et al., 2018) for 3 different locations (Fig. 4) for the NXM/WW3 and NXM/WW3_refine simulations. Note that because thermodynamical and dynamical processes are unactivated in NXM/WW3, the "fast-growth" and "slow-growth" FSDs are identical. The CDFs look very similar to the one reported by (e.g.) Toyota et al. (2011) with the curve gradually steepening as floe size increases. Following the method of Toyota et al. (2011), two lines can be fitted to the FSD for small and large floes, defining two regimes intersecting at a cut-off floe size: (i) a small floe regime, that follows a power-law, and (ii) a large floe regime, with a much steeper slope of the CDF. This interpretation may result from an artefact arising from the use of the CDFs and finite-size windows (Stern et al., 2018) and the use of CDFs in particular can give a false impression of scale-invariance, but has however been used in a number of wave-ice interactions studies, in particular the one used to calibrate the wave attenuation model we use here and in all the models using the work of Williams et al. (2013b). Here we use the CDFs to discuss the differences introduced by our parameterization of the FSD redistribution compared to the assumptions made by Williams et al. (2013b) and discuss how they could impact wave attenuation.

As in the study by Toyota et al. (2011), the cut-off floe size and the (negative) power-law exponent of the small floe regime increase with the distance from the ice edge. For the two simulations, all but one value of the (negative) power-law exponents related to the small floe-regime are greater than -2, as expected for a CDF depending on a 2-D fragmentation process (Toyota et al., 2011). The one value of the exponent that does not lie in this range is obtained for the NXM/WW3 run close to the ice edge, where the cut-off floe size is too close to the value of $D_0$ for the small floe size regime to be resolved. In the model of Williams et al. (2013b), the exponent of the power-law is equal to $\simeq -1.85$. Our FSDs and the one assumed by models derived from Williams et al. (2013b) are therefore likely to be very similar close the ice edge, but further away in the ice our parameterization gives less weight to small floes in the FSDs. For wave attenuation, it means that there will be less scattering occurring as waves propagate towards pack ice. Little impact on wave attenuation is expected, as scattering is mostly efficient for short waves, and short waves do no propagate far into the ice. $D_{\max}$ in the model of Williams et al. (2013b) corresponds to the "cut-off" floe size as interpreted by Toyota et al. (2011). Here, the values of $D_{\max}$ for each location lie in the floe size range corresponding to the transition between the two regimes, agreeing with the definition used by Williams et al. (2013b). $D_{\max}$ shows little sensitivity to the FSD definition, with a maximum difference between the two simulations presented on Fig. 4 not exceeding the value of $\Delta D$ in the refined-FSD simulation (5 m). Considering the large uncertainties due to the little knowledge of wave-ice interactions, choosing $\Delta D = 10$ m instead of more refined FSDs in our coupled framework has therefore little impact on the wave attenuation computed in WW3.

As the simulations in this study focus on fall-period, when the sea ice cover expands due to freezing, we also check that the timescales associated with freezing and sea ice healing are reasonable. Figures 5 and 6 illustrate the evolution of the FSDs in the CPL_DMG simulation, in which both mechanical and thermodynamical processes are active. Figures 5(a,c) show the proportion of "unbroken" ice (the proportion of sea ice associated with the $N^{\mathrm{th}}$ category of the FSD) for the "fast-growth" and "slow-growth" FSDs respectively, 17 days after the beginning of the simulation, leaving enough time for the waves and sea ice to spin-up. The regions of broken ice are relatively similar for both FSDs with the exception of the Barents Sea area. They actually closely follow the contour of 1-metre thick ice (not shown), after which the waves are too attenuated to fragment the sea ice. Floe size generally increases with distance from the ice edge, as the ice gets thicker and shorter waves are quickly attenuated (Fig. 5e,f). The Barents Sea area shows a wide area of broken thin ice in the "slow-growth" FSD, with only parts of this region being broken in the "fast-growth" FSD. This wide broken area is related to a strong wave event in this region occurring between 30 September 2015 and 01 October 2015. This event is associated with wave height up to 9 m and waves with period above 12 s propagating far into an ice cover made of relatively thin ice (less than 1m). About 24 hours after the event (not shown), the "fast-growth" FSD in pack ice has mostly "recovered" due to welding and freezing in leads. In pack ice, where floes are larger than at the ice edge, the speed of the floe size growth in the "fast-growth" FSD is mostly controlled by welding, and therefore depends on the value chosen for rate of decreases of the number of floes $\kappa$. This is because, like Roach et al. (2018), we use a constant value for $\kappa$, meaning that the fewer floes there are (*i.e. the larger the floe size*), the higher the proportion of floes that merge during a given time period. The growth of large floes in the "slow-growth" FSD takes much longer, with a timescale set by the value of $\tau_{\mathrm{heal}}$ (25 days in CPL_DMG), and the end of the mechanical healing of the Barents

Sea area is still visible on 01 November 2015 (Fig. 5d).

This difference in timescales is also visible in Figure 6, which illustrates the evolution of the "fast-growth" and "slow-growth" FSDs (a,c) and CDFs (b,d) for the location indicated by a cross on Figure 5(a,c), at the time shown on the snapshot (02 October 2015 at 00:00:00 GMT) and 24 hours later. On 02 October 2015 the proportion of sea ice that is unbroken is almost 0 in the "slow-growth" FSD, while welding and refreezing in the "fast-growth" FSD have allowed the re-formation of unbroken sea ice over more than 10% of the ice-covered part of the mesh element. The action of welding and refreezing in the "fast-growth" FSD results in a steepening of the slope of the CDF associated with the small floe regime, and flattening of the slope of the large floe regime. The "slow-growth" FSD shows no sign of any healing, the last fragmentation event being too recent, and its associated CDF clearly shows a small and a large floe regime resulting from the fragmentation by waves (Fig. 5). 24 hours later, more than 95% of the "fast-growth" FSD consists of unbroken sea ice, while the "slow-growth" FSD is still very similar to what it was on the previous day, illustrating the memory effect of the "slow-growth" FSD.

The evolution of the "slow-growth" FSD involves the use of an ad hoc term to ensure the conservation of sea ice area as well as a relaxation towards the fast FSD (see Eq. 5). These terms might affect the moments of the distribution in a non-trivial way, as shown in section 3.1 of Horvat and Tziperman (2017). This might affect the estimation $D_{\mathrm{max}}$ and $\langle D \rangle$ that are both computed from moments of the "slow-growth" FSD. In the absence of observations that could help to constrain our model, we represent in Fig. 7 the evolution of $D_{\mathrm{max}}$ and $\langle D \rangle$ for the location indicated on Fig. 5(a,c,e). This evolution is in line with what we could expect from the floe size evolution, with sudden drops due to fragmentation events and a slower growth over a timescale of about 20 days.

In summary, once the wave activity has decreased, refreezing and welding allow for the re-generation of a completely unbroken sea ice cover in timescales of a few hours to a few days in the "fast-growth" FSD, depending on the initial level of fragmentation. The timescale over which the "slow-growth" FSD re-generates large floes is associated with the value of $\tau_{\mathrm{heal}}$. As an indication of time, in the case illustrated here, with freezing conditions and compact ice, it takes about 4 days for the value of $D_{\mathrm{max}}$ to grow over $200 \, \mathrm{m}$.

## 4.2 Impact of sea ice fragmentation on sea ice dynamics in the MIZ

### 4.2.1 Case study: a fragmentation event in the Barents Sea (15-25 Oct. 2015)

To better understand the impact of (i) the WRS and (ii) fragmentation-induced damage on sea ice dynamics, we compare the results given by the REF, CPL_WRS, and CPL_DMG simulations in the Barents sea. Focusing only on this region simplifies the analysis, as this area is exposed to wave and sea ice conditions that experience little variations over the investigated period. The available fetch in particular remains relatively constant, and is large enough to allow for storm waves to penetrate far into the ice. The sea ice edge also remains oriented mostly east-west over this period. In our analysis, we can therefore consider

southward winds to be mostly off-ice, and conversely northward winds to be directed on-ice. The domain we define to perform our analysis is limited south and north by the 69°N and 84°N parallels respectively, and west and east by the 16°E and 60°E meridians (see for instance Fig. 8).

To highlight the various responses of sea ice to fragmentation depending on wind and waves conditions, we select a particular fragmentation event occurring on October 15$^{th}$ 2015 (see Fig. 8 for the initial sea ice conditions) which results in a growth of the surface occupied by broken ice (Fig. 9c). The 10 days following this event include both on-ice and off-ice conditions, allowing us to explore the impact of wave-induced sea ice fragmentation on sea ice dynamics in both cases.

The fragmentation event occurs during on-ice wind conditions (see the positive meridional component in Fig. 9a), with high waves (up to 5m at the ice edge) propagating far into the ice cover (see Fig. 9b and Fig. 10b). It results in an increase of the surface area made of recently broken sea ice in the domain (see for instance the evolution of the magenta contour between Fig. 8a and Fig. 10a), until it represents nearly 60% of the sea ice-covered part of the domain (Fig. 9c). Here we define the sea ice covered part of the domain as the area for which sea ice concentration $c > 0$. Recently-broken ice is defined as the region for which $D_{max} \leq 200$ m, thus corresponding to fragmented sea ice for which refreezing has not yet had time to heal a significant proportion of unbroken ice (at least 10%). In the domain, the recently-broken sea ice area is mostly made of compact sea ice (that we define as the area of the domain for which $c > 0.8$). At the end of the fragmentation event, 80% of the recently-broken sea ice is made of compact ice, and it represents nearly 40% of the ice-covered part of domain (Fig. 9c). Compact sea ice being broken is important in the scope of our study, as low-concentration sea ice experiences little resistance to deformation due to its low effective stiffness, and is therefore largely unaffected by damage.

      In the days following the fragmentation event, wind speed decreases and changes direction to become mostly southward (Fig. 9a). The wind speed then increases again, with a second maximum on October 20$^{th}$, corresponding this time to off-ice winds. The generated waves tend to propagate away from the sea ice, resulting in low wave heights inside the ice (Fig. 9b). It
coincides with a decrease in recently-broken sea ice surface area, which stops when the wind turns again to become parallel to the sea ice edge (Fig. 9c). This decrease is mostly due to sea ice recovering in the absence of waves, and can be seen by comparing the distributions of sea ice thickness and damage on October 16$^{th}$ (Fig. 10a and 11a) and October 21$^{st}$ (Fig. 10c and 11c), in which the limit between broken and unbroken ice tends to get closer to the ice edge, while damage in pack ice has visibly decreased. The band of recently broken ice remains however much larger than it was initially (Fig. 8b), as fragmented
sea ice produces lower wave attenuation, thus allowing for sea ice fragmentation even in low wave height conditions.

### 4.2.2  Effects of linking damage and sea ice fragmentation on sea ice dynamics in the MIZ

The conditions of the studied period now being described, the impact of adding the WRS and a relationship between wave-induced fragmentation and damage can be investigated. We first proceed by comparing (in Figure 12) the ice drift velocity

averaged over the ice-covered domain for the 3 simulations: REF, CPL_WRS and CPL_DMG. Overall, we observe no differences in the trends, however, the magnitudes of the ice drift velocities show intermittent differences between these 3 simulations. These differences have two maxima, the first one on October 16 at 18:00:00 GMT, and the second one on October 21 at 09:00:00 GMT. To understand these differences, we compare the CPL_DMG and REF simulations at these two dates.

On October 16[th], wind and waves are directed on-ice, thus compacting the sea ice (Fig. 10b). At the time of the snapshot, sea ice has been recently broken over a wide surface area (within the magenta contour in Figures 10(a,b),11(a,b) and 13(a,b). It results in the damage value being maximum everywhere in this recently broken sea ice area in the CPL_DMG simulation (Fig. 11a). Comparing damage between the CPL_DMG and REF simulations (Fig. 11b), we note that the increase in damage related to wave-induced fragmentation is strong at the immediate proximity of the ice edge, and lower but still sensible ($\simeq 0.05$) closer to the limit between broken and unbroken ice. Comparing Figure 11b with Figure 13b, it is interesting to note that the highest difference in the magnitude of ice drift velocities between CPL_DMG and REF is mostly located in this area where the increase in damage due to waves is limited (Fig. 13b). This result is at first counter-intuitive, the biggest impact on the sea ice drift occurring where the impact on damage is the weakest, but it is due to the nature of sea ice at the limit between broken and unbroken ice, which is thicker and more compact than at the proximity of the ice edge. As mentioned before, thin, loose sea ice does not provide much resistance to deformation. For such sea ice, the level of damage has therefore little impact on its behaviour. Conversely, thick compact ice is usually associated with high ice strength values, and its level of damage significantly impacts its resistance to deformation. The additional damage of thick compact ice due to wave-induced fragmentation, despite being small, allows for more sea ice convergence than in the REF and CPL_DMG simulations. Similarly, on October 21[st], the difference in ice drift velocity between CPL_DMG and REF is mostly due to an acceleration of compact sea ice that has been recently broken (Figure 13d). This acceleration follows the wind direction, creating additional convergence north of Svalbard, and divergence at the centre of the domain (Figure 13c), thus increasing the export of sea ice this time.

To better quantify the impact of this additional damage on the dynamics of compact sea ice, we compare the ice drift velocity between the CPL_DMG and CPL_WRS simulations averaged over the area covered by compact and recently broken sea ice only in Figure 12b. This area includes all the ice within the domain delimited by the magenta and the green solid lines in Fig. 13(b,d). For this particular part of the sea ice cover, the differences in ice drift velocity magnitude are very significant, with an increase by more than 20% on October 16[th], and exceeding 40% on October 21[st]. Over the whole 10 days following the fragmentation event, the ice drift velocity for recently broken and compact sea ice increases on average by 7% in CPL_DMG compared to CPL_WRS.

We also note that for both events, the maximum in the ice drift velocity difference between CPL_DMG and the two other simulations (Fig. 12a,b) does not happen when the wind speed, ice drift velocity and wave height reach their maximum, but rather a few hours or days later (Fig.9a,b and Fig.12a). A possible explanation is that for strong winds and waves, the magnitude of the sum of the external stresses applied to the ice is high enough to overcome the ice internal stress (in all three simulations).

However, for lower wind speeds and wave heights, the magnitude of the external stress decreases. In these conditions, only sea ice with a high enough damage value will continue to be deformed. The effect of additional damage is therefore more likely to be maximum in the wake of a storm than during the storm itself.

Waves also impact the sea ice dynamics through the WRS. Figures 10(b,d) show the relative importance of the WRS compared to the wind stress, as well as the direction in which both apply. On October 21 (Fig. 10d), the WRS exceeds the wind stress over the first kilometres of the sea ice cover, where the sea ice is rather thin and not compact. As described in Boutin et al. (2020), the WRS direction tends to be aligned with the wind at the sea ice edge, but further into the ice cover it aligns with the gradient of ice concentration or thickness. This is because the mean wave direction in the ice cover corresponds to

the least attenuated waves, which tends to be the waves that have travelled the shortest distance in-ice. As a consequence, the WRS is most often directed on-ice, and is thus a source of sea ice convergence (Stopa et al., 2018b; Sutherland and Dumont, 2018). When the WRS is taken into account, the external stress applied to the sea ice under on-ice (respectively off-ice) wind conditions is enhanced (respectively reduced).

With our coupled-model, the impact of the WRS on the sea ice dynamics in the MIZ strongly depends on the activation of the link between wave-induced sea ice fragmentation and damage. In particular, these interactions between the WRS and sea ice damage contribute to some of the differences in ice drift velocity between the simulations we see on Figure 12a. For the October 21$^{st}$ case, we see a larger difference between CPL_DMG and CPL_WRS than between CPL_DMG and REF. In REF, the off-ice wind stress reduces the sea ice concentration, lowering its effective stiffness, so that sea ice drifts freely with the

wind. In CPL_WRS, the WRS cancels some of the wind stress, as well as compacting the sea ice, which allows it to resist the off-ice external stress more. In CPL_DMG, the previous damage to the ice from the waves lowers its stiffness and it again drifts freely in the off-ice direction. The WRS thus limits the rate of sea ice export in off-ice wind events, to an extent determined by the rheology of the broken ice. Conversely, on October 16$^{th}$ (Fig. 10b), the wind stress is directed on-ice and therefore aligned with the WRS. Moreover, the relative importance of the WRS exceeds that of the wind stress over a wide part of the

recently broken sea ice area. In these conditions, the WRS contributes to accelerate the convergence of sea ice, which results in a larger difference of average ice drift velocity over the domain between CPL_DMG and REF than between CPL_DMG and CPL_WRS (Fig. 12a).

      These interactions between sea ice damaging related to wave-induced fragmentation and WRS also have an impact on sea

ice properties in the MIZ. Figure 14(a,b) displays the differences in the sea ice thickness and concentration fields between CPL_DMG and REF. Compaction of sea ice in the MIZ is clearly visible, with thicker, more compact sea ice in the area between the magenta and green solid lines, which corresponds to compact sea ice that has recently been broken. In contrast, between CPL_WRS and REF there are almost no differences in the sea ice thickness fields (Fig. 14c). Sea ice concentration is only impacted over the first kilometres of the ice cover, with compaction of the sea ice visible on the western part of the

domain (*i.e.* lower sea ice concentration than in REF at the ice edge, but higher concentration slightly further in the ice cover,

Fig. 14d). From this, we can conclude that when sea ice is damaged by wave-induced fragmentation, it enables sea ice convergence over all the broken sea ice area. As a consequence, on-ice wind events lead to thicker, more compact sea ice in the MIZ, and this phenomenon is enhanced by the WRS. When wave-induced sea ice fragmentation has no impact on sea ice rheology, compacted sea ice, if it is not damaged, resists convergence, preventing sea ice from thickening in the MIZ in CPL_WRS.

Note finally that the relative thickening of sea ice in CPL_DMG increases the wave attenuation, leading to a lower broken sea ice extent in CPL_DMG compared to CPL_WRS (visible on Fig. 14(b,d) for instance). As an example, in the domain we defined in the Barents sea, over all October, sea ice thickness is on average 2.9% thicker in CPL_DMG than in CPL_WRS, while the ratio of recently broken sea ice surface area over the total sea ice surface area decreases by 2.4% between CPL_DMG

and CPL_WRS.

## 5 Discussion

In our case study, the damage added by wave-induced sea ice fragmentation does not significantly enhance sea ice deformation during wave-induced fragmentation events, but after them, when the sea state relaxes. This is because these fragmentation events coincide with high wind speeds, with wind stress dominating the internal stress of sea ice in all simulations, whatever

the level of damage is. Once the wind speed lowers, the internal stress of sea ice dominates over the wind stress in places where sea ice is compact and not damaged, and limits deformation. However, in the regions that have been previously damaged by wave-induced fragmentation, the level of damage remains high in the first days following the storm, and sea ice can still deform relatively freely. This high level of damage significantly enhances sea ice mobility in the MIZ in the CPL_DMG simulation compared to CPL_WRS. This behaviour of the MIZ, with fragmentation events followed by calm periods during which sea ice

mobility is enhanced, is not limited to the particular event we describe here. In the Barents Sea, for instance, maxima in the difference between ice drift velocities in the CPL_WRS and CPL_DMG simulations during October 2015 occur after maxima in the ice drift velocity magnitude (Fig. 15). We also noted a similar behaviour in the Greenland Sea (not shown).

The impact of wave-induced fragmentation on sea ice dynamics is expected to vary spatially and temporally with waves and sea ice conditions in the Arctic. From our results, the magnitude of this impact depends mostly on the distance over which

waves break the ice. At the beginning of October, the low sea ice extent combined with a high frequency of storms favours the occurrence of fragmentation events over a large area, and waves therefore have a strong impact on sea ice dynamics in the MIZ. This impact seems to decrease towards the end of October, as visible on Figure 15 where the difference in drift speed between CPL_DMG and CPL_WRS does not show any more peaks after the one on October 21[st]. It coincides with refreezing in the Barents Sea: the generation of thin, sea ice over wide areas reduces the fetch and damps the waves over long distances, before

they can break more compact sea ice. It is therefore likely that, in fall and winter, fragmentation will have little effect on sea ice dynamics as waves will be attenuated by the thin and loose forming sea ice. In contrast, in spring and summer, with the sea ice melt increasing the available fetch and exposing compact ice to the waves, we expect wave-induced sea ice fragmentation

to have a significant impact on the MIZ. Note also that in melting conditions, sea ice healing will not occur, lengthening the effects of fragmentation over time.

In addition to the effects of interactions between sea ice fragmentation and the WRS on sea ice dynamics, our study allows us to isolate the effect of the WRS in neXtSIM. The effects of the WRS in a coupled wave–sea-ice model system have already been discussed by Boutin et al. (2020) (coupling WW3 with the sea ice model LIM3) and Williams et al. (2017) (coupling neXtSIM and a simplified waves-in-ice model), and we will therefore only comment here on the main differences and similarities with these two studies. When the effect of wave-induced sea ice fragmentation on the dynamics is not included in the ice model (in CPL_WRS), the WRS pushes the sea ice edge towards pack ice and increases the sea ice concentration gradient over the first kilometres of the MIZ. This compaction is, however, limited to regions where sea ice poses little resistance to deformation, either because it has a low concentration (hence ice strength) or because it has been previously damaged by the wind. It is similar to the results of the study by Boutin et al. (2020), in which they note that sea ice drift is only impacted by the WRS in low-concentration sea ice areas. However, Boutin et al. (2020) also found an acceleration by nearly 10% of the sea ice drift velocity in areas where sea ice has been broken compared to an uncoupled reference simulation. This is not the case in our study. The acceleration observed by Boutin et al. (2020) is attributed to the effect of the WRS on sea ice drift in regions with low sea ice concentration where waves can be generated in ice. In our simulations, low-concentration regions allowing for in-ice wave generation are very limited, and this effect is therefore not present. These differences in the sea ice concentration distributions might be related to the differences in the sea ice models and external forcings, but also to the different time period investigated. It is also interesting to note that Boutin et al. (2020) report a decrease of the sea ice melt in their coupled ocean-sea ice-wave framework, explained by the on-ice drift force associated with the WRS pushing sea ice away from the open water. In our case, we have described how sea ice that has been damaged by wave-induced fragmentation can be exported by an off-ice wind despite a resultant stress reduced by the on-ice push of the WRS. It would therefore be interesting to add an ocean component to our wave–sea-ice coupled framework, in order to compare the effects of wave–sea-ice interactions on sea surface properties with the results found by Boutin et al. (2020).

In the study by Williams et al. (2017), the WRS was shown to have little effect on the ice edge location, even when sea ice fragmentation was lowering the sea ice internal stress. This difference with our study is likely to be due to the removal of the pressure term used with the EB rheology which gave some resistance to compression, which was especially needed once the ice became damaged (Rampal et al., 2016). Fragmented ice in our model is thus easier to pile up when convergence occurs. In reality, the collisions and subsequent transfer of momentum between the floes (Shen et al., 1986) should generate some internal stress that should resist sea ice convergence, lowering the thickening at the ice edge we report here. The acceleration of broken, compact sea ice we give here is therefore likely to be an upper bound of the effect of wave fragmentation on sea ice deformation.

Our results show how waves can modulate the extent of the region of sea ice drifting freely in the MIZ. In most sea ice models, this extent is generally a function of sea ice concentration and thickness, and ignores wave activity. Horvat et al. (2020) have recently used ICESAT-2 observations to show how the extent of an MIZ defined on a sea ice concentration criterion can differ from an MIZ defined on a wave activity criterion. It is similar to what we show here, and our model simply illustrates

potential effects of waves on sea ice dynamics in the MIZ. We show that to impact sea ice dynamics, waves need to propagate far enough in the ice cover to fragment compact ice. One of the main uncertainties of our results therefore lies in the estimation of the ice cover area affected by waves. In our model, the extent of this area could be affected by two factors: the relaxation time associated with healing of damage ($\tau_{\text{heal}}$), and the extent of broken ice.

The relaxation time associated with healing of damage $\tau_{\text{heal}}$ controls the speed at which the ice "forgets" the impact of previous fragmentation events. Its impact was investigated by re-running our experiments using $\tau_{\text{heal}}$=15 days instead of 25 days, the default value in neXtSIM. 15 days corresponds to the lower limit of the range of $\tau_{\text{heal}}$ values for which neXtSIM reproduces well the multi-scaling of sea ice deformation (Rampal et al., 2016), while 25 days is close to the upper-limit of this range. We found that changing the value of $\tau_{\text{heal}}$ had very little effect on our results. The sensitivity to $\tau_{\text{heal}}$ is possibly low

because once sea ice is broken, waves are able to propagate more easily in the sea ice cover and maintain quite a high level of damage until the wave activity strongly decreases, or until the sea ice cover extends.

    The extent of broken ice depends strongly on the wave attenuation computed in WW3 as discussed before in Boutin et al. (2018) and Ardhuin et al. (2018). The stronger the attenuation is, the narrower the extent of broken ice is, and the less likely waves will break compact ice and impact sea ice dynamics. We have extended here the parameterization evaluated by Ardhuin

et al. (2018) to a pan-Arctic simulation which can include very different sea ice conditions. This parameterization, like most wave-in-ice attenuation models, is very sensitive to the values of sea ice concentration and thickness (Doble and Bidlot, 2013; Nose et al., 2020), and the extent of broken sea ice we show here is therefore subject to very large uncertainties. The fact that storm waves can propagate and break the ice over tens to hundreds of kilometres in the Barents Sea is however not surprising (see the event reported in Collins et al., 2015), and coherent with the results of Horvat et al. (2020) who find that the Barents

Sea is the most consistent region of high wave-ice activity in the Arctic. Comparing our wave attenuation model with other parameterizations used in wave-sea interactions studies is not straightforward, as its attenuation varies strongly with sea ice thickness and floe size. It was shown by Boutin et al. (2018) to generally yield much more attenuation than the scattering model by Kohout and Meylan (2008) from which is derived the attenuation model used by Roach et al. (2018). Bennetts et al. (2017) use the empirical wave attenuation formula by Meylan et al. (2014), which is also implemented in WW3. This

empirical parameterization was shown by Collins and Rogers (2017) to produce rather strong wave attenuation compared to other empirical formulations. The formula by Meylan et al. (2014) does not account for floe size and thickness, but the sensors used in the experiment it is derived from were located on floes with freeboards of 10 cm or more (Kohout et al., 2016), and it is therefore calibrated for a total thickness of about 1 m or more. If we compare The Meylan et al. (2014) formula with our parameterization in the Beaufort Sea case used for the evaluation (not shown), our parameterization leads to a faster decay of

the wave height below $50\,\text{cm}$ (the level at which it stops breaking the ice in Ardhuin et al., 2018) when sea ice is thicker than 50 cm (not shown), and slower when sea ice is thinner. In our case study in the Barents Sea, sea ice thickness in the area broken by waves varies between $\simeq 25\,\text{cm}$ and $1\,\text{m}$, the wave attenuation we estimate should therefore be in line with the one given by Meylan et al. (2014) formula.

    The estimation of the extent of broken ice is also likely to depend on the model used to determine the occurrence of sea

ice break-up. We use a break-up model identical to most studies interested in wave-ice interactions (e.g. Williams et al., 2017;

Bateson et al., 2020; Roach et al., 2019). It remains extremely simplified and assumes that break-up only occurs in the case of flexural failure in one dimension. However, recent results from laboratory experiments (Herman et al., 2018; Dolatshah et al., 2018) tend to show that there is not such a clear relationship between the wave forcing and the floe size resulting from fragmentation. This is because a complete break-up model should include effects that are currently missing (e.g. from the floe shape and size, 2-D flexure modes, floe-floe collisions, rafting). We also point out that while our model includes some memory of previous fragmentation events, we do not account for the fatigue of the ice when determining if break-up occurs or not. The "slow-growth" FSD is used to keep a memory of the distribution of consolidated floes. It is associated with large-scale mechanical properties of the ice cover, while fatigue is related to the micro-structure of the ice. Accounting for fatigue could significantly lower the ice resistance to flexural failure in some events (Langhorne et al., 1998).

Although our main results are related to the effects of waves on sea ice dynamics, and only depend on the FSD's capacity to provide a good estimate for the maximum and average floe size used in the wave model, this work also takes the opportunity to introduce a coupled wave–sea-ice framework using two FSDs to distinguish two definitions of the floe size, one more relevant to thermodynamical processes, which grows fast when refreezing occurs, and one more relevant for dynamical and mechanical processes, which is associated with a slower growth of the floe size of interest. The processes included theoretically allow us to represent the evolution of the FSD all-year round, even though simulations over different and longer time periods require further evaluation. The introduction of these two FSDs highlights the importance of distinguishing the different timescales involved in the floe size evolution.

As in the studies by Zhang et al. (2015) and Boutin et al. (2020), the main uncertainty related to the FSDs concerns the way sea ice is redistributed after fragmentation. In these early days of the implementation of FSDs in sea ice models, we have built on what was done in wave-in-ice models and used a redistribution scheme that yields FSDs relatively similar to the ones described by Toyota et al. (2011), although their methods and interpretations have been contested (Stern et al., 2018; Horvat et al., 2019). Our redistribution scheme could easily be adapted in case progress were made on the relative importance of the parameters affecting this cut-off floe size and the associated redistribution, either from observations thanks to new available FSD datasets (Hwang et al., 2017; Horvat et al., 2019), or from discrete element modelling (Herman, 2018).

## 6 Conclusions

Using a coupled wave–sea-ice model, we have shown how waves may contribute to modifying the sea ice dynamics in the MIZ by modulating the extent of ice regions that pose little resistance to deformations. As noted by Horvat et al. (2020), this extent does not necessarily coincide with areas of low ice concentrations. With our model, we note a significant acceleration of compact sea ice in both convergent and divergent sea ice drift conditions once sea ice has been fragmented by waves. Even though some assumptions we make here require further evaluation, the results are of particular interest as they highlight missing physics in current modelling systems used for short and long-term sea ice predictions, and concern key areas of the polar regions.

Reliable sea ice forecasts are essential to ensure the safety of human activities close to the MIZ. In this context, waves pose a hazard as they make sea ice more mobile and our results therefore stress the need for the addition of wave effects in sea ice models used in forecasts. On longer timescales, the impact of waves on sea ice dynamics could affect the amount of sea ice that is exported to the ocean. Indeed, eddies and/or filaments are likely to play an important role in this export in cases where
5  sea ice dispersion is possible (Manucharyan and Thompson, 2017). Moreover, the fragmentation of sea ice itself could also generate sub-mesoscale (Horvat et al., 2016) and mesoscale activity in the ocean (Dai et al., 2019). Future coupling with an ocean model could therefore bring new insight into the interactions between waves, sea ice, and the ocean in the MIZ.

*Code and data availability.*  The wave–sea ice coupling routines developed in WW3 will be included in the future release of WW3 and made publicly available at: https://github.com/NOAA-EMC/WW3 (last access: November 2020). Model outputs are available upon request from
10  guillaume.boutin@nersc.no.

*Author contributions.*  GB and TW developed the theory and set up the coupling with the support of EO. GB led the analysis and wrote the manuscript in close collaboration with TW and contributions from all co-authors.

*Competing interests.*  The authors declare no competing interests.

*Acknowledgements.*  This work has been carried out as part of the Copernicus Marine Environment Monitoring Service (CMEMS) WIzARd
15  project. CMEMS is implemented by Mercator Ocean in the framework of a delegation agreement with the European Union. It has also been funded by institutional support from the Nansen Center. We thank the editor, the two anonymous reviewers and Christopher Horvat for their suggestions and their support for the publication of the manuscript. Finally, we thank Fabrice Ardhuin for providing the wave height data from SAR.

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

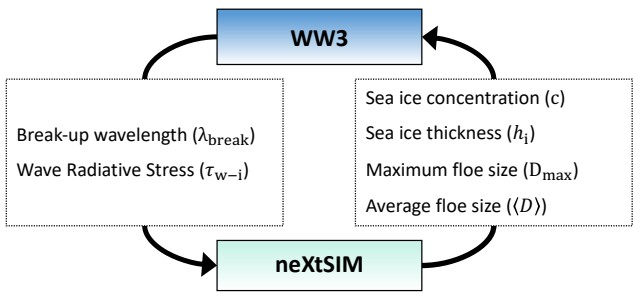

**Figure 1.** Summary of the exchanged variables in the neXtSIM-WW3 coupling framework.

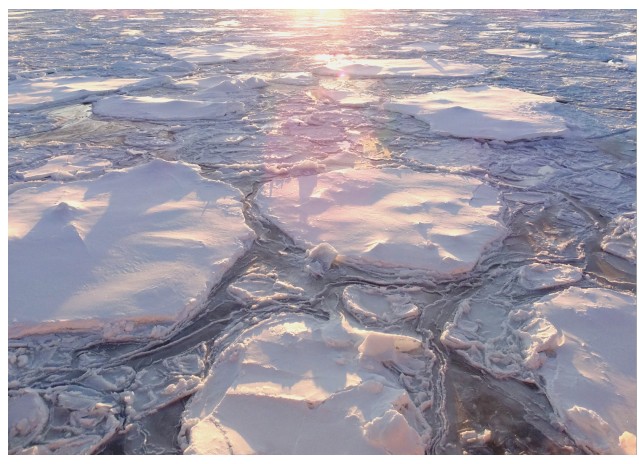

**Figure 2.** Aggregate of ice floes cemented together by thin ice in the Marginal Ice Zone (Weddell Sea, Antarctica). The size of the largest floes is of the order of 10 metres. Picture taken on board of RRS James Clark Ross in March 2014. Credit: *Heather Regan*

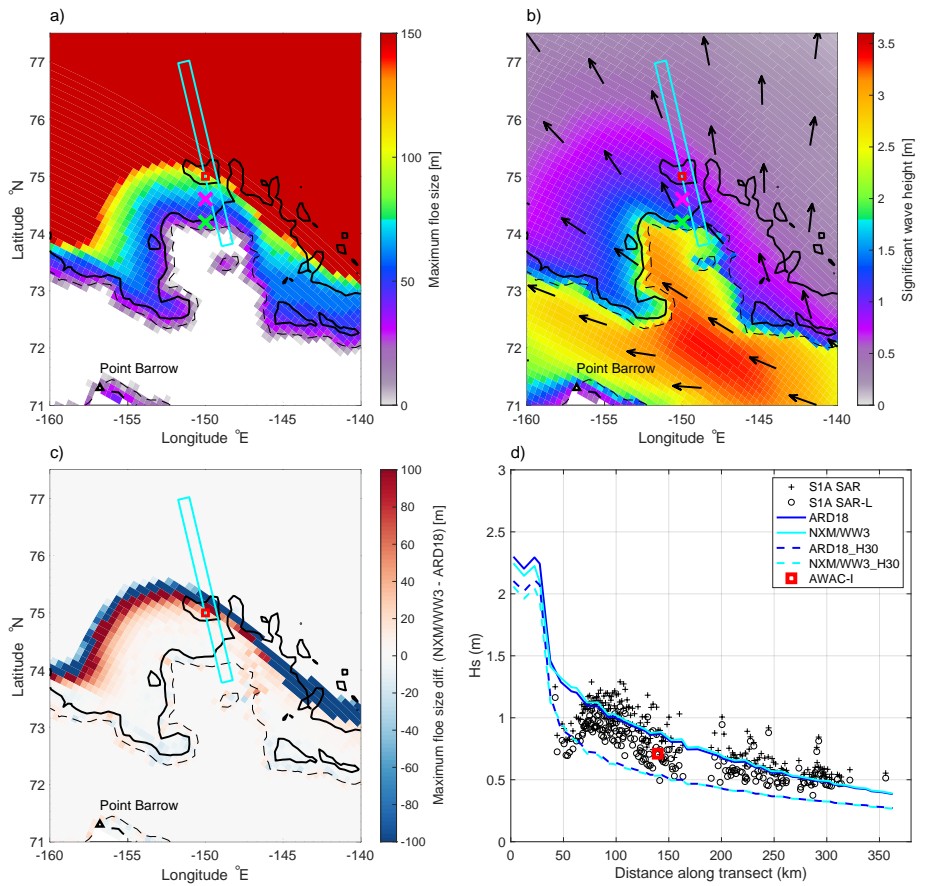

**Figure 3.** Spatial distributions of maximum floe size (a) and significant wave height (b) in the Beaufort Sea taken from the NXM/WW3 simulation on 12th October at 17:00:00 GMT. Black arrows indicate the wave mean direction. The difference of the maximum floe size distribution with the ARD18 simulation is shown on (c). Evolution of the significant wave height for different simulations along the transect depicted in cyan on panels (a,b,c) is presented on panel (d), along with significant wave height estimated from Sentinel-1a SAR images (see Stopa et al., 2018a; Ardhuin et al., 2018, for details) and measured by an AWAC buoy. The AWAC position is depicted by a red square on panels (a,b,c). The green and magenta crosses indicate the position at which are shown the FSD on Figure 4. Solid and dashed black lines represent contours of sea ice concentration equal to 0.8 and 0.15 respectively. Point Barrow location is indicated by a black triangle.

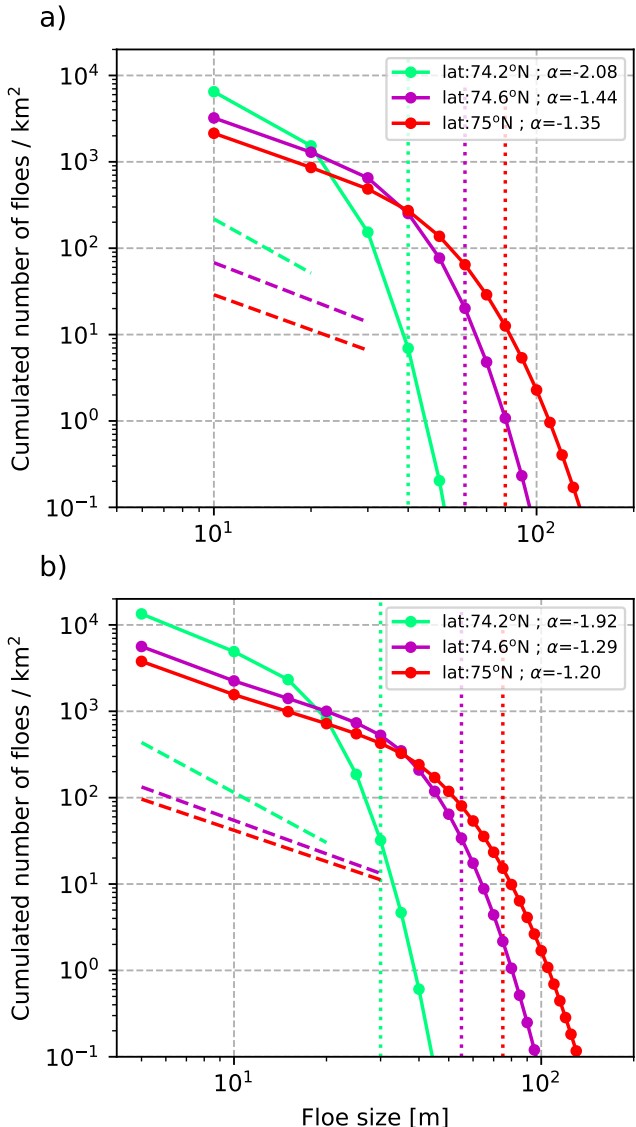

**Figure 4.** Cumulative distribution of floes taken at three different locations from the NXM/WW3 simulation (a), and from the NXM/WW3_refine simulation (b). The three locations share the same longitude (150°W) and their position is indicated by a symbol of the same color in Fig. 3(a,b) (red colour corresponds to the AWAC position). The dashed lines correspond to the linear regression over the smallest floe size categories for each location. Values of the slope are given in the legend. The vertical dotted lines represent the $D_{\mathrm{max}}$ values for each case.

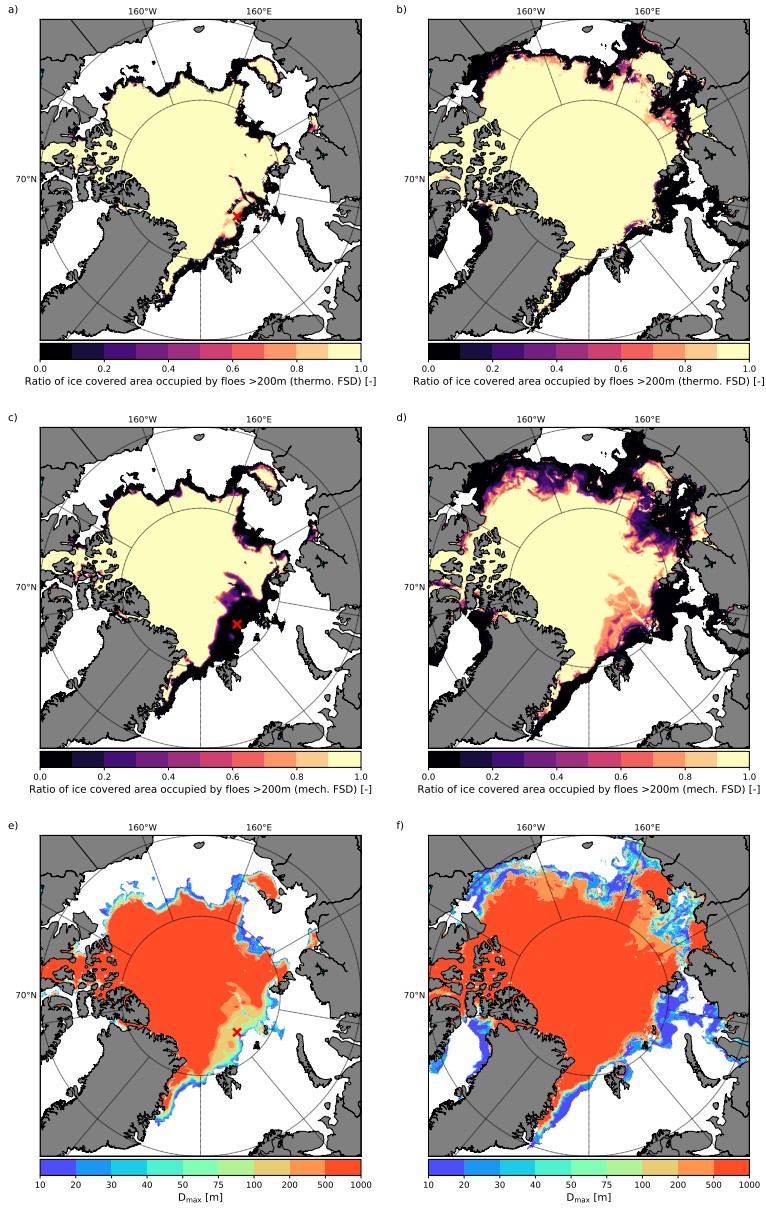

**Figure 5.** Pan-Arctic distribution of the area covered by floes with diameters larger than 200 m over the total sea ice cover area according to the "fast-growth" (a,b) FSD and to the "slow-growth" FSD (c,d), as well as the distribution of the maximum floe size (e,f). Each column corresponds to a different time: 02 October 2015 00:00:00 (a,c,e) and 01 November 2015 00:00:00 (b,d,f). The red cross (a,c,e) indicates the location at which the FSDs shown in Figure 6 and the floe size parameters in Fig. 7 are taken.

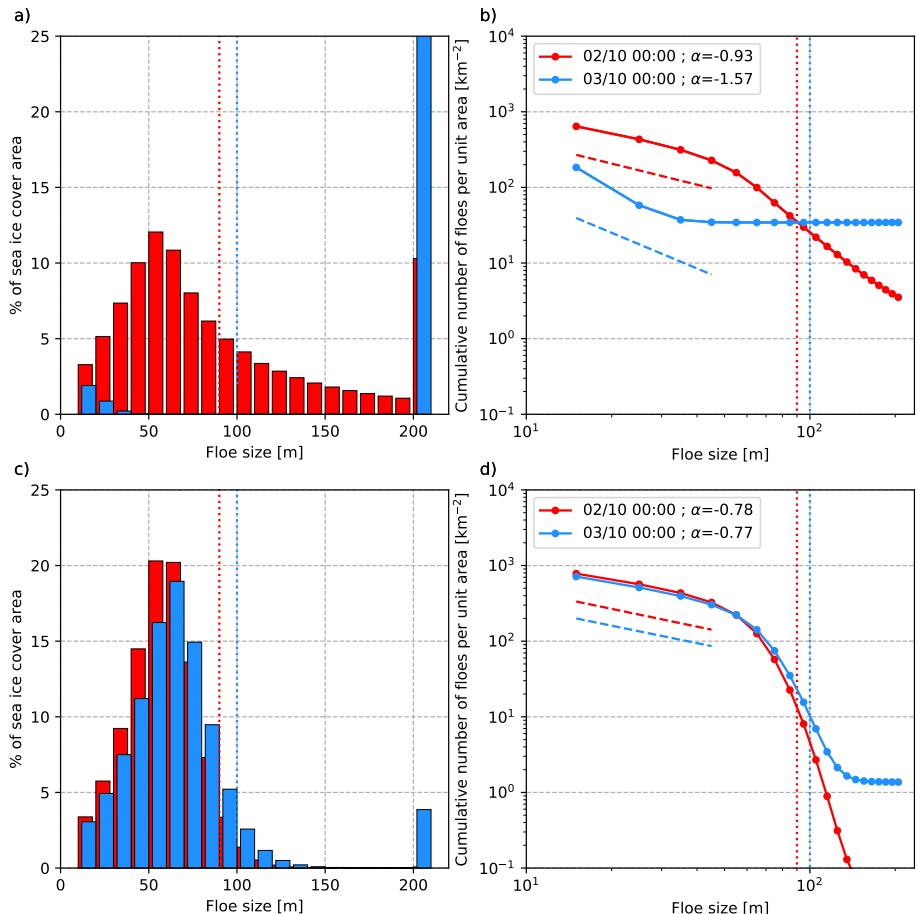

**Figure 6.** Areal (a,c) and cumulative distribution of floes (b,d) for the "fast-growth" (a,b) and "slow-growth" (c,d) FSDs taken at 45°E and 83.5°N. Each color refers to a different time: 02 October 2015 00:00:00 (red) and 03 October 2015 00:00:00 (blue). In (a,c), the bar at 200+ metres represent unbroken ice. The dashed lines correspond to the linear regression over the smallest floe size categories for each location. Values of the slope $\alpha$ are given in the legend. The vertical dotted lines represents the $D_{\max}$ values for each date.

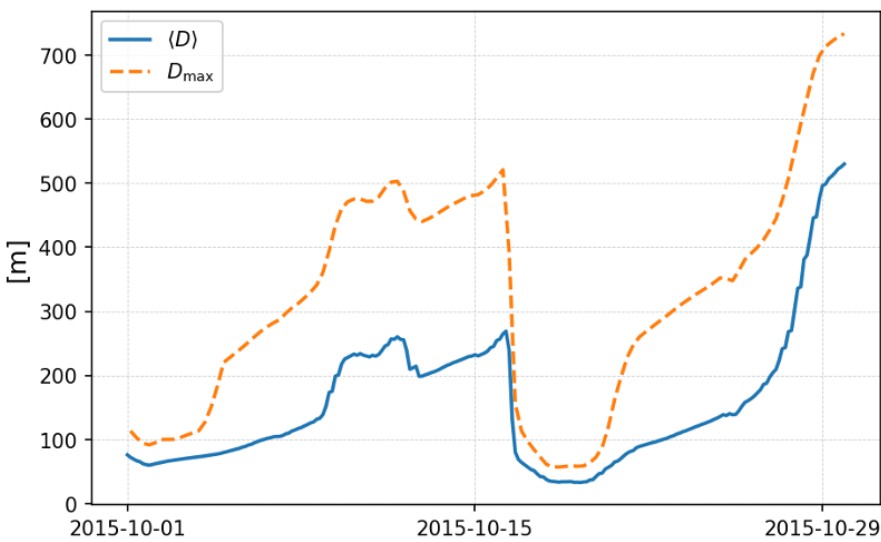

**Figure 7.** Temporal evolution of the two floe size parameters computed in the ice model and sent to the wave model taken at 45ºE and 83.5ºN (see red cross in Fig. 5(a,c,e)).

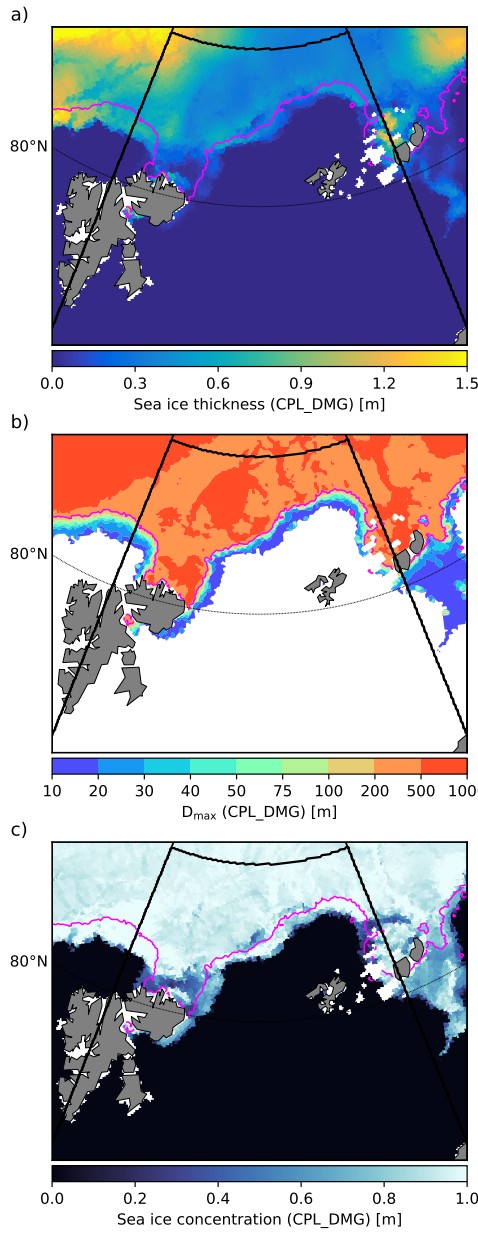

**Figure 8.** Distributions of sea ice thickness (a), maximum floe size (b) and sea ice concentration from the CPL_DMG simulation in the Barents sea on October 15$^{\text{th}}$ 2015, at 00:00:00. The magenta line corresponds to the contour $D_{\text{max}} = 200\,\text{m}$ in CPL_DMG. The black thick lines delimit the domain used to analyse results from the 3 simulations (REF, CPL_WRS, CPL_DMG) in figures 9, 12 and 15.

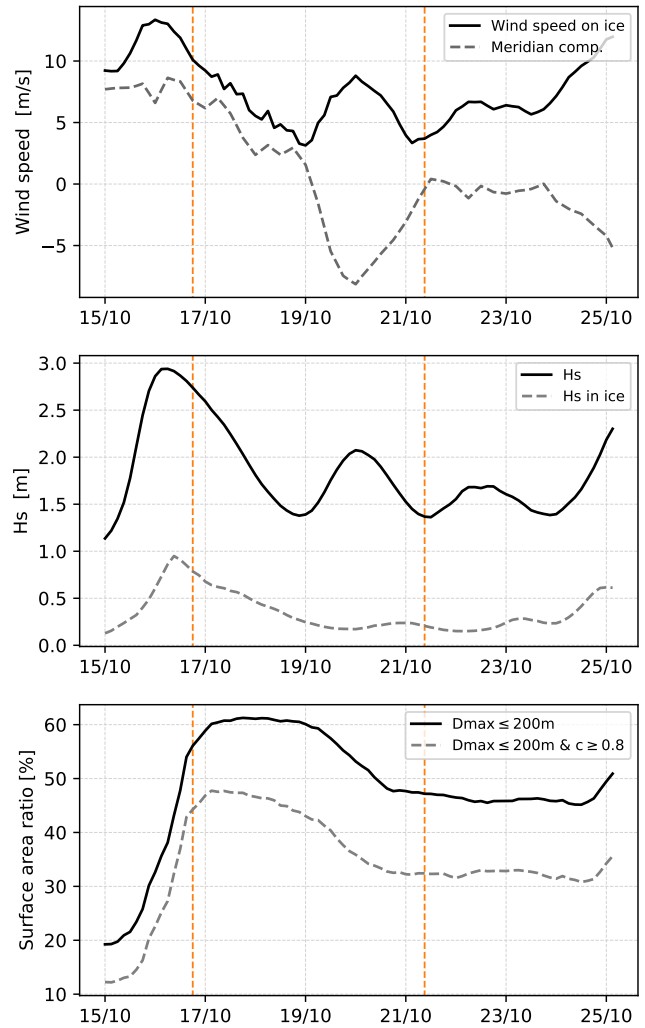

**Figure 9.** Temporal evolution of (a) the wind speed (black solid line) and its meridional component (dashed grey line) averaged over all the ice-covered part of the domain (see Fig. 8 for the domain definition), (b) the significant wave height averaged over all the domain (black solid line) and ice-covered part of the domain only (dashed grey line), and (c) the ratio of the surface area of regions covered by recently broken sea ice (defined as $D_{\max} \leq 200\,\text{m}$, black solid line) and compact sea ice that has been recently broken (defined as $D_{\max} \leq 200\,\text{m}$ and $c \geq 0.8$, grey dashed line) over the total sea ice-covered surface area. The time period shown covers from October $15^{\text{th}}$ to October $25^{\text{th}}$ 2015, for which initial conditions are given in Figure 8. The sea ice-covered part of the domain is the area for which the sea ice concentration $c$ is greater than 0. The two orange vertical lines indicates the dates of the snapshots shown in Figures 10, 11 and 13.

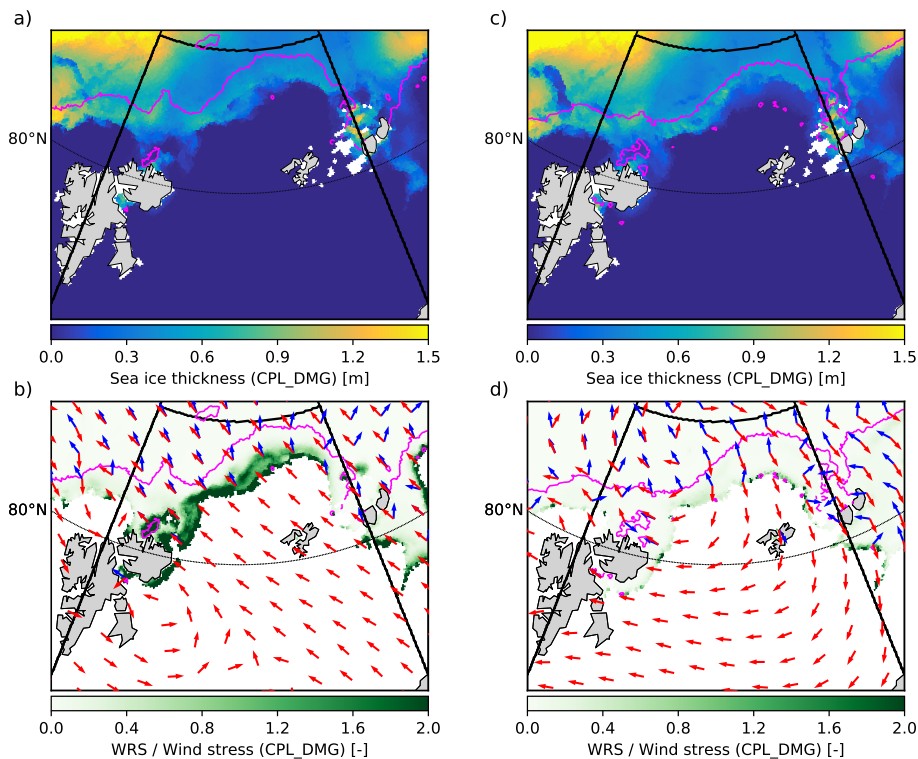

**Figure 10.** Snapshots of the distributions of sea ice thickness (a,c), and of the ratio between the WRS and the wind stress over sea ice (b,d) from the CPL_DMG simulation in the Barents sea taken on October $16^{\text{th}}$ 2015 at 18:00:00 GMT (a,b), and on October $21^{\text{st}}$ 2015 at 09:00:00 GMT (c,d). The wind and WRS directions for each date are given by the green and blue arrows respectively on panels (b,d). The magenta line corresponds to the contour $D_{\max} = 200\,\text{m}$ in CPL_DMG. The black thick lines delimit the domain used to analyse results from the 3 simulations (REF, CPL_WRS, CPL_DMG) in Figures 9, 12 and 15.

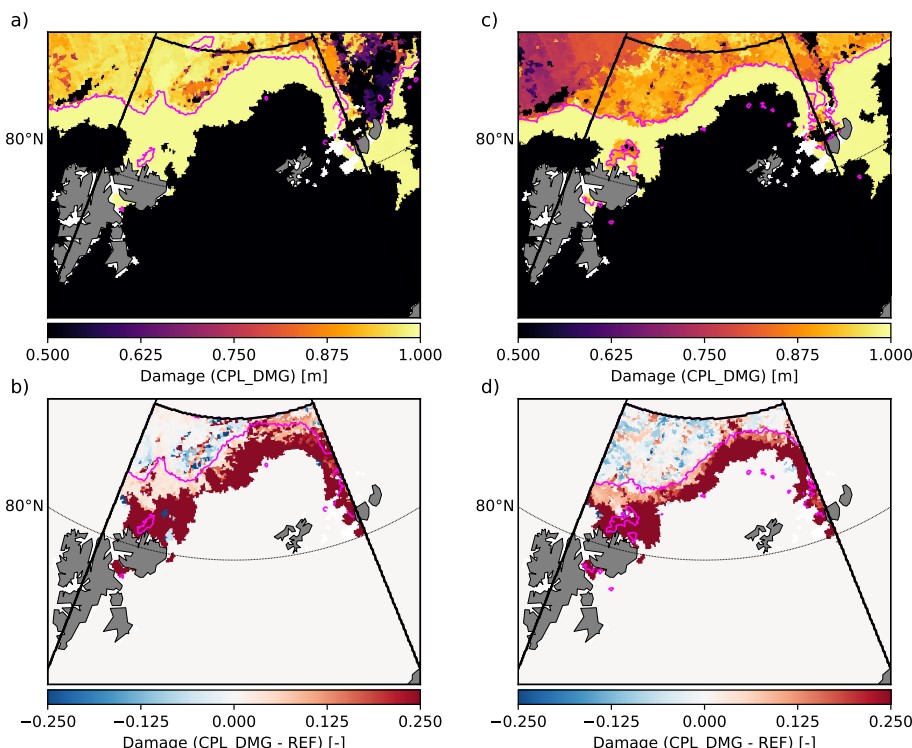

**Figure 11.** Snapshots of the distributions of sea ice damage in the CPL_DMG simulation (a,c), and of the differences in damage between the CPL_DMG and REF simulations (b,d) in the Barents sea taken on October 16$^{\text{th}}$ 2015 at 18:00:00 GMT (a,b), and on October 21$^{\text{st}}$ 2015 at 09:00:00 GMT (c,d). The magenta line corresponds to the contour $D_{\text{max}} = 200\,\text{m}$ in CPL_DMG. The black thick lines delimit the domain used to analyse results from the 3 simulations (REF, CPL_WRS, CPL_DMG) in Figures 9, 12 and 15.

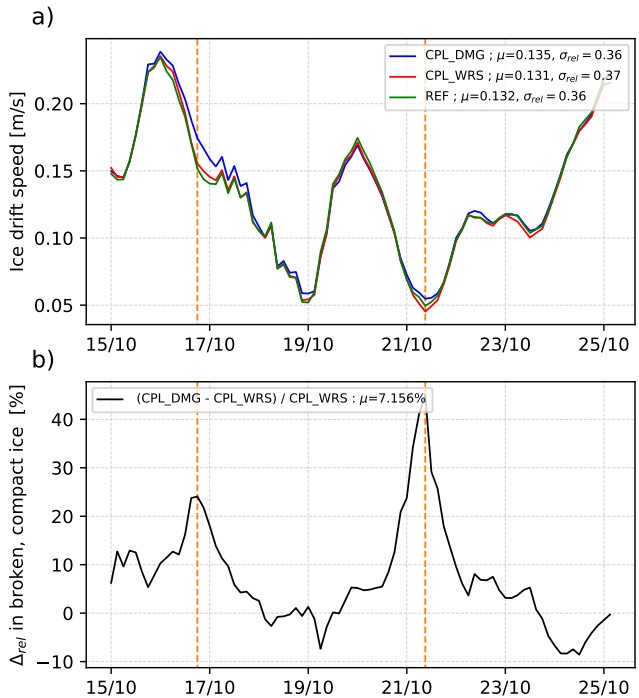

**Figure 12.** Temporal evolution of (a) the ice drift velocity averaged over all the ice-covered part of the domain (see Fig. 8 for the domain definition) for the three simulations (REF in green, CPL_WRS in red, and CPL_DMG in blue), and (b) the difference of ice drift velocity in the region covered by compact sea ice that has been recently broken (defined as $D_{max} \leq 200\,\text{m}$ and $c \geq 0.8$) between the CPL_WRS and CPL_DMG simulations. The time period shown covers from October $15^{th}$ to October $25^{th}$ 2015, for which initial conditions are given in Figure 8. The sea ice-covered part of the domain corresponds to the area for which the sea ice concentration $c$ is greater than 0. The two orange vertical lines indicates the dates of the snapshots shown in Figures 10, 11 and 13. In the panels legends, $\mu$ indicates the temporal mean associated with each curve, and $\sigma_{rel}$ is the standard deviation divided by the mean.

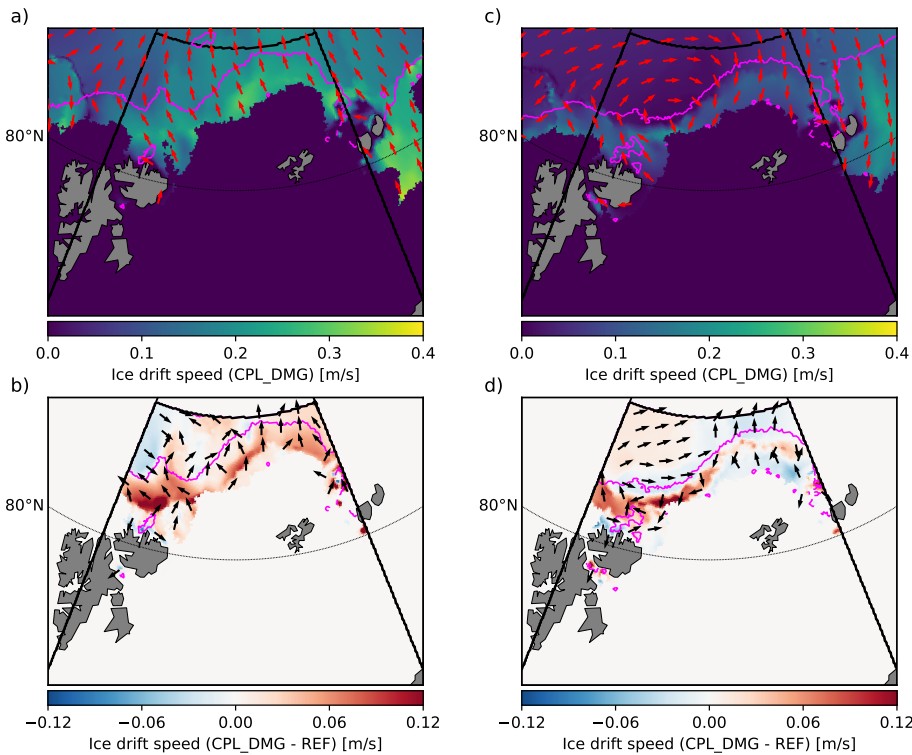

**Figure 13.** Snapshots of the distributions of sea ice drift velocity in the CPL_DMG simulation (a,c), and of the differences in sea ice drift velocity between the CPL_DMG and REF simulations (b,d) in the Barents sea taken on October 16[th] 2015 at 18:00:00 GMT (a,b), and on October 21[st] 2015 at 09:00:00 GMT (c,d). The green line delimits the area with compact sea ice (defined as $c \geq 0.8$) in CPL_DMG. The magenta line corresponds to the contour $D_{\max} = 200\,\text{m}$ in CPL_DMG. The black thick lines delimit the domain used to analyse results from the 3 simulations (REF, CPL_WRS, CPL_DMG) in Figures 9, 12 and 15.

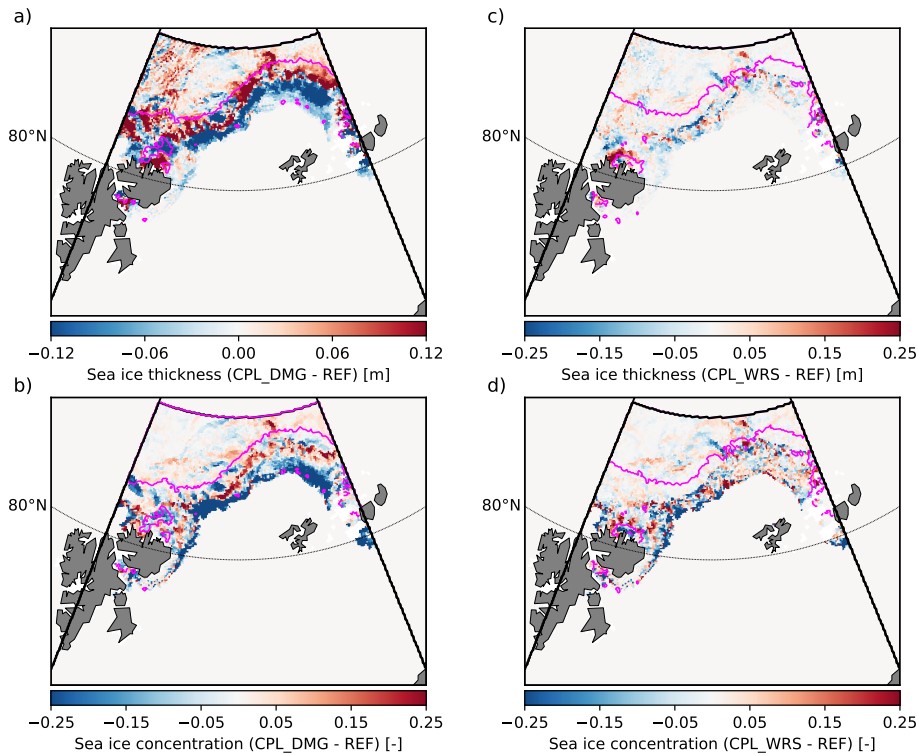

**Figure 14.** Snapshots of the distributions of difference in sea ice thickness (a,c) and sea ice concentration (b,d) between the CPL_DMG and the REF simulations (a,b), and between the CPL_WRS and REF simulations (c,d). All these snapshots are taken on October 21$^{st}$ 2015 at 09:00:00 GMT. The magenta line corresponds to the contour $D_{max} = 200$ m in CPL_DMG (a,b) and in CPL_WRS (c,d). The black thick lines delimit the domain used to analyse results from the 3 simulations (REF, CPL_WRS, CPL_DMG) in Figures 9, 12 and 15.

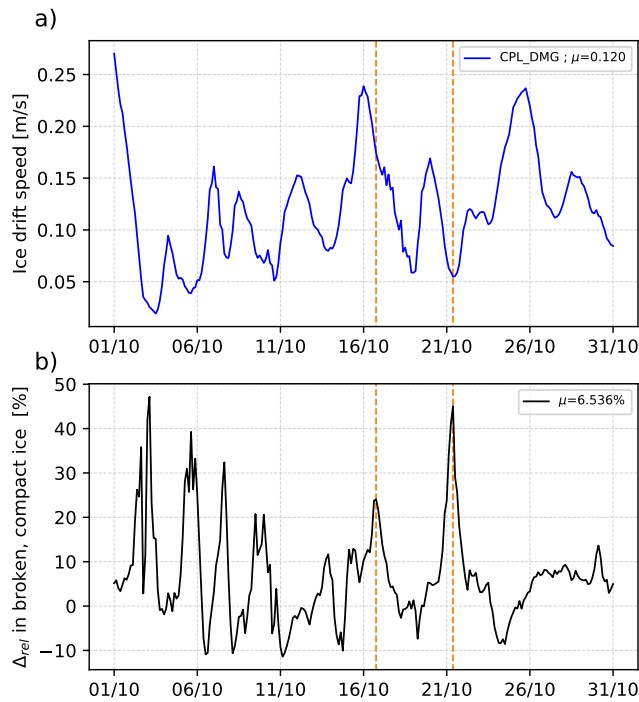

**Figure 15.** Temporal evolution of (a) the ice drift velocity averaged over all the ice-covered part of the domain (see Fig. 8 for the domain definition) for the CPL_DMG simulation, over October 2015, and (b) of the difference of ice drift velocity in the region covered by compact sea ice that has been recently broken (defined as $D_{\max} \leq 200\,\mathrm{m}$ and $c \geq 0.8$) between the CPL_WRS and CPL_DMG simulations. The sea ice-covered part of the domain corresponds to the area for which the sea ice concentration $c$ is greater than 0. The two orange vertical lines indicates the dates of the snapshots shown in Figures 10, 11 and 13. In each panel legend, $\mu$ indicates the temporal mean of the plotted quantity.

**Table A1.** List of symbols used in this study and their associated quantities. Parameters default values, or references used to compute each quantity, are indicated in the last column.

| Symbol | Quantity | Value/Reference |
|---|---|---|
| $c$ | Sea ice concentration | - |
| $h$ | Sea ice thickness | - |
| $D$ | Floe size (mean caliper diameter) | - |
| $D_{\mathrm{max}}$ | Maximum floe size | Initialized with 1000m |
| $Y$ | Young modulus | 5.49 GPa |
| $\nu$ | Poisson's ratio | 0.3 |
| $G_r$ | Lateral melt rate | see Maykut and Perovich (1987) |
| $\dot{c}_{\mathrm{new}}$ | Rate of formation of new ice | see Rampal et al. (2016) |
| $\beta_{\mathrm{weld}}$ | FSD redistribution term associated with welding of floes | see Roach et al. (2018) |
| $\kappa$ | Rate at which the number of floes decreases due to welding per surface area | $5 \times 10^{-8} \mathrm{m}^{-2}\mathrm{s}^{-1}$ |
| $\tau_{heal}$ | Relaxation time associated with damage (mechanical) healing | 25 days (default) |
| $\Delta t_{\mathrm{ice}}$ | Ice model time step | 20s |
| $\Delta t_{\mathrm{cpl}}$ | Coupling time step | 2400s |
| $\tau_{WF}$ | Relaxation time associated with fragmentation | 1800s |
| $Q$ | Redistribution probability function of floe size associated with fragmentation | see Zhang et al. (2015) |
| $\beta$ | Redistribution factor of floe size associated with fragmentation | see Zhang et al. (2015) |
| $\lambda_{\mathrm{break}}$ | Shortest wave wavelength triggering flexural failure of sea ice | see Boutin et al. (2018), sec. 2.3 |
| $D_{\mathrm{FS}}$ | Minimum floe size for flexural failure | see Mellor (1986) |
| $p_{\mathrm{FS}}$ | Probability that ice breaks depending on $D/D_{\mathrm{FS}}$ | Eq. 9a |
| $p_\lambda$ | Probability that ice breaks depending on $D/\lambda_{\mathrm{break}}$ | Eq. 9b |
| $c_{1,\mathrm{FS}}$ | Value of $D/D_{\mathrm{FS}}$ under which ice cannot be broken | 1. |
| $c_{2,\mathrm{FS}}$ | Param. controlling the range of $D/D_{\mathrm{FS}}$ over which $p_{\mathrm{FS}}$ goes from 0 to 1 | 2. |
| $c_{1,\lambda}$ | Value of $D/\lambda_{\mathrm{break}}$ under which ice cannot be broken | 0.3 |
| $c_{2,\lambda}$ | Param. controlling the range of $D/\lambda_{\mathrm{break}}$ over which $p_{\mathrm{FS}}$ goes from 0 to 1 | 2. |
| $c_{\mathrm{broken}}$ | Concentration of broken sea ice | Eq. 13 |
| $d_w$ | Damage value associated with broken ice | 0.99 |
| $\alpha$ | Exponent of the small-floe regime power-law FSD | see Toyota et al. (2011), note that the sign is reversed |
| $q$ | Exponent used in the redistribution factor | Eq. 11 |