# Peer review of "Wave-sea-ice interactions in a brittle rheological framework"

_The Cryosphere, 2020_

## Referee Comment (RC1) · Anonymous Referee #1 · 24 Feb 2020

This is a review of Boutin et al (2020) - who explore the interaction of wave fracture and improved rheological modeling in the neXtSIM model.

The paper is interesting and a piece of model development that ought to be done and published. My major comments are on their 2-FSD parameterization.

Main comment: a few questions about your model.

In your 2-FSD implementation, I would like some more clarity on the meaning of the "mechanical" FSD - this is an interesting idea. My read is the point is to provide memory of past deformation - but how is this separate from the role of damage in neXtSIM/MEB?

If one wanted to compare your output to observations, how would you do that?

[Figure]

Why should we expect your mechanical FSD to look like the thermodynamic one, i.e. obey the same evolution equations? Why should the mechanical healing term look like the thermodynamic one, couldn't it evolve independently? I think a figure to add would be plotting the mean floe size for both FSDs in time for the period documented in Fig 13, even for just a single grid cell.

The impact is clearly seen in Fig 4. Were I designing a separate depiction of sea ice fracturing, I'd expect it to be most relevant in the interior pack - this is where FSD models don't get crack features right yet. This mechanical FSD implementation seems to pinch in near the margins, not in dynamically active but waveless regions, but the neXtSIM model does get damage in the interior, doesn't it?

This leads me to believe that there is a difference between the description of where the mechanical FSD would be relevant (Sec 2.2, i.e. interior regions with leads) and where your model makes it relevant (exterior regions with waves). I think this approach is potentially fruitful for fixing the problem of bad pack ice fracturing, but you may be approaching it from the wrong place!

Minor Comments:

Please remove the mention of eddies from your abstract - the role of the ocean is not explored here except to cite a couple papers in the conclusions.

P2 L4 - I think you forget to explicitly mention the second main process?

Pg 2 - Using the power law FSD, especially in early days, is fine, but just note that meta-analyses (Herman 2011, Stern 2018) and new datasets (Horvat et al 2019) indicate an absence of power-law tails. Still... it is tough to justify (or putting the cart before the horse) designing a model that gets an answer, and then forcing its conservation via parameter choices. This is a particular problem because it is majority opinion that the "cutoff" power laws observed by Toyota and others are an artifact of the use of CDFs and finite measurement windows (Burroughs and Tebbens 2001, Stern 2018)

[Figure]

not physics. Now a model has been designed (more than one) that produce these distributions. But you have no windowing issues (so no expectation of a truncated power law distribution) or sampling issues (so no need to produce a CDF). I'd advise plotting the FSDs proper alone (as you do in Fig 5), living with the results. At these early days, you'll be forgiven for having weird distributions, and for making changes to your models, too.

Pg 3 L 5 and otherwise - (ITFSD –> FSTD).

Pg 5 L 30 it has been pointed out by Stern (and a wide literature from applied math, see Virkar and Clauset 2014) that fixed-width bins will bias your ability to represent or examine scale-invariant behavior.

P 6 - I think the most updated Roach model was published in 2019 and included coupled wave-ice physics. Might provide better sourcing for the comparison here.

P 6 L29 - do you mean that once the concentration is high, all the ice is in the highest size category? Is this also true for the mechanical FSD, or do you still require the relaxation?

P 7 L 25 "a quick and violent process" is a wonderful phrase albeit not exactly accurate. I know I should object scientifically but I really like it.

P9 L 25 - See earlier comment. At the very least, please explain these parameter choices naturally through your model design not as a post-facto requirement.

P13 L 5 "it also includes storms" - could you be more clear about what you mean here?

P13 L21 - What does it mean "very satisfactory results"? What is the metric?

P13 L27 - "Perfectly acceptable given the uncertainties" - I'm not sure what you mean - which is perfectly acceptable, and why does this relate to wave attenuation uncertainties?

P15 L2 - why not show this contour?

P16 L1 - "It is particularly true..." - rewrite?

P16 L17 - A bit confused here, "regenerate unbroken ice" isn't really the process - healing between floe joints is how you describe it.

Fig 3 - Again I'd advise not using the CDF here, preferring the FSD because as pointed out by Stern (2018) the CDF gives a false impression of scale-invariance, and

P19 - I would prefer a clearer description of this process. In effect, you are saying that the influence of fragmentation (at least in your model) is not because of wave events, but after them when the sea state relaxes?

P20 L 35 - "it depends on two factors" but then you mention it doesn't depend on reducing t_heal to 15 days. Also, this isn't a sensitivity experiment as you haven't also increased the healing time. You also don't really address the sensitivity to attenuation just mention it is uncertain.

---

## Referee Comment (RC2) · Anonymous Referee #2 · 1 Mar 2020

This paper follows a series of recent works by the authors and others in coupling ocean waves and sea ice in large scale models. Here, the Wavewatch III and neXtSIM models are coupled, and simulations with different levels of coupling are compared, with a focus on fragmentation of the ice cover and resulting changes in ice dynamics.

The main contribution of the paper is the inclusion of FSD memory, by using two FSD functions where one FSD (the "dynamic" FSD) evolves more slowly than the other one (the "thermodynamic" FSD). This is an interesting new avenue in sea ice modelling, and I'm surprised it hasn't been highlighted in the title of the paper. The authors motivate the memory by saying "there are reports of waves breaking ice at weak points such as refrozen leads and pressure ridges (e.g. Kohout et al 2016)" (p4, l3–4), but only ever refer to the one paper (Kohout et al, 2016). Are there any more reports of this kind?

[Figure]

If so, give them. If not, weaken the motivating statement. In either case, it would be useful to give brief descriptions of the events reported. I found the names of the FSDs difficult, as both depend on dynamic and thermodynamic source terms. Calling them, e.g., slow and fast would be easier (and probably more accurate).

Another motivation given for the study is that "neXtSIM is now including a Maxwell Elastomer-Brittle rheology" (p3, l31). But there is no explanation of why the MEB rheology is an improvement over the EB rheology for modelling marginal ice zone dynamics, or rheologies used in other models. Even if the authors don't intend this as a main motivation, they should discuss the relevance of the MEB rheology for the MIZ.

Wave attenuation, ice fragmentation and wave radiation stress models are very important for this study. The developments of these models, key assumptions, etc should be discussed, as all the models contain large uncertainties. Notably, only one sentence is given to wave attenuation models in the introduction (p2, l24; not including the short sentence referencing review papers), despite uncertainties in attenuation being identified as important later (bottom of p20). The scattering reference (Montiel & Squire, 2017) is actually a fragmentation (or ice breakup) study (see its title) and should be used elsewhere. The scattering model used for that study is the 3D model by Montiel et al (2016), but I'm not aware of 3D scattering models being available in Wavewatch III. The review of fragmentation models focusses on the FSD shape, and overlooks the models used to predict if waves are capable of breaking the ice cover. The lab model by Herman et al (2018) is relevant here (noting that it is for regular, low steepness waves only), as is the lab model by Dolatshah et al (2018). Similarly, the WRS model needs to be discussed; based on Williams et al (2017), it seems to contain large uncertainties and arbitrary assumptions.

Other comments, suggestions and questions:

The statement "There are two main processes through which waves can affect sea ice dynamics" (p2, l3) is far too strong. What about, e.g., collisions? Just say "we

investigate two processes..." or similar. The subsequent discussion on the role of WRS needs to be more balanced, e.g. Williams et al (2017) found that "wind stress dominates the WRS", and Alberello et al found negligible WRS in a pancake ice MIZ, even during large wave events.

On p2 l15, replace "fragmentation" by "floe size", as neither Shen et al (1986) nor Feltham (2005) included fragmentation in their models.

Saying "their FSD depends only on the wave field" (p3, l8) is not true, as D_max depends on the ice properties, as does the breaking criterion.

The work by Rynders (2018) is conspicuously missing from the introduction.

The line at the top of p6 is awkward and should be reworded. Regarding the reference to Roach et al two lines below, please clarify if you are considering an FSD or an ITFSD?

Does "lateral melt will not be discussed here" mean that it won't be included in the model for the study?

The opening paragraph of 2.2.2 is very long-winded for describing a simple method. Please give a reference to back the statement "sea ice fragmentation is a violent phenomenon ... impact the floe size". This doesn't seem to allow for fatigue.

Do the lines at the top of p8 mean D_N=1000m? On l10, "freedom" doesn't seem to be the correct word.

Say a bit more about tau_WF below Eq 7. Is it a numerical parameter of does it have physical meaning?

On p8 l11, please check the interval bounds, and on l13 reword "over within which".

At the bottom of p9, what exactly is being conserved?

Please clarify the two sentences starting p10 l19. Also, is this the ice-coupled or openwater wavelength?

Should it be "fragmentation and/or refreezing" on p11 l6?

Has the sensitivity of the the coupling time step between the wave and ice models been tested (section 3.1)? Also, why is tau_heal set to 25 days?

You say "the other main novelties..." (p14 l11), but this is the first mention of novelty.

The CPL simulations first appear in 4.1.2, but in section 3.3 it says they will be used in section 4.2.

Explain the statement "this quick re-generation ... making welding very efficient" (p15, l8).

What "impact of waves on sea ice dynamics" is being referred to in section 5? Also, does "large fragmentation events" mean fragmentation over a large area or something else?

The sentence at the end of the first paragraph of p21 is incorrect. Bennetts et al (2017) use a parameterization based on in-situ measurements by Meylan et al (2014). More generally, starting with Bennetts & Squire and Williams et al (2013a,b), it is usual to model attenuation using a viscous dissipation term for low-frequency waves and a scattering term for mid-frequency waves (see also Squire & Montiel, 2016).

Check the inequalities on l4 and l9 of p23.

What are the vectors in Fig 2b?

There are quite a few typos to correct.

References

Alberello et al, Drift of pancake ice floes in the Antarctic marginal ice zone during polar cyclones, arXiv.1906.10839

Bennetts & Squire, 2012, On the calculation of an attenuation coefficient for transects

of ice covered ocean, Proceedings of the Royal Society of London A, 468

Dolatshah et al, 2018, Hydroelastic interactions between water waves and floating freshwater ice, Physics of Fluids, 30

Meylan et al, 2014, In-situ measurements and analysis of ocean waves in the Antarctic marginal ice zone, Geophysical Research Letters, 41

Montiel et al, 2016, Attenuation and directional spreading of ocean wave spectra in the marginal ice zone, Journal of Fluid Mechanics, 790, 492-522

Rynders, 2018, Impact of surface waves on sea ice and ocean in the polar regions, University of Southampton

Squire & Montiel, 2016, Evolution of directional spectra in the marginal ice zone, Journal of Physical Oceanography, 46

---

## Referee Comment (RC3) · Anonymous Referee #3 · 4 Mar 2020

In this work the authors coupled a wave model with a sea-ice model to investigate the impact of wave-induced sea-ice fragmentation on the sea-ice floe size distribution (FSD) and sea-ice dynamics. The focus is on the Barents Sea in October 2015. To study the FSD, five simulations are run: coupled and uncoupled runs with sea-ice thickness equal to 15 cm and 30 cm, and a coupled run with smaller floe size bins (more floe size categories). To study sea-ice dynamics, three simulations are run: one with a stand-alone sea-ice model (REF), one with wave radiative stress (CPL_WRS), and one with "damage" (CPL_DMG). The result is that waves modify sea-ice dynamics in the marginal ice zone (MIZ) by lowering the resistance of ice to deformation. The authors recommend that waves be included in sea-ice models to improve their forecasts.

I have not looked at the other reviews of this paper. This is an independent review.

Main Comments

My main concern is with the FSD analysis. See page 14, lines 20-21, in reference to Figure 3: "we can distinguish two regimes separated by a cut-off floe size..." Look at Figure 3(a). I do not see two regimes separated by a cut-off floe size, and I don't believe that any statistical test would support such a conclusion. Look at the green curve for latitude 74.2 degrees north. It appears that a "line" has been fit using exactly 2 data points (see the green dashed line). By this method of analysis, one could distinguish a new "regime" for every pair of points. The purple and red dashed lines appear to be based on 3 data points. To my eye, all three curves appear to gradually steepen as the floe size increases. I don't see a cut-off or a regime shift.

The authors cite Toyota et al (2011) numerous times in the context of concave-down cumulative distribution functions (CDFs) with two regimes. A counterpoint may be found in this paper:

Stern, H.L., A.J. Schweiger, J. Zhang, and M. Steele, 2018. On Reconciling Disparate Studies of the Sea-Ice Floe Size Distribution, Elem Sci Anth, 6: 49. DOI: https://doi.org/10.1525/elementa.304

In particular, see their Figure 3 and the section called "Break-point analysis".

Page 23, Appendix B. "The shape of the CDFs shown in Figures 3 and 5 strongly depend on the parameterization detailed in section 2.2.2. The value of the cut-off floe size at which the transition between the small and large floes regime happens..." It seems highly undesirable that the shapes of the CDFs depend strongly on the parameterization. This would seem to inject a high degree of uncertainty into the whole simulation. And again, I question that a well-defined cut-off exists between small and large floes.

Minor Comments

Page 1, line 25. It looks like Lemieux et al (2016) is about landfast ice, not the sea-ice edge.

Page 6, line 21. "floes in the largest floe category are not affected by lateral melt." I don't see how equation (4) reflects this statement.

Page 7, lines 2-3. "a uniform FSD made of the smallest floes ... evolves into a uniform FSD made of the biggest possible floes." This does not make sense. A uniform FSD contains floes of all sizes, in equal proportions. The authors probably mean a delta-function FSD, in which all floes are of the smallest size, evolves into a delta-function FSD, in which all floes are of the largest size.

Page 7, line 4. Check whether "uniform FSD" is appropriate here – see previous comment.

Page 7, line 6. "setting kappa = 5 x 10ˆ(-8)" kappa is a rate (see line 1 of page 7). Please give the units.

Page 8, equation (8) and following. You need to say that Y is Young's modulus, nu is Poisson's ratio, and h is ice thickness. Please give values of DFS for h = 15 cm and h = 30 cm.

Page 10, equation (12). This equation is not correct – it is missing a factor of D inside the integral. If g(D) is a probability density function then the mean value of D is the integral of D*g(D) dD.

Page 10, lines 21-24. Dmax is supposed to be one order of magnitude larger than the longest wavelength, but lines 23-24 imply that Dmax does not become larger than 1000m. Shouldn't Dmax be 10 times larger than 1000m?

Page 11, end of Section 2. There are a LOT of parameters and empirical functions in this work. It might help to collect them in a table. My list includes these parameters: Gr, c_new, and beta_weld from equation (4); kappa from page 7; tau_heal from equation (5); tau_WF from equation (7); lambda_break, c_1FS, c_2FS, c_1Lambda, c_2Lambda, d_w, DELTA_t, Dmax. And these empirical functions: q (equation 11c), pFS (equation 9a), pLambda (equation 9b), beta (equation 10), Q (equation 7), and

c_broken (top of page 11).

Page 15, line 5, and throughout the paper. Dates are given in the form day/month/year, as in 01/10/2015 for 1 October 2015. Perhaps this is standard notation for The Cryosphere. Just be aware that it will confuse readers from the U.S., who will interpret "01/10/2015" as January 10, 2015. If you switch to the format "1 October 2015" it should be clear to everyone. Just a suggestion.

Page 17, lines 24-25. I can't see the convergence north of Svalbard nor the divergence at the center of the domain in Figure 11d.

Page 20 line 35 and page 21 line 1. "The sensitivity to tau_heal was investigated by re-running our experiments using this time tau_heal = 15 days..." You might want to remind readers that the default value is 25 days, because they probably won't remember (from page 11, line 27).

Page 22, lines 1-2. "waves pose a hazard as they make sea ice thicker" – this must be during freezing conditions, not during melting conditions, right?

Page 22, equation A1. What is G? What is "k" in the function N(k)? Is it supposed to be k_i?

Page 22, line 24. Is k_i,max the same thing as the quantity inside the square root on the right-hand side of equation A1? If yes, then wouldn't it make sense to first define k_i,max = max( ) (as in A1) and then lambda_break = 2*pi/k_i,max? And then go on to equations A2 and A3, if necessary?

Page 29, Figure 3. In panel a, the symbols are plotted at the mid-point of each bin. For example, the smallest bin represents floes of size 10-20 meters, and the symbol is plotted between 10 and 20 meters. But in panel b, the symbols are plotted at the left end of each bin. For example, the smallest bin represents floes of size 5-10 meters, and the symbol is plotted at 5 meters. So the data in panels a and b are not plotted consistently.

Typographical Notes

Page 2, line 13. "to conclude on" should probably be "to arrive at"

Page 5, line 6. "recovered" should be "covered"

Page 5, line 21. "the caliper diameter" should probably be "the mean caliper diameter"

Page 5, line 28. Delete the word "respectively"

Page 6, line 9. "associated to this process" should be "associated with this process"

Page 9, line 8. "B" should be "Appendix B"

Page 10, line 7. "ran" should be "run"

Page 10, line 8. Capitalize "Appendix A"

Page 10, line 27. Capitalize "Introduction"

Page 11, line 3. "in general of at least" – delete "of"

Page 11, line 22. "Wave-current [not currents] interactions"

Page 11, line 31. "similarly" should be "similar"

Page 13, line 3. "ran" should be "run"

Page 14, line 5. "Similarly" should be "Similar"

Page 14, line 28. "presented on 3" should probably be "presented in Figure 3"

Page 15, line 8. "large lambda values" – is this lambda_break?

Page 15, line 14. "CDFs (b,c)" should be "CDFs (b,d)"

Page 15, line 14. "at the time of shown" – delete "of"

Page 15, lines 18-19. "flatten the slope of the large floes regime" should be "flattening of the slope of the large floe regime"

Page 16, line 3. Delete "that"

Page 16, line 4. "16 and 60 meridians" should be "16E and 60E meridians"

Page 16, line 31. "sea ice produce" should be "sea ice produces"

Page 17, line 13. Delete "is responsible"

Page 17, line 16. "wave" should be "waves"

Page 18, line 28. "exceeds the one of the wind stress" should be "exceeds that of the wind stress"

Page 18, line 35. Something is missing after the word "REF"

Page 20, line 7. "opposes" should be "poses"

Page 20, line 27. Delete the word "a"

Page 23, lines 4 and 9. The parameter "$c_{1,FSD}$" should be "$c_{1,FS}$" (see page 9, equation 9a and following).

Page 23, line 4. "Basically, if $c_{1,\lambda} \lambda_{break} > c_{1,FS} D_{FS}$" Page 23, line 9. "Oppositely, if $c_{1,\lambda} \lambda_{break} > c_{1,FS} D_{FS}$" But the inequalities on lines 4 and 9 are the same, not opposite.

Page 24, line 10. "Tech. rep." is not enough information to locate this technical report.

Figures

Figure 2. (i) Consider labeling Point Barrow in the lower left corner of a, b, c. (ii) What are the solid and dashed curves in a, b, c? (iii) In panel b, it's almost impossible to see the green cross. (iv) In panel b, what are the black arrows? (v) In panel c, it's impossible to tell whether black represents +100 or -100. Both values are black on the color scale. (vi) In panel d or in the caption, say that the distance along the transect (km) is from north to south.

Figure 3. "Cumulated" should be "Cumulative" in the axis labels and in the caption.

Figure 4. The last sentence of the caption refers to a cross. I don't see it.

Figure 5. (i) In the caption, "cumulated" should be "cumulative". (ii) The caption should probably say that the histogram bars at 200+ meters in panels a and c represent unbroken ice.

Figure 7. In the caption and the legend, "meridian component" should be "meridional component".

Figure 8. In panel d, it's hard to tell the green arrows from the blue arrows.

Figure 9. (i) In b and d, it's impossible to tell whether black represents +0.25 or -0.25. Both values are black on the color scale. (ii) The caption says that panels a and c are "damage" but the x-axis labels in those panels say "Sea ice thickness". (iii) The caption refers to green and blue arrows in panels b and d. I don't see them.

Figure 10. In panels a and b in the legend, "DMG/WRS" should probably be "CPL_DMG".

Figure 11 (b,d) and Figure 12 (all panels). Same comment about the color scale – both ends are black. How can we distinguish the highest values from the lowest values?

Figures 2, 4, 6, 8, 9, 11, 12. Why not make all the panels larger?

―――――――――――――――――――

---

## Author Comment (AC1) · 17 Jun 2020

We thank the reviewer for their careful reading of our manuscript and for their comments and suggestions. We have tried to address their questions and concerns in our response. Our comments can be found in the attached .zip file, along with an updated version of the manuscript and a document highlighting the different changes between the two versions.

Please also note the supplement to this comment:
https://www.the-cryosphere-discuss.net/tc-2020-19/tc-2020-19-AC1-supplement.zip

---

## Author Response (AR1)

**This is a review of Boutin et al (2020) - who explore the interaction of wave fracture and improved rheological modeling in the neXtSIM model. The paper is interesting and a piece of model development that ought to be done and published. My major comments are on their 2-FSD parameterization.**

We thank the reviewer for their careful reading of our manuscript and for their comments and suggestions. We have tried to address their questions and concerns in our response. In our comments, PXLY refers to page X line Y of the updated manuscript (attached to this response).

In the updated manuscript, the main changes concern:

- The Introduction, which has been largely rewritten to clarify our motivations, and in which we shortened the description of previous FSD implementations in sea ice models, as it is not the core of our study.
- The FSD implementation section (2.2), in which we rewrote our motivation for the introduction of a second FSD to clarify its use. We also rewrote the part concerning the redistribution of the FSD to clarify the links between our model and previous studies and discuss more the assumptions we made.
- Section 4.2.1 in which the FSD is discussed more carefully following comments of reviewer #1 and #3.
- The Discussion, in which the estimation of the extent of broken ice is discussed more carefully.

**Main comment: a few questions about your model.**
**In your 2-FSD implementation, I would like some more clarity on the meaning of the "mechanical" FSD - this is an interesting idea. My read is the point is to provide memory of past deformation - but how is this separate from the role of damage in neXtSIM/MEB?**

It is indeed a good question, and we agree that we were not clear enough on this point. The short answer is that damage provides a qualitative memory of past deformations (a FSD would not be needed for that), while the mechanical FSD provides more quantitative information on the last fragmentation that occured.

The damage variable provides a qualitative estimate of the density of cracks in the ice associated with deformation events that can be due to winds, ocean currents, waves etc. Damage increases every time intense deformation events occur, and then reduces slowly with time, thus keeping a memory of previous deformation events.

It remains however a very qualitative information, useful in the case of ice dynamics, but that is very hard to transform into a more quantitative information (e.g the quantity of leads, floes, ridges in the mesh element). Conversely, the FSD provides quantitative information: a high proportion of small floes means that the density of cracks in the ice is likely to be very high, and that the damage value should be high too. In this way, it is therefore relatively easy to write a relationship between the FSD and the value of damage, but it does not work in the other way around: knowing the value of damage does not tell us much about what the FSD could look like (just like the spatial distribution of a quantity cannot be inferred from an integrated value).

Then, why do we need to keep a memory of the FSD? After a fragmentation event, once refreezing occurs, floes start to be aggregated together by welding together, or by joints of thin ice forming at the ocean surface. The ice layer that is formed might be continuous, in a sense that there is no lead, but remains heterogeneous as the mechanical properties of individual floes differ from the one of the continuous ice layer. For some processes like wave attenuation, the quantity that matters are the elasticity of the ice cover (scattering, dissipation associated to the flexion of ice) and its heterogeneity (scattering). For these processes, the length scale of interest is more likely to correspond to the one of individual consolidated floes than to the one of the continuous ice cover made of recently aggregated floes. The "mechanical FSD" keeps a memory of this information. Another motivation to implement this FSD is that in the case where a fragmentation event occurs in a sea ice cover made of recently

aggregated floe, the thin ice joining the floes is likely to break very quickly, just like at a larger scale cracks between ice plates can be re-activated. This failure of the joints will make the FSD return to a state close to the FSD resulting from the latest fragmentation event, and this is this state that the "mechanical" FSD keeps in its memory.

In order to address these comments, we have largely rewritten the beginning of section 2. to make our motivations for the introduction of the "mechanical" (that we now call "slow-growth") FSD clearer. We have also added a comment in section 2.4:

P13L1 : *Note also that floe size and damage are not explicitly linked by this relationship, but the relaxation time associated with the healing of damage and of the "slow-growth" FSD are the same, making their evolution parallel in the regions of broken ice.*

**If one wanted to compare your output to observations, how would you do that?**

The evolution of the "thermodynamical FSD" could be compared to "classical" FSD observations (from aerial photography for instance). In theory, the "mechanical" FSD is linked to mechanical properties of the sea ice cover, and could be inferred from local and repeated in-situ measurements of small-scale spatial thickness distribution in the MIZ to distinguish between "homogeneous thick floes" and "thin ice joints". The FSD observations would need to be at a sufficiently high temporal frequency (at least 1 per day) to monitor their evolution, and should be used in conjunction with information on the wave state. To evaluate the impact of fragmentation on sea ice deformation, Radar Doppler measurements of ice drift such as those proposed for future satellite missions like SKIM would be excellent; in situ drifters could also help if there were enough and if they were in the ice long enough - at least a few weeks, to be able to identify fragmentation events and determine its impact on the variability of sea ice drift.

**Why should we expect your mechanical FSD to look like the thermodynamic one, i.e. obey the same evolution equations?**

Both FSDs describe the evolution of floe size in the ice cover, but with a different definition of what the floe size is. The "thermodynamical FSD" considers the floe size as the length scale of the continuous ice cover: it ignores the heterogeneity within this ice cover, for instance if it is made of individual consolidated floes joined together by thin ice. The second "mechanical FSD" considers the floe size as the length scale of individual consolidated floes, even if they are joined by a thin ice layer. As a consequence, these two FSDs are not independent from each other, they undergo the same processes, mechanical and thermodynamical, at the same time. They should therefore follow the same general equations of evolution. Actually, these two FSD only differ after fragmentation has occurred:

- Floe size in the thermodynamic FSD will regrow quickly as the ocean surface refreezes.
- Floe size in the mechanical FSD will regrow slowly, as the healing of the cracks between individual floes takes several days to several weeks.

To make this clearer, and following the suggestions of Referee #2, we actually decided to rename the two FSD "slow-growth" and "fast-growth" instead of "mechanical" and "thermodynamical" that were misleading. We have also largely rephrased the introduction to section 2 to make this distinction between the two definitions clearer to the reader.

**Why should the mechanical healing term look like the thermodynamic one, couldn't it evolve independently? I think a figure to add would be plotting the mean floe size for both FSDs in time for the period documented in Fig 13, even for just a single grid cell.**

The two FSD are using independent healing rates, as described in the answer to the previous question. Moreover, the healing of the slow-growth (mechanical) FSD is a relaxation towards the fast-growth (thermodynamical) FSD because the two are not really independent. Mechanical healing depends on refreezing, as it only represents the additional time needed for the ice to thicken and strengthen.

We don't think that plotting the evolution of the mean floe size in the two FSDs would be of great interest here. The links between thermodynamics and the FSDs are not the main topic of our study: floe size does not impact the amount of ice that is formed, and we focus on a time period with negligible lateral melting. The timescales associated with floe size growth are discussed in section 4.1. The mean floe size evolution in a grid-cell for the slow growth FSD is equivalent to the maximum floe size Dmax, and its spatial distribution is similar to the one of Dmax (but with lower values) already shown in Fig. 4 and Fig. 6.

**The impact is clearly seen in Fig 4. Were I designing a separate depiction of sea ice fracturing, I'd expect it to be most relevant in the interior pack - this is where FSD models don't get crack features right yet. This mechanical FSD implementation seems to pinch in near the margins, not in dynamically active but waveless regions, but the neXtSIM model does get damage in the interior, doesn't it?**

Yes it does, damage in neXtSIM is defined everywhere there is ice. However, inverting the damage, or modelling the floe sizes produced by a fracturing event in the pack is an extremely difficult problem, out of the scope of our study.

**This leads me to believe that there is a difference between the description of where the mechanical FSD would be relevant (Sec 2.2, i.e. interior regions with leads) and where your model makes it relevant (exterior regions with waves). I think this approach is potentially fruitful for fixing the problem of bad pack ice fracturing, but you may be approaching it from the wrong place!**

We agree that the links between FSD and damage could be of interest for future model developments, however the focus of this study is wave-sea interactions, which are only relevant in the MIZ.

**Minor Comments:**
**Please remove the mention of eddies from your abstract - the role of the ocean is not explored here except to cite a couple papers in the conclusions.**
Done
**P2 L4 - I think you forget to explicitly mention the second main process?**

We rephrased this paragraph following your comment and a remark from Referee #2. The introduction sentence is now: *Waves can impact sea ice dynamics in the MIZ through a variety of processes.* (P2L3)

**Pg 2 - Using the power law FSD, especially in early days, is fine, but just note that meta-analyses (Herman 2011, Stern 2018) and new datasets (Horvat et al 2019) indicate an absence of power-law tails. Still... it is tough to justify (or putting the cart before the horse) designing a model that gets an answer, and then forcing its conservation via parameter choices. This is a particular problem because it is majority opinion that the "cutoff" power laws observed by Toyota and others are an artifact of the use of CDFs and finite measurement windows (Burroughs and Tebbens 2001, Stern 2018) not physics. Now a model has been designed (more than one) that produce these distributions. But you have no windowing issues (so no expectation of a truncated power law distribution) or sampling issues (so no need to produce a CDF). I'd advise plotting the FSDs proper alone (as you do in Fig 5), living with the results. At these early**

**days, you'll be forgiven for having weird distributions, and for making changes to your models, too.**

We have clarified our motivations for the redistribution towards a power-law FSD. We would however want to keep the CDFs here (while adding the comments raised by reviewers #1 and #3 on the limits of their use).
Our reason for keeping the CDFs is that they illustrate how our FSD redistribution model compares with the assumptions made previously in the studies of Dumont et al. (2011) and Williams et al. (2013), who followed the interpretation of the CDFs made by Toyota et al. (2011). This is needed here, as the FSD in neXtSIM is mostly used to provide the wave model with information on the floe size evolution when fragmentation occurs. The wave attenuation parameterization we use here has been evaluated before with a power-law FSD truncated at a cut-off floe size, it is therefore of interest to know how the FSD in our coupled model compares with these assumptions.
We have changed in the updated manuscript the way we discuss these CDFs (section 4.2.1). We don't use them to claim that our model reproduces well observations of FSD anymore, instead we discuss how our FSD redistribution scheme may affect wave attenuation compared to other wave-ice interactions studies, and in particular the FSD . We clearly mention the fact that using CDFs can be misleading to understand the FSD, and that the floe size cut-off and the distinction of two regimes are a way to interpret the CDF following Toyota et al. (2011), not necessarily a sign that the model reproduces "real" FSD well.
The paragraph introducing the FSD redistribution (2.2.2) due to fragmentation has therefore been largely rewritten, with extra-care brought to our motivations and the limits of our approach. We also added a sentence in the Discussion that reminds the reader that the "cutoff" power laws are certainly not the way to go in the future.
*P25L2: In these early days of the implementation of FSDs in sea ice models, we have built on what was done in wave-in-ice models and used a redistribution scheme that yields FSDs relatively similar to the ones described by Toyota et al. (2011), although their methods and interpretations have been contested (Stern et al., 2018; Horvat et al., 2019)*

**Pg 3 L 5 and otherwise - (ITFSD –> FSTD).**

It has been edited as suggested.

**Pg 5 L 30 it has been pointed out by Stern (and a wide literature from applied math, see Virkar and Clauset 2014) that fixed-width bins will bias your ability to represent or examine scale-invariant behavior.**

This is right, and we added this comment in the text:
*P6L24: Using fixed-width bins may bias our ability to represent or examine scale-invariant behaviour (Stern et al., 2018) but it has the advantage of being simple, and the study of the FSD evolution and its impact on sea ice is out of the scope of this study.*

**P 6 - I think the most updated Roach model was published in 2019 and included coupled wave-ice physics. Might provide better sourcing for the comparison here.**

We added this reference in the manuscript, but not in this section as the implementation we refer to is described in detail in the 2018 paper.

**P 6 L29 - do you mean that once the concentration is high, all the ice is in the highest size category? Is this also true for the mechanical FSD, or do you still require the relaxation?**

Once the concentration is 1, sea ice is supposed to cover all the area of the element and the model therefore considers that in the point of view of the thermodynamical FSD (which considers that floes are elements of ice separated by leads), all the ice is in the highest size category. However, the mechanical (slow-healing) FSD, which retains the memory of prior fragmentation events for longer, still requires the relaxation.

**P 7 L 25 "a quick and violent process" is a wonderful phrase albeit not exactly accurate. I know I should object scientifically but I really like it.**

Reviewer #2 was less sensitive to the "wonderfulness" of this sentence and objected more willingly. We now motivate our choice for the relaxation time of the "fast-growth" FSD towards the "slow-growth" FSD with the following arguments:

*P8L29: We justify this short relaxation time by the fact that (i) waves can fragment a consolidated sea ice cover in a few tens of minutes only (Collins et al., 2015) and (ii) the "fast-growth" FSD g_fast is only used for thermodynamical processes associated with timescales of at least a few hours, and is therefore relatively unaffected by the choice of a relaxation time value one order of magnitude lower.*

**P9 L 25 - See earlier comment. At the very least, please explain these parameter choices naturally through your model design not as a post-facto requirement.**

Following the comments on the CDFs by reviewers #1 and #3, we rewrote this section. The origin of all these parameters is now explained more clearly. We emphasize in particular the changes we made compared to the model introduced by Williams et al. (2013), and our motivations for these changes. It made Appendix B relatively useless, and we have therefore removed it. Instead, we have added a Table summarizing all the parameters we use in this study, as suggested by reviewer #3.

**P13 L 5 "it also includes storms" - could you be more clear about what you mean here?**

We have tried to clarify our sentence as follows:
P14L29: *This period of the year is also characterised by the combination of a low sea ice extent (thus a large available fetch) and regular occurrence of storms in the Arctic, which increases the opportunities to evaluate the impact of waves on sea ice with fragmentation events over wide areas.*

**P13 L21 - What does it mean "very satisfactory results"? What is the metric?**

The sentence has been rephrased:
P15L20: "[...] *shown to give a good match with observations for both the extent of broken ice and the wave attenuation in this particular case.*"

**P13 L 27 - "Perfectly acceptable given the uncertainties" - I'm not sure what you mean - which is perfectly acceptable, and why does this relate to wave attenuation uncertainties?**

Ice break-up is determined by wave properties, therefore the extent of broken ice depends directly on wave attenuation. In this section, we switched the order of the wave attenuation and the broken sea ice extent paragraph and rephrased to make the relationship between the two more explicit.
P16L5: *Although the extent of broken ice is slightly smaller in the coupled run, the difference does not exceed 2 grid cells, therefore representing a distance of about 25km, which is*

*acceptable given the large uncertainties associated with wave attenuation in ice (see for instance Nose et al., 2020).*

**P15 L2 - why not show this contour?**

As thickness is not a smooth field, there are a lot of very small "spots" of ice over 1m which deteriorates the readability of the panel when we plot this contour.

**P16 L1 - "It is particularly true..." - rewrite?**

This paragraph has been rephrased. The sentence is now:
P18L15: *The available fetch in particular remains relatively constant, and is large enough to allow for storm waves to penetrate far into the ice.*

**P16 L17 - A bit confused here, "regenerate unbroken ice" isn't really the process - healing between floe joints is how you describe it.**

We have substituted the word "regenerated" by "heal", as it is indeed more suitable here.

**Fig 3 - Again I'd advise not using the CDF here, preferring the FSD because as pointed out by Stern (2018) the CDF gives a false impression of scale-invariance, and P19 - I would prefer a clearer description of this process. In effect, you are saying that the influence of fragmentation (at least in your model) is not because of wave events, but after them when the sea state relaxes?**

We have added the comments to the drawbacks of using CDFs, but kept the CDFs in the plots as discussed above.
Concerning your comment on P19: Fragmentation in the model can only occur because of wave events, and this increases the damage variable which in turn can influence the ice drift. What we see from our model results is that the drift of sea ice damaged by waves is not modified during extreme events, but instead is modified after these events, when wind speed lowers but damage remains high. This is because fragmentation events in our simulations coincide with high winds, and these high winds are able to deform the ice cover whatever the internal stress of ice is. When the wind speed lowers however, sea ice deformation is only possible if the internal stress of ice is low, i.e if sea ice is not compact or has been damaged. We have rephrased the paragraph as follows:
P21L30: *In our case study, the damage added by wave-induced sea ice fragmentation does not significantly enhance sea ice deformation during wave-induced fragmentation events, but after them, when the sea state relaxes. This is because these fragmentation events coincide with high wind speeds, with wind stress dominating the internal stress of sea ice in all simulations, whatever the level of damage is. Once the wind speed lowers, the internal stress of sea ice dominates over the wind stress in places where sea ice is compact and not damaged, and limits deformation. However, in the regions that have been previously damaged by wave-induced fragmentation, the level of damage remains high in the first days following the storm, and sea ice can still deform relatively freely. This high level of damage significantly enhances sea ice mobility in the MIZ in the CPL_DMG simulation compared to CPL_WRS. This behaviour of the MIZ, with fragmentation events followed by calm periods during which sea ice mobility is enhanced, is not limited to the particular event we describe here. In the Barents Sea, for instance, maxima in the difference between ice drift velocities in the CPL_WRS and CPL_DMG simulations during October 2015 occur after maxima in the ice drift velocity magnitude (Fig. 13, and we noted a similar behaviour in the Greenland Sea (not shown).*

**P20 L 35 - "it depends on two factors" but then you mention it doesn't depend on reducing t_heal to 15 days. Also, this isn't a sensitivity experiment as you haven't also increased the healing time. You also don't really address the sensitivity to attenuation just mention it is uncertain.**

This is true, and we replaced "depends on" by "could be affected by".
We have not increased the healing time, as 25 days is already close to the upper limit of the range of values for which neXtSIM reproduces well the range of deformation (Rampal et al., 2016).
For the sensitivity to attenuation, we have rewritten the paragraph so that:

- We refer the reader to Ardhuin et al. (2018) and Boutin et al. (2018) in which sensitivity of the extent of broken ice to wave attenuation in WW3 is already extensively discussed.
- We have strengthened and clarified our discussion on the sensitivity of our results to wave attenuation, in particular the comparison with wave attenuation parameterizations used in other studies.

**This paper follows a series of recent works by the authors and others in coupling ocean waves and sea ice in large scale models. Here, the Wavewatch III and neXtSIM models are coupled, and simulations with different levels of coupling are compared, with a focus on fragmentation of the ice cover and resulting changes in ice dynamics. The main contribution of the paper is the inclusion of FSD memory, by using two FSD functions where one FSD (the "dynamic" FSD) evolves more slowly than the other one (the "thermodynamic" FSD). This is an interesting new avenue in sea ice modelling, and I'm surprised it hasn't been highlighted in the title of the paper.**

We thank the reviewer for their careful reading of our manuscript and for their comments and suggestions. We have tried to address their questions and concerns in our response. In our comments, PXLY refers to page X line Y of the updated manuscript (attached to this response).
In the updated manuscript, the main changes concern:

- The Introduction, which has been largely rewritten to clarify our motivations, and in which we shortened the description of previous FSD implementations in sea ice models, as it is not the core of our study.
- The FSD implementation section (2.2), in which we rewrote our motivation for the introduction of a second FSD to clarify its use. We also rewrote the part concerning the redistribution of the FSD to clarify the links between our model and previous studies and discuss more the assumptions we made.
- Section 4.2.1 in which the FSD is discussed more carefully following comments of reviewer #1 and #3.
- The Discussion, in which the estimation of the extent of broken ice is discussed more carefully.

**The authors motivate the memory by saying "there are reports of waves breaking ice at weak points such as refrozen leads and pressure ridges (e.g. Kohout et al 2016)" (p4, l3–4), but only ever refer to the one paper (Kohout et al, 2016). Are there any more reports of this kind? If so, give them. If not, weaken the motivating statement. In either case, it would be useful to give brief descriptions of the events reported.**

To the best of our knowledge, only Kohout et al. (2016) mention that break-up occurs at a weak point in the ice cover, and we, therefore, rephrased our motivating statement (P6L4).

We have also removed this motivating statement from the introduction to include it at the beginning of section 2.2, with more details. We think it improves the clarity of the paper. In particular, we wanted to insist on the difference between the damage variable and the "mechanical/slow-growth" FSD. The memory of previous deformation events in our model is contained in the damage variable, not in the FSD, and this is the link between fragmentation and this damage variable that we found to have an impact on sea ice dynamics. This is why we prefer to insist on the "Brittle-rheological framework" (hence the damage) in the title than on the second FSD (although we think it is an interesting addition to describe the mechanical nature of the sea ice cover).

**I found the names of the FSDs difficult, as both depend on dynamic and thermodynamic source terms. Calling them,e.g., slow and fast would be easier (and probably more accurate).**
We agree with the statement that these names were misleading. We have changed their names to "slow-growth" and "fast-growth", as it will make the reading easier.

**Another motivation given for the study is that "neXtSIM is now including a Maxwell-Elastomer-Brittle rheology" (p3, l31). But there is no explanation of why the MEB rheology is an improvement over the EB rheology for modelling marginal ice zone dynamics, or rheologies used in other models. Even if the authors don't intend this as a main motivation, they should discuss the relevance of the MEB rheology for the MIZ.**

The interesting feature about both the EB and MEB rheologies is the presence of the damage variable which can be increased for fragmented ice to make it more mobile. The MEB itself is not any better than EB, or even visco-pastic rheologies, for the MIZ. All these rheologies assume that when sea ice has a low concentration, it behaves almost in free drift. In terms of sea ice dynamics, the MIZ is therefore defined as a function of sea ice concentration. The damage variable in EB/MEB offers an easy way to account for the effects of fragmentation, and therefore to allow wave to modulate sea ice dynamics in the MIZ, in addition to sea ice concentration. We have tried to make this motivation clearer in the Introduction. We have also made clearer the differences between EB and MEB, and how it could affect MIZ dynamics (in the Introduction also).

**Wave attenuation, ice fragmentation and wave radiation stress models are very important for this study. The developments of these models, key assumptions, etc should be discussed, as all the models contain large uncertainties.**

We agree. We have therefore added modifications to answer their following comments.
**Notably, only one sentence is given to wave attenuation models in the introduction (p2, l24; not including the short sentence referencing review papers), despite uncertainties in attenuation being identified as important later (bottom of p20).**

We rewrote this paragraph of the introduction in the updated manuscript (P2L31). We have tried to give more context to the reader, and make clearer the importance of interactions between floe size and wave attenuation.
We also rewrote the paragraph concerning wave attenuation in the Discussion (P23L29).

**The scattering reference (Montiel & Squire,2017) is actually a fragmentation (or ice breakup) study (see its title) and should be used elsewhere. The scattering model used for that study is the 3D model by Montiel et al (2016), but I'm not aware of 3D scattering models being available in Wavewatch III.**

The whole paragraph has been rephrased (see our answer to the previous comment), and we now refer to Montiel et al. (2016) when we mention scattering.

**The review of fragmentation models focuses on the FSD shape, and overlooks the models used to predict if waves are capable of breaking the ice cover. The lab model by Herman et al (2018) is relevant here (noting that it is for regular, low steepness waves only), as is the lab model by Dolatshah et al (2018).**

This is actually a very good point. We added a few sentences about the uncertainties of the 1-D break-up model in the Discussion.
P24L16: *The estimation of the extent of broken ice is also likely to depend on the model used to determine the occurrence of sea ice break-up. We use a break-up model identical to most studies interested in wave-ice interactions (e.g. Williams et al., 2017; Bateson et al., 2020; Roach et al., 2019). It remains extremely simplified and assumes that break-up only occurs in the case of flexural failure in one dimension. However, recent results from laboratory experiments (Herman et al., 2018b; Dolatshah et al., 2018) tend to show that there is not such a clear relationship between the wave forcing and the floe size resulting from fragmentation. This is because a complete break-up model should include effects that are*

*currently missing (e.g. from the floe shape and size, 2-D flexure modes, floe-floe collisions, rafting).*

**Similarly, the WRS model needs to be discussed; based on Williams et al (2017), it seems to contain large uncertainties and arbitrary assumptions.**

We have added some comments on these assumptions when we introduce the WRS:
P5L12: *This computation provides an estimate of the WRS which is likely to be an upper-bound of its real value, as it assumes that all the momentum lost by attenuated waves is transferred to sea ice, therefore ignoring a potential partitioning of this momentum transfer between the ocean, sea ice and atmosphere.* [...] *As discussed by Williams et al. (2017), the estimation of the WRS and its distribution in the MIZ also strongly depends on the parameterization chosen for the wave attenuation.*

**Other comments, suggestions and questions:**
**The statement "There are two main processes through which waves can affect sea ice dynamics" (p2, l3) is far too strong. What about, e.g., collisions? Just say "we investigate two processes..." or similar.**

This is exact, and we rewrote this paragraph following this comment and the following one, see below.

**The subsequent discussion on the role of WRS needs to be more balanced, e.g. Williams et al (2017) found that "wind stress dominates the WRS", and Alberello et al found negligible WRS in a pancake ice MIZ, even during large wave events.**
We agree, and rewrote the paragraph as follows to make it more accurate:
*P2L3: Waves can impact sea ice dynamics in the MIZ through a variety of processes. For instance wave attenuation transfers momentum from waves to sea ice through the wave radiation stress (WRS, Longuet-Higgins, 1977), which acts as a force that pushes the sea ice in the direction of the incident waves. Being mostly directed on-ice, its main effect is to maintain a compact sea ice pack near the ice edge, but its importance is still discussed. Estimating wave attenuation from SAR images, Stopa et al.(2018b) estimated it to be as important as the wind stress over the first 50 km of the MIZ in the Southern Ocean, whereas Alberello et al. (2019) do not observe any wave-induced sea ice drift in pancake ice in the Southern Ocean from in situ measurements, despite a strong wave-in-ice activity. Fragmentation is also likely to change the mechanical properties of the ice, but the evolution of dynamical and mechanical properties of a sea ice cover with the floe size remains poorly understood.*

**On p2 l15, replace "fragmentation" by "floe size", as neither Shen et al (1986) nor Feltham (2005) included fragmentation in their models.**

It has been replaced as suggested.

**Saying "their FSD depends only on the wave field" (p3, l8) is not true, as D_max depends on the ice properties, as does the breaking criterion.**

We actually removed this sentence from the introduction.

**The work by Rynders (2018) is conspicuously missing from the introduction.**

We added a few sentences mentioning the approach of Rynders (2017) in the Introduction:
P3L25: *Concerning the impact of wave-induced fragmentation, Rynders (2017) suggests combining the classical elasto-visco-plastic rheology used in most sea ice models with a*

*granular rheology in the MIZ to better represent floe-floe interactions. This granular rheology depends on the floe size. Numerical simulations with this approach show an overall increase of the sea ice drift speed in the Arctic all year round compared to a reference simulation using a standard version of the sea ice model CICE (Hunke, 2010).*
**The line at the top of p6 is awkward and should be reworded.**

This sentence was reworded as:
*P6L23 Thus, floes with sizes in $[D_0\ D_1]$ cannot be broken into smaller pieces, and we refer to floes with sizes in $[D_{N-1}\ D_N]$ as unbroken floes.*

**Regarding the reference to Roach et al two lines below, please clarify if you are considering an FSD or an ITFSD?**

We are considering a FSD only. We substituted the reference to Roach by one to Zhang et al. (2015), which is more appropriate in the case of a FSD only.

**Does "lateral melt will not be discussed here" mean that it won't be included in the model for the study?**

Lateral melt is included in the model for this study, but we look at a time and a region where it does not happen. To make it clearer, we rewrite the sentence as "Note that lateral melt is included in the model in this study but will not be discussed here..."

**The opening paragraph of 2.2.2 is very long-winded for describing a simple method.**
This paragraph has been rewritten to be sharper and clearer.

**Please give a reference to back the statement "sea ice fragmentation is a violent phenomenon ... impact the floe size". This doesn't seem to allow for fatigue.**

We have rewritten this sentence to justify the quick relaxation of the "fast-growth" towards the "slow-growth" FSD. It now reads:
P8L29: *We justify this short relaxation time by the fact that (i) waves can fragment a consolidated sea ice cover in a few tens of minutes only (Collins et al., 2015) and (ii) the "fast-growth" FSD g_fast is only used for thermodynamical processes associated with timescales of at least a few hours, and is therefore relatively unaffected by the choice of a relaxation time value one order of magnitude lower.*
This idea of keeping a memory of previous fragmentation here must be distinguished from the concept of fatigue. Fatigue depends on the evolution of sea ice microstructure and there are too many things we ignore about it: how much it lowers sea ice resistance to break-up, how much bending is needed to significantly affect the sea ice, if it can heal and how long it would take. We thus do not account for fatigue in our model. We added a mention to fatigue in the discussion, when discussing the break-up model as suggested in one of the following comments
P24L23: *We also point out that while our model includes some memory of previous fragmentation events, we do not account for the fatigue of the ice when determining if break-up occurs or not. The "slow-growth" FSD is used to keep a memory of the distribution of consolidated floes. It is associated with large-scale mechanical properties of the ice cover, while fatigue is related to the micro-structure of the ice. Accounting for fatigue could significantly lower the ice resistance to flexural failure in some events (Langhorne et al., 1998).*

**Do the lines at the top of p8 mean D_N=1000m?**

No, it does not. We added the mention "in WW3" to remove the ambiguity.

**On l10, "freedom" doesn't seem to be the correct word.**

We replaced freedom by flexibility, which is more appropriate given that the shape of the FSD is indeed not free.

**Say a bit more about tau_WF below Eq 7. Is it a numerical parameter of does it have physical meaning?**

Tau_wf is mostly numerical, as its introduction mostly serves the purpose of avoiding the FSD redistribution to depend too much on the coupling time step. Physically, it also represents the fact that the fragmentation of the ice cover experiencing a constant sea state is not immediate, but rather associated with a timescale of the order 1 hour. We rewrote this paragraph and added a bit more information in the updated manuscript:
P10L10: *We introduced tau_WF to avoid dependency of the FSD redistribution to the coupling time step. It represents the timescale needed for the FSD of a fragmenting sea ice cover to reach a new equilibrium under a constant sea state. We set it to 30 minutes, as it corresponds to the timescale of the fragmentation event described in Collins et al. (2015).*

**On p8 l11, please check the interval bounds, and on l13 reword "over within which".**

These sentences have been edited as the notations (D,D', D_1, D_2) were confusing. The sentence containing "over within which" was not necessary after these edits, and has therefore been removed.

**At the bottom of p9, what exactly is being conserved?**
This is sea ice cover surface area. This sentence has been removed from the text, as we refer instead to Boutin et al. (2020) which uses the same formulation and give the details for the choices of D1 and D2.

**Please clarify the two sentences starting p10 l19. Also, is this the ice-coupled or open-water wavelength?**

We rewrote these two sentences to explain the motivation of increasing the value of Dmax sent to WW3:

P12L10: *Besides, the flexure dissipation mechanisms included in WW3 by Boutin et al. (2018) requires to discriminate between a sea ice cover made of large floes with size of the order of O(100)m and an unbroken sea ice cover for which the default Dmax in WW3 is set to 1000m. This is because flexure only occurs if the wave wavelength is shorter or of the same order as the floe size. Knowing that long swells can have wavelengths of the order of O(100m), they will only be fully attenuated by inelastic dissipation if floe size is of the order of O(1000)m, which can be larger than the floe size range covered by the FSD defined in neXtSIM. In the case where $D_N < 1000m$, to make sure that swells are still attenuated in an unbroken sea ice cover by WW3, we linearly increase the value of Dmax sent to WW3 from Dmax=$D_N$ to Dmax= 1000m with the proportion of sea ice in the largest floe size category $\int_{D_N}^{D_{N-1}} g_{slow}(D)dD/c$*
In the general case, the wavelength that we refer to here is the one relevant for waves in ice, hence the ice-coupled wavelength. However, the evolution of the wave dispersion relationship in a fragmented sea ice cover being largely unknown, we chose in our simulations to use the "open water" wavelength everywhere. This information was actually missing from the text, and we now justify this choice in the beginning of section 2:

P4L31: *Like in the study by Ardhuin et al. (2018), we assume that deviations from the ice-free wave dispersion relationship induced by the presence of ice are small and can be neglected. This is likely to be the case once sea ice has been broken (Sutherland and Rabault, 2016).*

**Should it be "fragmentation and/or refreezing" on p11 l6?**

If the reviewer refers to the sentence "This process is repeated every time fragmentation occurs in the sea ice model.", then the current sentence is correct. The process we describe only increases the value of damage. Refreezing reduces the value of damage in a way that is unchanged by the addition of waves in this study (the damage healing is described in Rampal et al., 2016).

**Has the sensitivity of the coupling time step between the wave and ice models been tested (section 3.1)?**

The sensitivity of the coupling time step on the FSD resulting from fragmentation has been tested in the Beaufort Sea test case, which led us to add the tau_wf parameter we discussed above. We found that wave attenuation and sea ice broken extent estimations were almost unchanged between stand-alone WW3 runs and coupled simulation, giving us confidence that the coupling timestep was appropriate. However, the sensitivity of the REF, CPL_WRS and CPL_DMG simulations to the coupling time step has not been investigated.

**Also, why is tau_heal set to 25 days?**

This is the default value in the model. Details are given in Rampal et al. (2016): nextsim results have little sensitivity for healing time relaxation ranging between 15 and 30 days, and using 25 days gives a very good match with observation for the temporal scaling analysis of sea ice deformation. We added the Rampal et al. (2016) reference in the text.

**You say "the other main novelties..." (p14 l11), but this is the first mention of novelty.**

We rewrote and shorten the introduction of section 4.1.2 that is now:
*P16L11 : Our coupled framework introduces two FSDs to represent the evolution of the floe size from two different points of view. It also introduces a new redistribution scheme used when wave-induced sea ice fragmentation occurs. This section provides a quick evaluation of these new features.*

**The CPL simulations first appear in 4.1.2, but in section 3.3 it says they will be used insection 4.2.**

This is true and now fixed.
*P14L26: We also evaluate the evolution of the two FSDs with refreezing/healing using the CPL_DMG simulation described in the following section 3.3*

**Explain the statement "this quick re-generation ... making welding very efficient" (p15,l8).**

Roach et al. (2018) use a constant welding rate in their welding parameterization, meaning that the reduction of the number of floes during a period delta t is independent from the floe size. As a consequence, the welding of O(10)m floes occurs at the same rate as the welding of O(1)m floes. This has consequences in pack ice, where floes are larger, hence with fewer

floes per surface area, the proportion of floes that merge during delta t is higher than at the edge, and the growth of the floe size is then controlled by the rate of welding.
We rephrased our statement to make this importance of welding in pack ice clearer:
P17L21: *In pack ice, where floes are larger than at the ice edge, the speed of the floe size growth in the "fast-growth" FSD is mostly controlled by welding, and therefore depends on the value chosen for rate of decreases of the number of floes kappa This is because, like Roach et al., (2018), we use a constant value for kappa, meaning that the fewer floes there are {i.e. the larger the floe size), the higher the proportion of floes that merge during a given time period is.*

**What "impact of waves on sea ice dynamics" is being referred to in section 5?**

We replaced wave by wave-induced fragmentation to be more specific here.

**Also, does "large fragmentation events" mean fragmentation over a large area or something else?**

Yes, fixed.

**The sentence at the end of the first paragraph of p21 is incorrect. Bennetts et al (2017) use a parameterization based on in-situ measurements by Meylan et al (2014). More generally, starting with Bennetts & Squire and Williams et al (2013a,b), it is usual to model attenuation using a viscous dissipation term for low-frequency waves and a scattering term for mid-frequency waves (see also Squire & Montiel, 2016).**

This is true, and we removed this sentence as we rewrote this paragraph as advised by the reviewer's major comments on wave attenuation.

**Check the inequalities on l4 and l9 of p23.**

There was indeed a mistake in the sense of the first inequality, but this appendix has been removed as we improved our motivations for the choices of the parameters controlling the redistribution of the FSD in the main text.

**What are the vectors in Fig 2b?**
These black arrows indicate the wave mean direction. We have added this information in the caption.

**In this work the authors coupled a wave model with a sea-ice model to investigate the impact of wave-induced sea-ice fragmentation on the sea-ice floe size distribution(FSD) and sea-ice dynamics. The focus is on the Barents Sea in October 2015. To study the FSD, five simulations are run: coupled and uncoupled runs with sea-ice thickness equal to 15 cm and 30 cm, and a coupled run with smaller floe size bins (more floe size categories). To study sea-ice dynamics, three simulations are run: one with a stand-alone sea-ice model (REF), one with wave radiative stress (CPL_WRS), and one with "damage" (CPL_DMG). The result is that waves modify sea-ice dynamics in the marginal ice zone (MIZ) by lowering the resistance of ice to deformation. The authors recommend that waves be included in sea-ice models to improve their forecasts.**

We thank the reviewer for their careful reading of our manuscript and for their comments and suggestions. We have tried to address their questions and concerns in our response. In our comments, PXLY refers to page X line Y of the updated manuscript (attached to this response).

In the updated manuscript, the main changes concern:

- The Introduction, which has been largely rewritten to clarify our motivations, and in which we shortened the description of previous FSD implementations in sea ice models, as it is not the core of our study.
- The FSD implementation section (2.2), in which we rewrote our motivation for the introduction of a second FSD to clarify its use. We also rewrote the part concerning the redistribution of the FSD to clarify the links between our model and previous studies and discuss more the assumptions we made.
- Section 4.2.1 in which the FSD is discussed more carefully following comments of reviewer #1 and #3.
- The Discussion, in which the estimation of the extent of broken ice is discussed more carefully.

**My main concern is with the FSD analysis. See page 14, lines 20-21, in reference to Figure 3: "we can distinguish two regimes separated by a cut-off floe size..." Look at Figure 3(a). I do not see two regimes separated by a cut-off floe size, and I don't believe that any statistical test would support such a conclusion. Look at the green curve for latitude 74.2 degrees north. It appears that a "line" has been fit using exactly 2 data points (see the green dashed line). By this method of analysis, one could distinguish a new "regime" for every pair of points. The purple and red dashed lines appear to be based on 3 data points. To my eye, all three curves appear to gradually steepen as the floe size increases. I don't see a cut-off or a regime shift.**

We acknowledge that the distinction between the two regimes is somewhat arbitrary. However, looking at the study by Toyota et al. (2011) who first suggest this distinction, their distributions also gradually steepen, and the existence of the two regimes and a cut-off floe size have been contested numerous times before, as raised by reviewer #1 and in the paper you suggest to reference below.
However, the question of whether FSDs follow power-laws with a cut-off floe size or not is not the topic of this paper. Whether this interpretation is wrong or not, it has been used to calibrate wave-in-ice attenuation in the wave model we are using, and it is therefore of interest to know how the FSD we produce compares with the FSD assumed in the wave model before. To this purpose, we want to know what is the exponent of a power-law FSD fitted to small floe categories (it determines the weight given to small floes in the FSD, which can impact scattering), and where Dmax is located compared to the two regimes that could be deduced by fitting two lines like in Toyota et al. (2011).
We have rewritten section 2.2.2 that introduces the redistribution of the FSD to make our motivations clearer. We detail what are the evolutions brought by our study compared to

previous wave-in-ice models using FSDs. In section 4.2, we do not claim that the FSDs we produce are realistic, as there is no consensus about what should be the shape FSD resulting from wave-induced fragmentation. Instead, we present the FSDs we get and still fit lines to the small floes categories, but in order to discuss how our FSDs compare with the fixed-exponent power-law FSDs assumed for small floes in previous waves-in-ice studies and the observations reported by Toyota et al. (2011).

**The authors cite Toyota et al (2011) numerous times in the context of concave-down cumulative distribution functions (CDFs) with two regimes. A counterpoint may be found in this paper:**
**Stern, H.L., A.J. Schweiger, J. Zhang, and M. Steele, 2018. On Reconciling Dis-parate Studies of the Sea-Ice Floe Size Distribution, Elem Sci Anth, 6: 49. DOI:https://doi.org/10.1525/elementa.304In particular, see their Figure 3 and the section called "Break-point analysis".**

References to this paper have been added in various places of the text, as it is a nice reminder of the strong assumptions made for the FSD in this study (and in all other wave-ice interactions models based on Toyota et al., 2011).

**Page 23, Appendix B. "The shape of the CDFs shown in Figures 3 and 5 strongly depend on the parameterization detailed in section 2.2.2. The value of the cut-off floe size at which the transition between the small and large floes regime happens..." It seems highly undesirable that the shapes of the CDFs depend strongly on the parameterization. This would seem to inject a high degree of uncertainty into the whole simulation. And again, I question that a well-defined cut-off exists between small and large floes.**

We agree with this statement. What we meant here is that in the absence of a consensus concerning the shape of the FSD, and with the little knowledge we have of the physics of sea ice break-up due to waves, the shape of the redistributed FSD depends on the hypothesis made in the redistribution process. These hypotheses are however necessary at this stage, and the ones we make are almost the same as the ones in the model by Williams et al. (2013) and have been re-used in many wave-ice interactions studies since. These hypotheses originated from the work by Toyota et al. (2011), and as noted in the previous comments, have been contested since. The only differences with the model by Williams et al. (2013) are that:

- Instead of having a well-defined cut-off floe size with a sharp steepening of the CDF, we have a progressive steepening of the CDF, which is more coherent with the observations reported by Toyota et al. (2011). A sharp steepening of the CDF like in Williams et al. (2013) is all the more unsatisfying as the steepening reported by Toyota et al. (2011) might be the result, at least partly, of windowing issues, as raised in the previous comment. To obtain a more progressive steepening of the CDF, we introduce a continuous function for the probability that an ice floe breaks up instead of a step function in the model by Williams et al. (2013). The steepening of the CDF depends on the values of $c_{2,FS}$ and $c_{2,\lambda}$, which are the only two new coefficients introduced by our study. The model by WIlliams et al. (2013) is equivalent to having $c_{2,FS}$ and $c_{2,\lambda}$ tending towards 0. We found that setting $c_{2,FS}$ and $c_{2,\lambda}$ to 2 was a good compromise between a progressive steepening of the CDF and coherence with the truncated power-law FSD used to calibrate the wave model.
- Williams et al. (2013) assume that the FSD of the "small floe regime" follows a power law with a constant exponent set to $\approx$-1.85. This value originates from the work by Toyota et al. (2011) by assuming that, if waves can trigger flexural failure, then the probability that a floe breaks up is always 0.9. As written above, we already introduced a continuous function for the probability that an ice floe breaks up. Therefore, we

substituted the value of 0.9 by our probability function. As a result, the exponent that we obtain when fitting a power-law to the "small floe" regime is allowed to vary, just like in the observations reported by Toyota et al. (2011).

The section 2.2.2 has been largely rewritten to clarify the choices made for the values of our parameters, and to make more apparent the links between our parameterization and the one initially described by Williams et al. (2013), including the comments above. As the hypotheses we use for the FSD redistribution are still mostly based on the work by Toyota et al. (2011), we insist on the potential caveats of this study, and in particular the fact that it is unclear whether a well-defined cut-off exists or not. As we describe in more detail the role of each parameter in the redistribution, Appendix B was found to be useless, and we have therefore removed it. We have also rewritten paragraphs in section 4.1, and now we only use the CDFs to discuss how the changes we introduced may affect wave attenuation compared to the FSDs assumed previously in wave attenuation models.

**Minor Comments**
**Page 1, line 25. It looks like Lemieux et al (2016) is about landfast ice, not the sea-ice edge.**

Yes, we were actually thinking about another paper by Lemieux et al. (2016) focusing on a Regional ice prediction system. We eventually found a publication by Shweiger & Zhang (2015) that was more appropriate here.
Schweiger, A. J. and Zhang, J.: Accuracy of short-term sea ice drift forecasts using a coupled ice-ocean model, Journal of Geophysical Research: Oceans, 120, 7827–7841, https://doi.org/10.1002/2015JC011273, https://agupubs.onlinelibrary.wiley.com/doi/abs/10.1002/2015JC011273, 2015.

**Page 6, line 21. "floes in the largest floe category are not affected by lateral melt." I don't see how equation (4) reflects this statement.**

We only mentioned it in the text, as adding the very special case of this category to equation (4) would deteriorate its readability in our opinion. Besides, this is only a choice we made here as we are not interested in resolving large floes of size >O(100m), the general case remains well described by equation (4) as it is. We rephrased the sentence to make our motivations clearer:
P7L12: *Here, we neglect lateral melt for the largest floe size category as floes with size O(100) m and more are not resolved in this study and are expected to contribute very little to lateral melt.*

**Page 7, lines 2-3. "a uniform FSD made of the smallest floes ... evolves into a uniform FSD made of the biggest possible floes." This does not make sense. A uniform FSD contains floes of all sizes, in equal proportions. The authors probably mean a delta-function FSD, in which all floes are of the smallest size, evolves into a delta-function FSD, in which all floes are of the largest size.**
**And**
**Page 7, line 4. Check whether "uniform FSD" is appropriate here – see previous comment.**

This is true, and we changed our formulation following the referee's comment.

**Page 7, line 6. "setting kappa = 5 x 10ˆ(-8)" kappa is a rate (see line 1 of page 7).Please give the units**

The units (m-2 s-1) have been added to the text.

**Page 8, equation (8) and following. You need to say that Y is Young's modulus, nu is Poisson's ratio, and h is ice thickness. Please give values of DFS for h = 15 cm and h= 30 cm.**
We added the missing variables to the text, as well as a comment on values of DFS (P10L6).

**Page 10, equation (12). This equation is not correct – it is missing a factor of D inside the integral. If g(D) is a probability density function then the mean value of D is the integral of D*g(D) dD.**

This is right, it has been corrected. It was only the case in the text, not in the model.

**Page 10, lines 21-24. Dmax is supposed to be one order of magnitude larger than the longest wavelength, but lines 23-24 imply that Dmax does not become larger than 1000m. Shouldn't Dmax be 10 times larger than 1000m?**

Wavelengths associated with storm swells are in general of the order O(100)m, setting Dmax to 1000m for unbroken sea ice ensures they are dissipated by flexion dissipation. We actually rewrote this part of text to make these motivations clearer.

P12L10: *Besides, the flexion dissipation mechanisms included in WW3 by Boutin et al. (2018) require to discriminate between a sea ice cover made of large floes with size of the order of O(100)m and an unbroken sea ice cover for which the default Dmax in WW3 is set to 1000m. This is because flexion only occurs if the wave wavelength is shorter or of the same order as the floe size. Knowing that long swells can have wavelengths of the order of O(100m), they will only be fully attenuated by inelastic dissipation if floe size is of the order of O(1000)m, which can be larger than the floe size range covered by the FSD defined in neXtSIM. In the case where* $D_N$<1000m, to make sure that swells are still attenuated in an unbroken sea ice *cover by WW3, we linearly increase the value o fDmax sent to WW3 from Dmax=$D_N$ to Dmax= 1000m with the proportion of sea ice in the largest floe size category* $\int_{D_N}^{D_{N-1}} g_{slow}(D)dD/c$

**Page 11, end of Section 2. There are a LOT of parameters and empirical functions in this work. It might help to collect them in a table. My list includes these parameters: Gr, c_new, and beta_weld from equation (4); kappa from page 7; tau_heal from equation (5); tau_WF from equation (7); lambda_break, c_1FS, c_2FS, c_1Lambda,c_2Lambda, d_w, DELTA_t, Dmax. And these empirical functions: q (equation 11c),pFS (equation 9a), pLambda (equation 9b), beta (equation 10), Q (equation 7), and c_broken (top of page 11).**

We have collected all these parameters and others in a table that we added in an appendix (Appendix A1).

**Page 15, line 5, and throughout the paper. Dates are given in the form day/month/year, as in 01/10/2015 for 1 October 2015. Perhaps this is standard notation for The Cryosphere. Just be aware that it will confuse readers from the U.S., who will interpret "01/10/2015" as January 10, 2015. If you switch to the format "1 October 2015" it should be clear to everyone. Just a suggestion.**

We followed this suggestion. It also seems to be what is recommended by the journal guidelines.

**Page 17, lines 24-25. I can't see the convergence north of Svalbard nor the divergence at the center of the domain in Figure 11d.**

This is true, convergence and divergence of sea ice can be seen on Fig.11c, not d. Reference to this panel has been added in the text.

**Page 20 line 35 and page 21 line 1. "The sensitivity to tau_heal was investigated by re-running our experiments using this time tau_heal = 15 days..." You might want to remind readers that the default value is 25 days, because they probably won't remember (from page 11, line 27)**

We rephrased Page 20 line 35 to give a reminder and a bit more context to the reader. P23L23*: [...] Its impact was investigated by re-running our experiments using this time $\tau_{heal}$=15 days instead of 25 days, the default value in neXtSIM. 15 days corresponds to the lower limit for which neXtSIM reproduces well the multi-scaling of sea ice deformation (Rampal et al., 2016), while 25 days is close to the upper-limit of this range.*

**.Page 22, lines 1-2. "waves pose a hazard as they make sea ice thicker" – this must be during freezing conditions, not during melting conditions, right?**

This is right, and we actually removed this reference to thickening in the sentence.

**Page 22, equation A1. What is G? What is "k" in the function N(k)? Is it supposed to be k_i?**

Appendix A has been removed as it was adding more confusion than referring to section 2.3 of Boutin et al. (2018), which is a step by step description of the break-up process in WW3.

**Page 22, line 24. Is k_i,max the same thing as the quantity inside the square root on the right-hand side of equation A1? If yes, then wouldn't it make sense to first define k_i,max = max( ) (as in A1) and then lambda_break = 2*pi/k_i,max? And then go on to equations A2 and A3, if necessary?**

We thank the reviewer for this remark as it made us realize that (i) the definition of lambda break we gave was wrong (it corresponds to the shortest wavelength for which the wave-induced stress exceeds sea ice resistance to flexural failure) and (ii) this section contained a few mistakes and was actually quite misleading. We decided to remove Appendix A from the manuscript, and to instead refer to section 2.3 of Boutin et al., 2018 that explains the determination of lambda_break with the right level of details.

**Page 29, Figure 3. In panel a, the symbols are plotted at the mid-point of each bin. For example, the smallest bin represents floes of size 10-20 meters, and the symbol is plotted between 10 and 20 meters. But in panel b, the symbols are plotted at the left end of each bin. For example, the smallest bin represents floes of size 5-10 meters, and the symbol is plotted at 5 meters. So the data in panels a and b are not plotted consistently.**

We updated the two panels to make the plotting of our data consistent. The computation of the exponents of the fitted power-laws was also not consistent between the two panels and has therefore been redone. The changes in the new exponent values we obtain are quite small and do not require modifications in the text.

**Typographical Notes**

**Page 2, line 13. "to conclude on" should probably be "to arrive at"**
Edited
**Page 5, line 6. "recovered" should be "covered"**
Edited
**Page 5, line 21. "the caliper diameter" should probably be "the mean caliper diameter"**
Edited
**Page 5, line 28. Delete the word "respectively"**
Edited
**Page 6, line 9. "associated to this process" should be "associated with this process"**
Edited
**Page 9, line 8. "B" should be "Appendix B"**
Edited
**Page 10, line 7. "ran" should be "run"**
Edited
**Page 10, line 8. Capitalize "Appendix A"**
Edited
**Page 10, line 27. Capitalize "Introduction"**
Edited
**Page 11, line 3. "in general of at least" – delete "of"**
Edited
**Page 11, line 22. "Wave-current [not currents] interactions"**
Edited
**Page 11, line 31. "similarly" should be "similar"**
Edited
**Page 13, line 3. "ran" should be "run"**
Edited
**Page 14, line 5. "Similarly" should be "Similar"**
Edited
**Page 14, line 28. "presented on 3" should probably be "presented in Figure 3"**
Edited
**Page 15, line 8. "large lambda values" – is this lambda_break?**
The whole sentence has been rephrased:
P17L21: *In pack ice, where floes are larger than at the ice edge, the speed of the floe size growth in the "fast-growth" FSD is mostly controlled by welding, and therefore depends on the value chosen for rate of decreases of the number of floes kappa.*
**Page 15, line 14. "CDFs (b,c)" should be "CDFs (b,d)"**
Edited
**Page 15, line 14. "at the time of shown" – delete "of"**
Edited
**Page 15, lines 18-19. "flatten the slope of the large floes regime" should be "flattening of the slope of the large floe regime"**
Edited
**Page 16, line 3. Delete "that"**
Edited
**Page 16, line 4. "16 and 60 meridians" should be "16E and 60E meridians"**
Edited
**Page 16, line 31. "sea ice produce" should be "sea ice produces"**
Edited
**Page 17, line 13. Delete "is responsible**
Edited
**Page 17, line 16. "wave" should be "waves"**
Edited

**Page 18, line 28. "exceeds the one of the wind stress" should be "exceeds that of the wind stress"**
Edited
**Page 18, line 35. Something is missing after the word "REF"**
Edited
**Page 20, line 7. "opposes" should be "poses"**
Edited
**Page 20, line 27. Delete the word "a"**
Edited
**Page 23, lines 4 and 9. The parameter "c_1,FSD" should be "c_1,FS" (see page 9, equation 9a and following).**
This sentence has been removed as details on the role of $c_{1,FS}$ are now given in section 2.2.2.
**Page 23, line 4. "Basically, if c_1,lambda lambda_break > c_1,FS D_FS"**
This sentence has been removed as details on the role of $c_{1,FS}$ are now given in section 2.2.2.
**Page 23, line9. "Oppositely, if c_1,lambda lambda_break > c_1,FS D_FS" But the inequalities onlines 4 and 9 are the same, not opposite.**
This sentence has been removed as details on the role of $c_{1,FS}$ are now given in section 2.2.2.

**Page 24, line 10. "Tech. rep." is not enough information to locate this technical report.**

We replaced this reference by a more recent one:
Yumashev, D., van Hussen, K., Gille, J. et al. Towards a balanced view of Arctic shipping: estimating economic impacts of emissions from increased traffic on the Northern Sea Route. Climatic Change 143, 143–155 (2017). https://doi.org/10.1007/s10584-017-1980-6

**Figures**
**Figure 2. (i) Consider labeling Point Barrow in the lower left corner of a, b, c. (ii) What are the solid and dashed curves in a, b, c? (iii) In panel b, it's almost impossible to see the green cross. (iv) In panel b, what are the black arrows? (v) In panel c, it's impossible to tell whether black represents +100 or -100. Both values are black on the color scale. (vi) In panel d or in the caption, say that the distance along the transect (km) is from north to south.**

(i) Done
(ii) They represent contours of sea ice concentration equal to 0.8 and 0.15 respectively.
(iii) All crosses have been made larger and bigger to be more visible.
(iv) They represent the wave mean direction. It is now stated in the caption.
(v) We have truncated the divergent color scale at both ends. Extreme values now correspond to lighter blue and red, which improves the readability of our figures.

**Figure 3. "Cumulated" should be "Cumulative" in the axis labels and in the caption.**
Edited.

**Figure 4. The last sentence of the caption refers to a cross. I don't see it.**
The cross has been made bigger and we now mention the panels where it can be seen.

**Figure 5. (i) In the caption, "cumulated" should be "cumulative". (ii) The caption should probably say that the histogram bars at 200+ meters in panels a and c represent un-broken ice.**

(i) and (ii) : We have edited the caption as suggested.

**Figure 7. In the caption and the legend, "meridian component" should be "meridional component".**

Edited.

**Figure 8. In panel d, it's hard to tell the green arrows from the blue arrows.**

The arrows are now blue and red. They are also bigger, slightly less numerous, and over a green color scale. It should be easier to read.

**Figure 9. (i) In b and d, it's impossible to tell whether black represents +0.25 or -0.25. Both values are black on the color scale. (ii) The caption says that panels a and c are"damage" but the x-axis labels in those panels say "Sea ice thickness". (iii) The caption refers to green and blue arrows in panels b and d. I don't see them.**

(i) We have truncated the divergent color scale at both ends. Extreme values now correspond to lighter blue and red, which improves the readability of our figures.
(ii) It has been corrected.
(ii) This sentence was in the wrong place, we removed it. There are no arrows in panels b and d.

**Figure 10. In panels a and b in the legend, "DMG/WRS" should probably be"CPL_DMG".**
Edited.

**Figure 11 (b,d) and Figure 12 (all panels). Same comment about the color scale – both ends are black. How can we distinguish the highest values from the lowest values?**

We have truncated the divergent color scale at both ends. Extreme values now correspond to lighter blue and red, which improves the readability of our figures.

**Figures 2, 4, 6, 8, 9, 11, 12. Why not make all the panels larger?**

The size of the figures correspond to the one prescribed by the template provided by Copernicus.

[revised manuscript text omitted]

As it is, the  "slow-growth" FSD therefore relaxes to the  "fast-growth" FSD over a time $\tau_{\text{heal}}$, representing the (slow) strengthening of the joints between the floes. Note that this healing only occurs if the sea ice is exposed to freezing conditions. In melting conditions, we assume that the shape of the  "slow-growth" FSD is not affected by lateral melt.

**2.2.2  Wave-induced sea ice fragmentation**

In this section, we describe the implementation of the terms  $\Phi_{\text{m,slow}}$ and $\Phi_{\text{m,fast}}$ that represent the mechanical redistribution of floes associated with the fragmentation of sea ice by waves in each FSD. ~~As mentioned before, during a wave event, floes are likely to break at their weakest point, i.e the freshly refrozen joints between floes (Kohout et al., 2016). In the following, we assume that when waves are able to break sea ice, the old fractures in the ice cover are immediately re-activated. To represent this process, the mechanical redistribution is first performed in the mechanical FSD $g_{\text{mech}}$, in which the growth of the floe size is slower than in the thermodynamical FSD, and that has therefore kept a memory of previous fragmentation events. Besides, because sea ice fragmentation is a quick and violent phenomenon, it only needs a few minutes to impact the floe size. It is therefore likely to overcome all the thermodynamical processes going-on, as they are associated with longer timescales. When fragmentation occurs, we therefore revert the thermodynamical FSD $g_{\text{thermo}}$ to $g_{\text{mech}}$. This is done by setting $\Phi_{\text{m,thermo}}\Delta t = g_{\text{thermo}} - g_{\text{
[revised manuscript text omitted]

in which $\tau_{\text{WF}}$ is a relaxation time associated with wave-induced fragmentation events, used to remove dependency on the time step, and $p_{\text{FS}}$ and $p_\lambda$ are probabilities that floes break depending on their size. We set introduced $\tau_{\text{WF}}$ to avoid dependency of the FSD redistribution to the coupling time step. It represents the timescale needed for the FSD of a fragmenting sea ice cover to reach a new equilibrium under a constant sea sate. We set it to 30 minutes, as it corresponds to the timescale of the fragmentation event described in Collins et al. (2015).

In the absence of known relationships linking wave and sea ice properties to floe break-up probabilities, we use The probability functions $p_{\text{FS}}$ and $p_\lambda$ to express the idea that the smaller the floes are, the less chance they have to break. The function $p_{\text{FS}}$ $p_\lambda$ compares the floe size $D$ with the minimum floe size for which flexural failure can occur, $D_{\text{FS}}$ (see Mellor, 1986), which is computed as:

$$D_{\text{FS}} = \frac{1}{2}\left(\frac{\pi^4 Y h^3}{48\rho g(1-\nu^2)}\right)^{1/4}$$

where $g$ is gravity and $\rho$ is the density of sea water. The value of $p_{\text{FS}}$ therefore only depends on sea ice properties. Similarly, the function value of $D_{\text{FS}}$, and $p_\lambda$ compares the floe size $D$ with the wave wavelength associated with the highest stress

experienced by sea ice $\lambda_{\text{break}}$, introducing a dependency of $Q(D)$ on the wave field. These two probabilities are heuristically defined asThe difference with the model Williams et al. (2013b) is that instead of step functions, we use hyperbolic tangents to get a continuous transition of $Q(D)$ between 0 and 1:

$$p_{\text{FS}}(D) = \max\left(0, \tanh\left(\frac{D - c_{1,\text{FS}}D_{\text{FS}}}{c_{2,\text{FS}}D_{\text{FS}}}\right)\right), \tag{9a}$$

5  $$p_\lambda(D, \lambda_{\text{break}}) = \max\left(0, \tanh\left(\frac{D - c_{1,\lambda}\lambda_{\text{break}}}{c_{2,\lambda}\lambda_{\text{break}}}\right)\right), \tag{9b}$$

in which $c_{1,\text{FS}}$, $c_{2,\text{FS}}$, $c_{1,\lambda}$, and $c_{2,\lambda}$ are tuning parameters $c_{2,\text{FS}}$ are parameters of the FSD that control the range of floe size that will be broken or not. The use of a continuous $Q(D)$ instead of a step function aims to relax the constraint on the FSD shape imposed by Williams et al. (2013b). With a step function, the probability of having floes larger than the cut-off floe size is 0, i.e above the cut-off floe size, the FSD is suddenly infinitely steep. This approach is particularly problematic, as firstly the FSDs

10  reported in Toyota et al. (2011) rather show a gradual steepening than a sharp transition, and secondly the steepening of the FSD slope that led to the identification of the two floe size regimes by Toyota et al. (2011) could actually be due to windowing issues (Stern et al., 2018). Here, instead of having a single cut-off floe size, we have a transition occurring in the floe size range for which $0 < Q(D) < 1$. The width of this range is controlled by $c_{2,\text{FS}}$ and $c_{2,\text{FS}}$. As floes smaller than $D_{\text{FS}}$ cannot be broken, we $
[revised manuscript text omitted]

---

## Referee Report (RR1)

This is a second review of Boutin et al (2020) - who look at the interaction of wave fracture and improved rheological modeling in the neXtSIM model. The paper was substantially revised between the first and second revisions, largely following comments made by the set of reviewers. The literature review has been improved and in general I am happy with the way comments were received. With the central premise of the model developments aside (the dual FSD), the remainder of the paper seems appropriately constructed and discussed.

Now that the explanation has been improved, the challenges with the second FSD are clearer to me. The authors need to be wary of including this "intuitive" parameterization without presenting it rigorously. This can lead to confusion for users and readers, and make extending this work a challenge in the future. Or, it can lead to counterintuitive results, one of which I highlight here:

**Evolution of the dual-FSD model under thermodynamic-only forcing**   The actual evolution of the second FSD can be determined within some limits. I tried to do this using the available text, and found some typos. For example in Eq. (5), those Ds should not have subscripts, I think - as $\Phi_{th,slow}$ should be $\Phi_{th,slow}(D)$. Anyways, in a limiting case we can take u=0, $\Phi_m$=0, and observe:

$$\frac{\partial}{\partial t}\left(g_{fast} - g_{slow}\right) = -\frac{g_{fast} - g_{slow}}{\tau} + \Phi_{th,fast}. \tag{1}$$

Generally speaking, the implementation of lateral melt in CICE/LIM/neXtSIM follows,

$$\int_{\mathbf{r}^+} \Phi_{th,fast} \propto \alpha Q, \tag{2}$$

where $Q$ is the heat available to the sea ice, and $\alpha$ is the proportion of that heat flux that goes to lateral processes. Then integrating over all $r > 0$, and noting as in your eq. (1)-(2) that the integral of $g$ over all positive $r$ is the sea ice concentration,

$$\frac{\partial}{\partial t}\left(c_{fast} - c_{slow}\right) = -\frac{c_{fast} - c_{slow}}{\tau} + \alpha Q^*. \tag{3}$$

where we use $c_{fast/slow}$ to denote the integral of $g_{fast/slow}$ over positive $r$, and $Q^*$ has suitable units. Unless the RHS of that equation is zero always, which can only happen if $\tau$ and $\alpha$ are variables instead of parameters, there are clearly two concentrations that evolve independently - not just two FSDs! This will be a particular problem when you have a wave event, which essentially resets $g_{fast}$ to $g_{slow}$ over a model timestep - since they have a different concentration, you will add or lose sea ice concentration instantly because of a mechanical process that should preserve concentration.

This type of restoring equation (in that case, advection from a stationary distribution) was examined in detail in *Horvat and Tziperman* (2017), Sec. 3.1, and the consequences are significant: all moments of these two FSDs, not just the concentration, will have different evolutions over time even if they are apparently identical!

While it might be true that the two observational definitions of the FSD might also have different concentrations, I believe this would only be that the "slow" sea ice concentration is less than the "fast" sea ice concentration, as it might not count the frazil between

floes as sea ice. But here the slow concentration *lags* the time evolution of the fast concentration. When melting, the slow ice will have a higher sea ice concentration than the "fast" ice.

Here are my main suggestions:

1. Add a figure clearly explaining the difference between these two FSDs. In the text you comment that SAR imagery offers motivation, but without a demonstration of this point. Please do so, so that the reader can understand the choice.

2. Proofread the math in Section 2.2., adding a section explaining the influence of this second FSD on quantities that arise from the FSD - I would do this in the context of conserved quantities so you know nothing crazy is going to happen - and consider adding terms to your Eq.s (3)-(5) that account for this lack of conservation.

3. Evaluate the lack of conservation in your two FSDs and put that in the text. If you integrate them, do both yield the area/volume? If they are normalized to one in your code, are the equations you are using to evolve them suitably normalized as well? This is a trickier problem than it might appear.

I look forward to discussing further, and encourage the authors to contact me with any questions. I again find this an interesting approach - it makes plenty of sense! But the actual equations being solved need to be analyzed with a bit more rigor because the consequences might be more significant than they appear at first glance and should be constrained some.

Chris Horvat

**References**

Horvat, C., and E. Tziperman (2017), The evolution of scaling laws in the sea ice floe size distribution, *Journal of Geophysical Research: Oceans*, *122*(9), 7630–7650, doi: 10.1002/2016JC012573.

---

## Author Response (AR2)

We thank the referees for their thorough reading of our manuscript.

We have fixed the text following the suggestions and corrections from the two anonymous referees. We also wanted to answer 2 comments from referee #2:

- Clarify if welding and cementing are the same:

We now only use the word *cementing* when we refer to the thin ice between consolidated floes. This thin ice acts as the cement in the welding process. It should be clearer this way.

- Add a reference to support the sentence ending on line 7/p6;

We have removed this sentence, as explained below in our answer to C. Horvat.

We would also like to thank C. Horvat for his suggestions and careful proofreading of our equations. Please find below our answers and comments.

**1. Add a figure clearly explaining the difference between these two FSDs. In the text you comment that SAR imagery offers motivation, but without a demonstration of this point. Please do so, so that the reader can understand the choice.**

We think the addition of a picture illustrating the difference in the definition of the floe size distribution depending on what definition is used for a floe could indeed help the reader to understand the concept of using 2 FSDs. We suggest adding the following picture with a comment in section 2.2. We removed the sentence mentioning the SAR imagery: it was used to make an analogy between the behaviour of sea ice in pack ice and in the MIZ, which was in the end quite confusing. Our intuition that recently welded floes will be fragmented along the not-yet-consolidated joints (where the ice resistance to deformation is likely to be the lowest) is therefore mostly supported by the observation of Kohout et al. (2016). Note that cracks around the consolidated floes are visible in the thin ice on Fig. 2.

[Figure]

Figure 2 (in the manuscript): Aggregate of ice floes cemented together by thin ice in the Marginal Ice Zone (Weddell Sea, Antarctica). The size of the largest floes is of the order of 10m. Picture taken on board of RRS James Clark Ross in March 2014. Credit: *Heather Regan*

*P6L1: The picture in Figure 2 illustrates the different definitions of the floe size in our study. On the one hand, sea ice concentration is about 100% with a continuous sea ice cover, represented by the fast FSD with unbroken floes. Processes like lateral melting are unlikely to occur in these conditions. On the other hand, consolidated floes in Figure 2 are easy to distinguish from the thin ice joining them. The "slow-growth" FSD represents the distribution of sizes of these consolidated floes. In this case, the "slow-growth" FSD will be dominated by small floes (of the order of ten metres). This information can be useful for the study of mechanical processes. For example, it can represent the inhomogeneous nature of the ice cover, which is particularly relevant for wave attenuation processes like scattering and flexure-induced dissipation, for which the mechanical properties and ice thickness continuity of the ice cover are the quantities of interest.*

*P6L14: An ice cover such as illustrated in Figure 2 is likely to have a very different mechanical strength/response under external stresses compared to a consolidated continuous ice cover, due to the high likelihood of break-up of the thin ice between the consolidated floes. In the case of a new wave event, the fragmentation of the ice cover is likely to occur at its weakest points, hence the thin ice joints.*

**2. Proofread the math in Section 2.2., adding a section explaining the influence of this second FSD on quantities that arise from the FSD - I would do this in the context of conserved quantities so you know nothing crazy is going to happen - and consider adding terms to your Eq.s (3)-(5) that account for this lack of conservation.**

Quantities like sea ice area and volume are well conserved by the model, it has been thoroughly checked during the simulations. We proofread the maths in the manuscript: indeed, our presentation in the manuscript is wrong and is missing a term. Yet that term is taken into account properly in our code. This term goes in Eq. 5 and ensures the conservation of sea ice area. It is an ad hoc term to ensure that the slow FSD is affected by changes in sea ice area due to thermodynamical gain of ice (in which case all the new ice is added to the largest floe size category) or loss of ice (in which case the distribution is scaled in order to keep its shape). We have added this term and explained its meaning in the manuscript in section 2.2.1.

**3. Evaluate the lack of conservation in your two FSDs and put that in the text. If you integrate them, do both yield the area/volume? If they are normalized to one in your code, are the equations you are using to evolve them suitably normalized as well? This is a trickier problem than it might appear.**

As mentioned in our answer to the previous suggestion, the sea ice area and volume are conserved and identical for the two FSDs, which only describe how the sea ice area is distributed among floe size and are normalized in the code. We understand that the reviewer is concerned by how this normalization and the terms ensuring the conservation of sea ice area in the slow FSD, as well as the relaxation of one FSD towards the other, could affect the floe size parameters computed from the moments of this FSD. The relaxation of the FSD and the ad hoc term to ensure the conservation of sea ice area represent indeed physical processes involved in the evolution of the floe size that cannot be constrained by observations for now. However, as shown by Horvat et Tziperman (2017), these terms could affect the evolution of the moments of the FSD in a non-trivial way.

We have therefore added comments in the text highlighting these concerns in the context of the future studies we want to conduct. We also plot below the evolution of the floe size parameters derived from the moments of the slow FSD for a particular location, following a suggestion from the first review. The evolution of the average floe size and the maximum floe size are very similar, as mentioned in our first answer to the reviews.

[revised manuscript text omitted]